# TechKG: A Large-Scale Chinese Technology-Oriented Knowledge Graph

## Abstract

Knowledge graph is a kind of valuable knowledge base which would benefit lots of AI-related applications. Up to now, lots of large-scale knowledge graphs have been built. However, most of them are non-Chinese and designed for general purpose. In this work, we introduce TechKG, a large scale Chinese knowledge graph that is technology-oriented. It is built automatically from massive technical papers that are published in Chinese academic journals of different research domains. Some carefully designed heuristic rules are used to extract high quality entities and relations. Totally, it comprises of over 260 million triplets that are built upon more than 52 million entities which come from 38 research domains. Our preliminary experiments indicate that TechKG has high adaptability and can be used as a dataset for many diverse AI-related applications.

## 1 Introduction

Generally, a knowledge graph (KG) stores factual knowledge in the form of structural triplet like $<h, r, t>$, in which $h$ and $t$ are entities (called head entity and tail entity respectively) and $r$ denotes the relation type from $h$ to $t$. KG is a kind of valuable knowledge base and is important for many different kinds of AI-related applications. Nowadays, great achievements have been made in building large scale KGs. And there are many available KGs like *FreeBase* Bollacker et al. (2008), *WordNet* Miller (1995), *YAGO* Suchanek et al. (2007), etc. Usually there are millions of entities and billions of relational facts in these KGs. With them, great progress have been made in lots of AI-related applications, such as knowledge graph embedding (KGE), distantly supervised relation extraction (RE), knowledge based question and answering (KBQA), etc. However, there are still following two challenges inherited in these KG-dependent applications.

First, almost all of these KGs can only be used in English-related applications. But the fact is that each language has its own characteristics. It is uncertain whether a model that performs well in an English-related KG-dependent task could still perform well in the corresponding task of other language. Thus there is a pressing need to build KGs of different languages other than English.

Second, most of existing KGs are designed for general purpose. It is also uncertain whether a model that performs well in an application of general domain could still perform well in applications of specific domains, such as in the technical domains liking *medicine*, *biology*, *electronics*, etc. It is necessary to evaluate whether there are some new kinds of characteristics hidden in KGs that haven't been noticed by researchers up to now.

To address these issues, this work introduces TechKG, a large scale Chinese KG built from massive technical papers that are published in Chinese academic journals of different research domains. Compared with existing KGs, TechKG has following important characteristics.

First, TechKG is a large scale Chinese KG that is technology-oriented. Totally, it comprises of over 280 million triplets that are built upon more than 60 million entities which come from more than 30 research domains. To the best of our knowledge, this is the first Chinese KG that is built by using multi-domain technical papers.

Second, TechKG is a high quality KG. Both of its entities and relations are acquired in two ways. One is directly taking some components of a paper as entities and building relations between entities by simply re-organizing a paper's components. The other is taking carefully selected domain

terminologies as entities and building relations between them by some carefully designed heuristic rules. These two ways guarantee TechKG have high accuracies at both entity and relation levels.

Third, TechKG can provide more kinds of information associated with the original papers besides common triplets. This makes it can be easily taken as a benchmark dataset for many diverse AI-related applications, such as KGE, distantly supervised RE, NER, KBQA, text classification(TC), machine translation(MT), among others.

Fourth, TechKG can be continually updated as long as there are new available published papers. As the needed data source is readily available in almost all languages, researchers can easily build similar KGs for other languages.

In the rest of this paper, we first introduce the building process of TechKG. Then we analyze the characteristics of TechKG in detail. We also conduct some primary experiments using TechKG as a dataset to three diverse tasks (see the *Appendix* part for detail). Both our analyses and experimental results show that lots of characteractics distinguish TechKG from existing KGs and it raises many new challenges for lots of existing applications.

## 2 RELATED WORK

At present, there are many available KGs, such as *SUMO*Niles and Pease (2001), *Cyc* Matuszek et al. (2006), *FreeBase* Bollacker et al. (2008), *WordNet* Miller (1995), *YAGO* Suchanek et al. (2007), *YAGO2* Hoffart et al. (2013), *YAGO3* Mahdisoltani et al. (2015), etc. Among them, *FreeBase*, *WordNet*, *YAGO* and some of their subsets are the most widely used KGs in many diverse applications. We will introduce them in the following.

**FreeBase** Bollacker et al. (2008) is a database system designed to be a public repository of the word's knowledge. Its design is inspired by Semantic Web and Wikipedia. It tries to merge the scalability of structured databases with the diversity of collaborative wikis into a practical and scalable database of structured general human knowledge. Totally, it contains more than 100 million asserts about topics spread over 4000 types, including *people*, *media*, *locations*, and many others.

**FB15k** Bordes et al. (2013) is a subset of Freebase. It is one of the most widely used datasets for many AI-related tasks. Totally, it contains 14,951 entities, 592,213 triplets, and 1,345 different relations. A large fraction of the triplets in this KG describe facts about *movies*, *actors*, *awards*, etc.

**WordNet**Miller (1995) is an electronic lexical database that is considered to be the most important resource available to researchers in computational linguistics, text analysis, and many related areas. Its design is inspired by psycholinguistic and computational theories of human lexical memory. In WordNet, English nouns, verbs, adjectives, and adverbs are organized into synonym sets, each represents one underlying lexicalized concept, and semantic relations link the synonym sets. Specifically, WordNet includes some kinds of semantic relations like *synonymy*, *antonymy*, *hyponymy*, *memonymy*, *troponymy*, *entailment*, etc.

**WN18** Bordes et al. (2013) is a subset of WordNet. Like FB15k, it is widely used in many AI-related tasks. Totally, it consists of 18 kinds of relations, 40,943 entities, and 151,442 triplets. Most of its triplets consist of *hyponym* and *hypernym* relations.

**YAGO**Suchanek et al. (2007) is a light-weight and extensible ontology with high coverage and quality. It contains more than 1 million entities and 5 million facts. These facts are expressed by relations and are automatically extracted from Wikipedia and unified with WordNet by a carefully designed combination of rule-based and heuristic methods. In YAGO, the relations include the *Is-A* hierarchy as well as non-taxonomic relations like *HasWonPrize*, *BornInYear*, *DiedInYear*, etc.

**YAGO2** Hoffart et al. (2013) is an extension of YAGO knowledge base, in which entities, facts, and events are anchored in both time and space. YAGO2 is built automatically from Wikipedia, GeoNames, and WordNet. It contains 447 million facts and about 9.8 million entities.

**YAGO3** Mahdisoltani et al. (2015) is another extension of the YAGO knowledge base. It combines the information from the Wikipedia in 10 languages including English, German, French, Dutch, Italian, Spanish, Romanian, Polish, Arabic, and Farsi. The choice of languages was determined by the proficiency of the authors, so as to facilitate manual evaluation. Totally, there are 8,936,324 triplet facts built upon 4,595,906 entities.

**YAGO3-10** Mahdisoltani et al. (2015) is a subset of YAGO3 and is also widely used in many applications. It consists of entities which have a minimum of 10 relations each. Totally, it has 123,182 entities and 37 kinds of relations. Most of the triplets deal with descriptive attributes of people, such as *citizenship*, *gender*,*profession*, etc.

**NOTE** that here we copy these KGs' statistics directly from their corresponding reference papers. There may be some differences for these statistics because almost all of these KGs are continually updating.

## 3 TECHKG BUILDING

Every day, there are massive Chinese technical papers published in journals of different research domains. These papers often have following characteristics.

First, there is a clear structure for each of these papers. For example, each paper often has such components like: a title, an author list, an authors' affiliation list, keyword list, abstract, etc.

Second, each technical paper can be accurately classified into a proper research domain according to the journal where this paper is published. This is because that each academic journal has its own scope for submissions. Taking "*Chinese Journal of Computers*" for example, it only focuses on the latest research in *computer science* domain. And it would never accept a paper in *chemistry* or *artist* domain. In other word, as long as we know the focused research domain of a journal, we can confidently classify all of its accepted papers into this research domain. There may be some exceptions, but the proportion would be very small.

These characteristics inspire us that we can use technical papers as data source to build a large scale Chinese KG that is technology-oriented.

### 3.1 PAPER COLLECTION

We take technical papers as data source to build TechKG. These papers were mainly collected from journals' websites. According to the submission scopes of the journals[1], we classify all of them into 38 types: each of which corresponds to a specific research domain.

For simplicity[2], we omit the body part of a paper and only collect its: title, author list, authors' affiliation list, keyword list, and abstract. If a paper's these components also have English translations, the corresponding English parts will also be collected.

Totally, over 70 million technical papers are collected.

### 3.2 ENTITY SELECTION

In TechKG, each component of a paper except abstract is taken as an entity. That is to say, title, author, keyword, and affiliation, either in Chinese or English, are all taken as entities. And the name of each research domain is also taken as an entity.

Among the obtained entities, keywords account for a very large proportion. We had expected that all of them were high quality domain terminologies, but there are lots of exceptions. For example, in some papers of the *computer science* domain, there are keywords like "*three phases*", "*two steps*", "*generation*", etc. Obviously, these keywords are not good domain terminologies for they have less to do with *computer science*. So we use a *tf\*idf* method to filter such kind of keywords. Specifically, we concatenate all the keyword lists in each research domain to form a long domain-related document. Then the *tf* value of a keyword in each domain can be computed. Different domain-related documents are viewed as the counterpart of each other, thus the *idf* value of a keyword can be computed accordingly. The higher a keyword's *tf\*idf* value, the more possible this keyword is a good domain terminology.

---

[1]Here we also use *CNKI's*(http://www.cnki.net/) categories for reference, but we merge some similar research domains in *CNKI* and reduce the amount of research domains.

[2]It is difficult to effectively analyse the body part of a paper.

## 3.3 RELATION EXTRACTION

The role of relations is to describe the attributes of entities and the relationships between entities. In TechKG, keywords and authors are two kinds of entities with the largest amounts. So we define relations mainly centered upon them. Besides, papers themselves contain much valuable factual knowledge. So we define some paper-centered relations to describe the attributes of papers themselves. Totally, 16 relations are defined in TechKG.

**Author-centered relations** include: *author_of*, *first_author*, *second_author*, *co_author*, *research_interest*, and *affiliation*.

**Keyword-centered relations** include: *belonged_domain, co-occurrence_with*, and *hierarchical*.

**Paper-centered relations** include: *published_year*, *contained_chn_keywords*, *contained_eng_keywords, other_author*, *all_authors*, and *published_journal*.

Many technical papers also have English translations for their titles, affiliations, keywords, etc., so we define a new relation named *translation_of*.

Among these defined relations, *co_author*, *translation_of* and *co_occurrence* are a kind of symmetrical relations and their directions are not taken into consideration.

In TechKG, a set of heuristic rules are employed to assign relations between entities.

**First**, for each keyword $k_i$, if it occurs in the keyword list of a paper and this paper is published in a journal that has been categorized into a research domain $d_j$, we assign a "*belonged_domain*" relation from $k_i$ to $d_j$.

**Second**, we notice that most Chinese researchers have such a writing habit: in a keyword list, they are more likely to place high-level (more abstract) keywords ahead of low-level (more concrete) keywords. For example, we are more likely to see such a keyword list "*natural language processing*, *neural machine translation*, *attention mechanism*" than a list of "*neural machine translation*, *natural language processing*, *attention mechanism*". In other word, there is a natural hierarchical structure inherent in the keyword list of a paper. Based on this finding, we design a simple heuristic rule to mine the "*hierarchical*" relations between keywords. Specially, for each keyword pair "$k_1$-$k_2$", we count the number of papers that:1) both $k_1$ and $k_2$ occur in their keyword lists; and 2) $k_1$ occurs ahead of $k_2$ in their keyword lists. If this number is larger than a predefined threshold, we assign a "*hierarchical*" relation from $k_1$ to $k_2$.

**Third**, for any two keywords $k_1$ and $k_2$, if they co-occurrence in a paper's keyword list, we assign a "*co-occurrence_with*" relation between them. We also record the number of papers that $k_1$ and $k_2$ co-occurrence. This number is taken as a confidence factor to evaluate the reliability of this "*co-occurrence_with*" relation between $k_1$ and $k_2$.

**Fourth**, if there is an English part for a paper's title, authors, or affiliations, we assign a "*translation_of*" relation to the corresponding bilingual entity pairs. It should be noted that there may be multi authors or affiliations in a paper. In such case, we will assign the "*translation_of*" relation to two entities according to their occurrence orders in the corresponding bilingual parts. Taking authors for example, a "*translation_of*" relation will be assigned to the first Chinese author and the first English author; to the second Chinese author and the second English author, and so on.

**Fifth**, if there are bilingual keyword lists in a paper, we first construct candidate entity pairs according to two entities' occurrence orders in their corresponding bilingual parts. For example, a possible entity pair "$c_i$-$e_j$" may be: the first Chinese keyword with the first English keyword, the second Chinese keyword with the second English keyword, and so on. Then for each candidate entity pair "$c_i$-$e_j$", we count the number of papers that can generate it. If this number is larger than a predefined threshold, we assign a "*translation_of*" relation between $c_i$ and $e_j$. Here we take different extraction rules for the "*translation_of*" relation due to the reason that the numbers of keywords in Chinese keyword list and English keyword list are not always the same.

For other kinds of relations, we can precisely obtain them by re-organizing the meta-items of the original papers.

### 3.4 BUILDING RESULTS

Finally, more than 52 million entities and more than 260 million triplets are extracted. Some basic statistics of TechKG are shown in Table 1. It should be noted that entities maybe overlapped in different kinds of relations. So in Table 1, the total entity number is less than the sum number of entities in all relations. Statistics of the "*hierarchical*" relation are not listed in Table 1 because they depend on the predefined threshold. One can see the detailed statistics about each relation (including the "*hierarchical*" relation) in the *Appendix* part.

Currently, we don't mine complex relations for TechKG because we put more emphasis on its accuracy. We can see that most of the triplets in TechKG are generated by re-organizing the meta-items of the original papers. Thus, TechKG is a large-scale KG that has high accuracy for both its entities and relations. Besides, the *tf*\**idf* method in entity selection and the *threshold* method in relation extraction can further guarantee TechKG's accuracy.

## 4 TECHKG CHARACTERISTICS

We find there are three main characteristics that distinguish TechKG from most of existing KGs: the duplicate name issue, the imbalance issue, and the good application adaptability.

### 4.1 DUPLICATE NAME ISSUE

In TechKG, *duplicate name* is a very common phenomenon: not only some authors share some common names, but also some research domains also share some common terminologies. For example, *Ning Li*, a researcher name that appears in 34 research domains. That is to say, there is a researcher named *Ning Li* in almost all of research domains. We call this kind of duplicate names as cross-domain duplicate name issue.

Table 1: Some basic statistics of TechKG.

| Relation Types | EntityNum | TripletNum |
|---|---|---|
| author_of | 7,397,886 | 15,403,800 |
| first_author | 6,662,907 | 5,588,103 |
| second_author | 4,645,848 | 3,858,252 |
| co_author | 1,764,270 | 17,188,520 |
| research_interest | 5,179,265 | 50,748,054 |
| affiliation | 1,900,697 | 10,775,303 |
| belonged_domain | 4,636,780 | 9,494,972 |
| co-occurrence_with | 4,628,635 | 72,101,139 |
| published_year | 15,628,610 | 16,367,393 |
| contained_chn_keyword | 22,345,343 | 11,697,138 |
| contained_eng_keyword | 8,338,872 | 4,304,324 |
| other_author | 3,651,823 | 5,958,552 |
| all_authors | 9,403,013 | 5,591,189 |
| published_journal | 15,636,388 | 16,413,111 |
| translation_of | 19,681,543 | 15,263,050 |
| Total | 52,716,813 | 260,763,332 |

In Table 2, we report some statistics for the duplicate name issue of authors. Here *dpa* means the average appeared domains per duplicated author. It should be noted that here we ignore *nature science* and *social science* during making statistics. Both of them are comprehensive research domains and it is very likely to generate some false duplicated author names if we take them into consideration. For example, a researcher may publish his paper in both *computer science* domain and in *nature science* domain.

However, when we make statistics according to research domain, we find both the *duplicate ratio* and the *dpa* increase greatly (see the *Appendix* part for detail). These statistics mean that there are a very small number of duplicated names that appear in many different research domains. Or most of

Table 2: Cross-domain duplicate name statistics for authors.

| total # | duplicate # | duplicate ratio | *dpa* |
|---------|-------------|-----------------|-------|
| 1,826,217 | 654,884 | 35.86% | 3.97 |

the duplicated names appear in a relative smaller research domain set. To verify this hypothesis, we further make statistics of the authors' domain distribution. The results are shown in Table 3.

From Table 3 we can see that more than 80% of the duplicated authors appear in less than 5 research domains. There are 7,769 duplicated authors that appear more than 20 research domain. More extremely, we find that there are 167 author names (such as: *Lin Zhang, Hua Liu, Yang Wang, Lei Zhang*, etc) that appear in all research domains. These results well verify our hypothesis and can explain the reason of why both the *duplicate ratio* and the *dpa* increase greatly in separated research domains.

Table 3: Distribution of domain numbers for cross-domain duplicated authors.

|  | Domain distribution | Duplicate # | Duplicate ratio |
|--|---------------------|-------------|-----------------|
|  | [30, 36] | 1,390 | 0.21% |
|  | [20, 30) | 6,369 | 0.97% |
|  | [10, 20) | 25,976 | 3.97% |
|  | [5, 10) | 73,082 | 11.16% |
|  | [1, 5) | 548,067 | 83.69% |
| Total | [1,36] | 654,884 | 100% |

There is not only cross-domain duplicate name issue, but also another kind of in-domain duplicate name issue. For example, in the research domain of *computer science*, there will be many duplicated authors that are in different affiliations. Among the in-domain authors that have several affiliations, some of them are real duplicated names, some are not. Although the entity linkage (or entity resolution) issue has been well studied, it is so hard that it is almost impossible for us to accurately pick up the true in-domain duplicate names due to the following two reasons.

First, there may be some researchers that work in different affiliations in different times.

Second, some authors provide their affiliations with different granularity in different papers. For example, an author may write his affiliation as "*Northeastern University*", "*School of Computer Science and Engineering, Northeastern University*", or even "*Natural Language Processing Lab, Northeastern University*", etc.

Here we try to provide a rough extreme analysis of this in-domain duplicate name issue.

For the worse case, as long as an author name appears in different affiliations, we take it as a duplicated name.

For the better case, for an author name appears in different affiliations, we take it as a duplicated name only when there is no inclusion relationship between these affiliations.

In Table 4, we report the in-domain duplicate name statistics for *metallurgical industry*, *computer science*, *electric industry*, and *traffic transportation*, 4 randomly selected research domains. Here *apa* means the average appeared affiliations per duplicated author.

From Table 4 we can see that the in-domain duplicate name issue is very serious. In both cases, more than 50% authors will have multi affiliations. An interesting finding is that the duplicate ratio of the better case is lower than the worse case, but the *apa* value of the better case in larger than the worse case. The former results are in line with our expectations, but the latter are not. This means the filtered authors have fewer affiliations. In other words, authors are more likely to use fewer numbers of affiliations with different granularity.

Here we are keenly aware that both of our two rules for the extreme analysis can be criticized for there are lots of exceptions for each of them. We leave the accurate statistics of this issue as an open issue and to solve it in the future.

Table 4: Extreme analysis of in-domain duplicate author name.

| domain | total # | Worse/Better | | |
| | | duplicate # | ratio(%) | *apa* |
|---|---|---|---|---|
| metal | 92637 | 56160/54065 | 60.62/58.36 | 4.38/4.46 |
| computer | 95163 | 54333/52719 | 57.09/55.40 | 4.25/4.31 |
| electric | 92230 | 51040/49055 | 55.34/53.19 | 4.19/4.26 |
| traffic | 53354 | 29661/28843 | 55.59/54.06 | 3.44/3.48 |

For keywords, they will only involve the cross-domain duplicate name issue. The number of duplicated keywords will be affected by the predefined *tf\*idf* threshold. We make statistics under different *tf\*idf* settings and the results are shown in Table 5. Here *dpk* means the average appeared domains per duplicated keyword.

Table 5: Cross-domain duplicate name statistics for keywords.

| *tf\*idf* setting | total # | duplicate # | duplicate ratio | *dpk* |
|---|---|---|---|---|
| Top-100% | 8,003,868 | 1,817,771 | 22.71% | 4 |
| Top-90% | 8,003,293 | 1,808,536 | 22.60% | 3.32 |
| Top-80% | 7,978,805 | 1,676,176 | 21.01% | 2.84 |
| Top-70% | 7,883,539 | 1,363,444 | 17.29% | 2.68 |
| Top-60% | 7,494,000 | 820,490 | 10.95% | 2.87 |
| Top-50% | 7,234,018 | 691,549 | 9.56% | 2.91 |
| Top-40% | 6,357,230 | 662,029 | 10.41% | 2.88 |
| Top-30% | 5,224,408 | 594,194 | 11.37% | 2.84 |
| Top-20% | 2,870,181 | 412,436 | 14.37% | 2.8 |
| Top-10% | 1,153,819 | 202,060 | 17.51% | 2.62 |

From Table 5 we can see that the number of duplicated items decreases as the *tf\*idf* increases. However, both the *duplicate ratio* and the *dpk* don't change too much. Even only remaining the keywords whose *tf\*idf* values rank in top-10%, there are still 17.51% of the keywords appearing in average 2.62 research domains.

We also make statistics of the keywords' domain distribution as we made for authors. But different from authors' statistics, keywords' statistics will depend on the *tf\*idf* settings. Here report the domain distribution of keywords under top-10% *tf\*idf* setting in Table 6 for *metallurgical industry*, *computer science*, *electric industry*, and *traffic transportation*, 4 randomly selected research domains.

Table 6: Distribution of domain numbers for keywords.

| domain | Duplicate num /duplicate ratio(%) | | | | |
| | [1, 5) | [5, 10) | [10, 20) | [20, 30) | [30, 36] |
|---|---|---|---|---|---|
| metal | 14518/86.77 | 1761/10.53 | 443/2.65 | 9/0.05 | 0/0 |
| computer | 21463/92.83 | 1372/5.93 | 278/1.20 | 8/0.03 | 0/0 |
| electric | 22102/92.43 | 1453/6.08 | 349/1.46 | 8/0.03 | 0/0 |
| traffic | 16376/86.82 | 2019/10.70 | 459/2.43 | 9/0.05 | 0/0 |

From Table 6 we can see that compared with the domain distribution of authors, the keywords' domain distribution is more balanced: most keywords appear less than 5 research domain. Generally, the more strict a *tf\*idf* required, the less possible for a keyword appears in different research domains. For example, under the *tf\*idf* setting of above top-80%, more than 90% keywords appear less than 5 domains (see the *Appendix* part for detail). But there are some keywords that appear in many research domains. For example, under the *tf\*idf* setting of top-10%, there are still many keywords appearing in more than 20 research domains, such as *genetic algorithm, numerical simulation, finite element method, preventive measure*, etc.

## 4.2 IMBALANCE ISSUE

TransH[3] classifies a KG's relations into 4 types: *1-1*, *1-n*, *m-1*, and *m-n*. Following this definition, we make statistics for the proportion related triplets in different type of relations under different *tf\*idf* settings. For the latter 3 types of relations, we also make statistics of the average value of *n* and the average value of *m/n*. The statistics results are shown in Table 7.

From Table 7 we can see that the triplet distribution for these 4 relation types is extremely imbalanced. In each *tf\*idf* setting, the *m-n* relations account for more than 60% triplets. On the other hand, the *1-n* relations account for so small a proportion that they can even be ignored. The proportion is almost unchanged for different type of relation under different *tf\*idf* settings.

Table 7: Cross-domain duplicate name statistics for keywords.

| *tf\*idf* setting | 1-1(%) | 1-n(%)/$\tilde{n}$ | m-1(%)/$\widetilde{m}$ | m-n(%)/$\bar{u}$ |
|---|---|---|---|---|
| Top-100% | 13.61 | 0.27/1.51 | 20.64/14.99 | 65.47/1.21 |
| Top-90% | 14.36 | 0.01/1.87 | 19.93/13.69 | 65.70/1.23 |
| Top-80% | 14.6 | 0.01/1.87 | 19.83/13.13 | 65.56/1.25 |
| Top-70% | 14.77 | 0.01/1.87 | 19.78/12.71 | 65.44/1.27 |
| Top-60% | 14.91 | 0.01/1.87 | 19.80/12.39 | 65.28/1.29 |
| Top-50% | 14.99 | 0.01/1.87 | 19.84/12.27 | 65.17/1.3 |
| Top-40% | 14.99 | 0.01/1.87 | 19.87/12 | 65.13/1.32 |
| Top-30% | 15.15 | 0.01/1.87 | 19.65/11.48 | 65.19/1.36 |
| Top-20% | 15.43 | 0.01/1.87 | 19.88/10.73 | 64.68/1.46 |
| Top-10% | 16.32 | 0.01/1.87 | 20.74/10.14 | 62.93/1.58 |

When we look into the average number of tails per head ($\tilde{n}$ in Table 7) or the average number of head per tail ($\widetilde{m}$ or $\bar{u}$ in Table 7), the difference among different types of relations is far huge: in *m-1* relations, the average value of *m* is more than 10, but both the average value of *n* in *1-n* relations and the average value of *m/n* in *m-n* relations is relative smaller. These average values are also almost unchanged for different type of relation under different *tf\*idf* settings.

The imbalance issue absolutely will have some impacts on the performance of some applications like *knowledge graph embedding,* etc. In the experiment section, we will further discuss these impacts.

## 4.3 APPLICATION ADAPTABILITY

Table 8: Comparisons of different KGs.

| | WordNet | FreeBase | YAGO | YAGO2 | YAGO3 | TechKG |
|---|---|---|---|---|---|---|
| Language | English | English | English | English | MultiLingual (noChn) | Chinese |
| Purpose | General | General | General | General | General | Technology |
| DataSource | words | Wikipedia | Wikipedia | Wikipedia | Wikipedia | TechPapers |
| EntityNum | 117,597 | $\approx$ 80M | $\geq$1M | 2,648,387 | 4,595,906 | 162,521,245 |
| TripletNUM | 207,016 | $\approx$1.2KM | $\geq$5M | 124,333,521 | 8,936,324 | 258,554,528 |
| SupportedTasks | KGE | KGE,RE, KBQA | KGE,RE, KBQA | KGE,RE, KBQA | KGE,RE, KBQA | KGE,RE,TC, KBQA,NER,MT |

In Table 8, we make a comparison between TechKG and some other KGs. TechKG can support more kinds of applications because it can provide other kinds of information as listed below.

**Bilingual Translation Pairs.** These pairs include bilingual titles, bilingual keywords, and bilingual abstracts. These bilingual translation pairs usually have very high quality for they are written by experts in the corresponding research domain. They can provide high quality training data for a machine translation task.

---

[3]For each relation r, we compute averaged number of tails per head ($tph_r$), averaged number of heads per tail ($hpt_r$). If $tph_r <$1.5 and $hpt_r <$ 1.5, *r* is treated as 1-1. If $tph_r \geq$ 1.5 and $hpt_r \geq$ 1.5, *r* is treated as m-to-n. If $hpt_r <$ 1.5 and $tph_r \geq$1.5, *r* is treated as 1-n. If $hpt_r \geq$ 1.5 and $tph_r <$ 1.5, *r* is treated as m-1.

**Domain Categorizations.** For each triplet or a paper's abstract, it has a pre-organized domain category. With these categories, one can use the collected abstracts as a dataset for a text classification task. TechKG can also be easily divided into different subsets according to the research domains. Thus different kinds of transfer learning methods can also be evaluated by these subsets.

**Domain Terminologies.** In TechKG, a large proportion of its entities are domain terminologies. Thus a new kind of named entity recognition (NER) task that aims to recognize domain terminologies from text can be evaluated with these domain terminologies. In such a NER task, the required training data can be obtained by the distantly supervision based method as used in Yang et al. (2018).

**Cross-language KG based Applications**. The *translation_of* relation in TechKG provides a possibility to link KGs of different languages together. This would benefit lots of cross-language KG based applications like *cross-language Q&A*, *cross-language information retrieval*, etc.

Finally, we also construct following kinds of knowledge bases (KBs) together with TechKG. Each of these KBs can be viewed as a by-product of TechKG and can be taken as a dataset for some applications.

### 4.3.1 TECHTERM/TECHBITERM

They are two KBs that consist of Chinese domain terminologies and *Chinese-Englis*h terminology pairs respectively.

For the former one, we randomly select 10,000 terminologies with the highest *tf\*idf* values for each research domain.

For the latter one, we randomly select 10,000 bilingual translation pairs with the highest *co-occurrence* values from each research domain.

### 4.3.2 TECHABS

It is a KB that stores papers' abstracts. It can be used as a dataset for a text classification task. Here we randomly select 100,000 abstracts from each research domain to construct TechAbs.

### 4.3.3 TECHQA

TechQA is a dataset designed for the KBQA task. It is constructed based on TechKG-10. All the questions in it are generated by some predefined patterns. Currently, it focuses on 4 kinds of questions which are "*who*", "*what*", "*when*", and "*where*".

We list the used question patterns and detailed statistics of TechQA in the *Appendix* part.

### 4.3.4 TECHRE/TECHNER

They are two KBs designed mainly for relation extraction (*RE* for short) and domain terminology recognition tasks respectively. The former one is constructed based on TechKG-10 (a subset of TechKG, see the *Appendix* part for detail) and the latter one is constructed based on TechTerm.

RE aims to extract semantic relations between entity pairs from plain text. It is a typical classification task when the entity pairs and their possible relations are specified. To achieve high quality results, a large-scale training corpus is required. However, it is very time-consuming and expensive to construct such a corpus manually. To address this issue, distant supervision is proposed Mintz et al. (2009) to automatically generate training data via aligning KBs and texts. Its basic idea is that: if two entities have a relational fact in KBs, all the sentences that contain these two entities will express the corresponding relation. Accordingly, these sentences can be taken as positive samples for training this relation. Distant supervision provides an effective strategy to automatically construct large-scale training data for a RE task. Here TechKG-10 is used as the required supervision KB and papers' abstracts are used as the required text corpus. It should be noted that except keywords, it is little possible for other types of components occur in the abstract part of a paper. So in TechRE, only *hierarchical* relation is considered. We set the co-occurrence threshold to 10 for the extraction of *hierarchical* relation. Specially, for each two terminologies in a *co-occurrence_with* type of triplet, we first pick up all the sentences that both of these two terminologies appear. We call each

terminology pair along with its mentioned sentence set as a *bag*. If the co-occurrence value of a bag's terminology pair is greater than 10, we assign a *hierarchical* relation to this bag and take it as a positive *hierarchical* training sample. Otherwise a *NA* relation is assigned to this bag and it is taken as a negative *hierar*chical training sample. By this way, we extract more than 14M training bags. Averagely, there are about 42 sentences per bag and 5 entities per sentences. The whole dataset is too large. So we randomly select at most 200,000 training bags for each research domain to construct TechRE. Besides, we limit the *average sentences per bag* in TechRE to 6 at most.

For TechNER, we also adopt a distant supervision method as TechRE, but TechTerm is used as the required KB. Papers' abstracts are taken as the required text corpus. If a sentence contains a terminology, this sentence will be regarded as a positive training sample for recognizing this terminology. This distant supervision method for constructing NER training corpus is also used in Yang et al. (2018). By this way, we extract more than 210M training sentences. Averagely, there are about 3 entities per sentences. We randomly select 30,000 training samples in each research domain to construct TechNER. There are averagely 3 entities per training sample. In TechNER, if there are nested domain terminologies in a sentence, we will only remain the longest one.

The detailed and complete statistics of TechRE and TechNER can be found in the *Appendix* part.

## 5 CONCLUSIONS AND FUTURE WORK

In this paper, we introduce TechKG, a large-scale Chinese KG. To the best of our knowledge, TechKG is: both the first KG that is constructed from massive technical papers and the first KG that is technology-oriented and can be divided into different subsets according to the research domains.

We analyze the main characteristics of TechKG. Lots of statistics show that there are many new characteristics hierent in TechKG distinguish it from other existing KGs. We also conduct some primary experiments (see the *Appendix* part for detail) to evaluate the adaptability of TechKG. Experiments show that: 1)TechKG provides a new choice for lots of applications; and 2)a lot of new characteristics hierent in TechKG raise new challenges for many existing methods.

Currently, we generate TechKG mainly by re-arranging the meta-items in the collected technical papers. In the future, we will explore to extract more types of relations such as <*method*, *solve*, *problem*>, etc.

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

# A  EXPERIMENTS AND ANALYSES

To evaluate the adaptability of TechKG, we apply it to three completely different applications: knowledge graph embedding, distantly supervised RE, and distantly supervised NER. For each of them, we select one of its state-of-the-art methods as baseline and use these baselines to conduct experiments on TechKG10 [4], TechRE, and TechNER respectively. For each application, we report its experimental results on 4 randomly selected research domains that are *metallurgical industry*, *computer science*, *electric industry*, and *traffic transportation*.

**NOTE** that the aim of our experiments is not to tell whether a model is good or not. On the contrary, we just want to find whether our TechKG could show some new characteristics hidden in KGs that haven't been noticed up to now.

Table 9: Some basic statistics of TechKG10.

|  | EntityNum | TripletNum |
|---|---|---|
| author_of | 4701043 | 12107175 |
| first_author | 3826827 | 3210759 |
| second_author | 3808524 | 3240132 |
| co_author | 1291222 | 15324905 |
| research_interest | 1955144 | 29705592 |
| affiliation | 1055761 | 6428984 |
| belonged_domain | 780466 | 1160626 |
| co-occurrence_with | 777960 | 30553582 |
| published_year | 1713143 | 1793187 |
| contained_chn_keyword | 9377 | 6800 |
| contained_eng_keyword | 426 | 308 |
| other_author | 3342604 | 5657370 |
| all_authors | 36462 | 22328 |
| published_journal | 3332185 | 3547271 |
| translation_of | 146204 | 281555 |
| Total | 5750579 | 113040574 |

## A.1  KNOWLEDGE GRAPH EMBEDDING

KGE projects each item (entities and relationships) of a KG into continuous low-dimensional space(s). It aims to solve two major challenges that prohibit the availability of KGs. One is the incompleteness issue: although most of existing KGs contain large amount of triplets, they are far from complete. The other is the computation issue: most of existing KGs are stored in symbolic and logical formations while applications often involve numerical computing in continuous spaces. *Link prediction*, which is to predict the missing $h$ or $t$ for a correct triplet $<h, r, t>$, i.e., predict $t$ given $<h, r>$ or predict $h$ given $<r, t>$, is often used to evaluate the performance of a KGE method. Rather than requiring one best answer, *link prediction* emphasizes more on ranking a set of candidate entities from the KG. *Hits@10*, the proportion of correct entities ranked in top 10, is often used as the evaluation metric for *link prediction*.

In this paper, we select *ConvE*Dettmers (2018), one of the latest state-of-the-art KGE methods as baseline. We want to find: 1) whether the duplicate name issue would affect the performance; 2) whether the imbalance issue would affect the performance; and 3) whether TechKG also suffers from the test set leakage issue that is severe in WN18 and FB15k Dettmers (2018).

For these aims, we report the experimental results on following kinds of datasets.

1) The original TechKG10 dataset.

---

[4]The whole TechKG is a huge and growing KB, which is almost impossible for many applications to directly experiment on. Thus, we create TechKG10: a subset of it. We construct TechKG10 by selecting the subset of entities that meet following two requirements. First, if the entity is a keyword, its *tf\*idf* value must rank in top-10%. Second, each selected entity has at least 10 mentions. Some basic statistics of TechKG10 are shown in Table 9.

2) A revised TechKG10 dataset that no cross domain author duplicate names included.

3) A revised TechKG10 dataset that no in domain author duplicate names included.

4) A revised TechKG10 dataset that no cross domain keyword duplicate names included.

5) A revised TechKG10 dataset that no any duplicate names included.

6) A revised TechKG10 dataset that no relations of *co_ocurrence* or *research_interest*.

7) A revised TechKG10 dataset[5] that no relations of *co_ocurrence* or *research_interest* and no any duplicate names included.

Even in TechKG10, the triplet number of its each domain is far larger than FB15k. So in experiments, we randomly select 30,000 triplets for each of above 7 datasets. The number of triplets used here is less than that of FB15k: there are about 48,000 triplets in FB15k. However, for each of these 7 datasets, its involved parameter size is far larger than FB15k. FB15k has 14951 entities and 1345 relations, but the average number of contained entities in these 7 datasets is about 140,000. This means it needs more time for training and testing on these 7 datasets.

Table 10: Hits@10 results of KGE.

| | | ConvE | InverseModel |
|---|---|---|---|
| WN18 | | 0.955 | 0.969 |
| FB15k | | 0.873 | 0.737 |
| WN18RR | | 0.48 | 0.36 |
| FB15k-237 | | 0.491 | 0.012 |
| TechKG10-original | Metal | 0.051 | 0.0017 |
| | Computer | 0.057 | 0.0013 |
| | Electric | 0.053 | 0.0011 |
| | Traffic | 0.067 | 0.0013 |
| TechKG10-NoCrossDomainAuthor Duplicate | Metal | 0.095 | 0.0005 |
| | Computer | 0.109 | 0.0002 |
| | Electric | 0.096 | 0.0004 |
| | Traffic | 0.087 | 0.0003 |
| TechKG10-NoInDomainAuthor Duplicate | Metal | 0.093 | 0.0062 |
| | Computer | 0.094 | 0.0003 |
| | Electric | 0.092 | 0.0008 |
| | Traffic | 0.085 | 0.0007 |
| TechKG10-NoCrossDomainKeywordDuplicate | Metal | 0.12 | 0.0048 |
| | Computer | 0.13 | 0.0036 |
| | Electric | 0.127 | 0.0038 |
| | Traffic | 0.185 | 0.004 |
| TechKG10-NoAnyDuplicateName | Metal | 0.389 | 0.0023 |
| | Computer | 0.308 | 0.0006 |
| | Electric | 0.315 | 0.0012 |
| | Traffic | 0.306 | 0.0013 |
| TechKG10-NoTwoSelectedRelations | Metal | 0.156 | 0.0052 |
| | Computer | 0.155 | 0.0033 |
| | Electric | 0.164 | 0.0049 |
| | Traffic | 0.167 | 0.0062 |
| TechKG10-NoTwoRelationsNoAnyD plicateNames | Metal | 0.488 | 0.0023 |
| | Computer | 0.348 | 0.0009 |
| | Electric | 0.407 | 0.0017 |
| | Traffic | 0.413 | 0.0024 |

---

[5]Because the duplicated names and the two selected relations account for a large proportion in TechKG10, there are only about 235,000 and 224,000 triplets for the *traffic transportation* and *metallurgical industry* domains respectively. For the rest two research domains, there are about 290,000 triplets in this dataset.

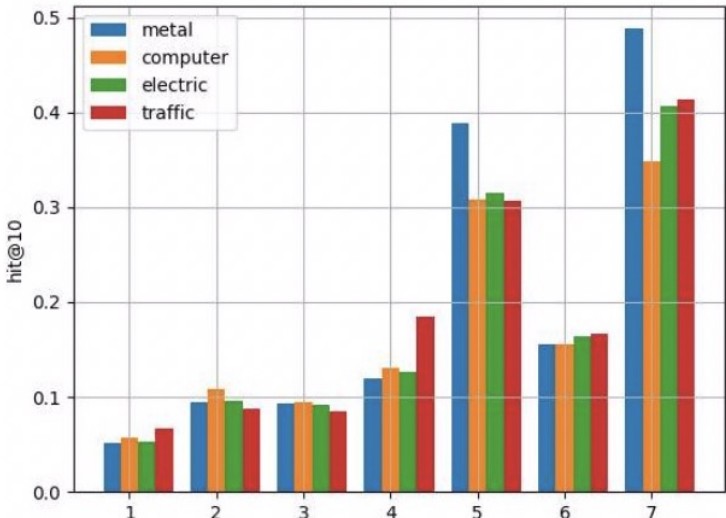

Figure 1: Hits@10 results of KGE. The ID numbers in the bottom line correspond to the datasets.

The datasets 2-5 are used to evaluate the effects of duplicate names for the KGE task. The dataset 6 is used to evaluate the effects of imbalance issue. In TechKG10, both the *co_ocurrence* relation and the *research_interest* relation account for a large proportion in TechKG10, so we remove them to generate a more balanced dataset.

In experiments, the proportion of training, development and test sets is 8:1:1. The experimental results are shown in Table 10 and Figure 1, from which we can find following observations. During training, we iterate 300 rounds for each experiment.

First, compared with FB15k or WN18, the performance on TechKG10 is far lower. We think this is caused mainly by the data sparsity issue. Compared with other datasets, there are more parameters needed to be trained, but the number of training triplets is less.

Second, there is almost no the test set leakage issue at all in TechKG, which means it is impossible for a simple rule-based inverse model to achieve state-of-the-art results on TechKG.

Third, both the cross-domain and in-domain author duplicate name issues have a heavily negative effect on the performance. When there are no any duplicated names in the data set, each domain' performance increases greatly. Taking *metallurgical industry* for example, the Hits@10 value on the original dataset is only 0.051, but this value increases to 0.389 when all the duplicated items are removed from the dataset. What's more, the cross-domain keyword duplicate name issue has more negative effect on the performance, which is not in line with our original expectation. The reason of this result should be further studied.

Fourth, the imbalance issue has a heavy negative effect on the performance. When we remove two types of relations from the dataset, the performance obtains a huge improvement. This indicates that the more balanced a dataset's relation types' distributions, the better performance could be obtained.

Fifth, there are obvious performance gaps between different kinds of research domains. For example, when there is no any duplicated names, the Hits@10 value of the *metallurgical industry* domain is 0.389, which is much higher than that of the *computer science* domain, which is only 0.308.

Sixth, when there are no any duplicated names and the two special relations are removed, the best results are obtained, which is similar with that of on the datasets of WN18RR and FB15k-237.

Based on these observations we can roughly draw following conclusions. First, TechKG10 is a more challenging dataset than traditional WN18 and FB15k. Second, both the duplicate name issue and the imbalance issue are two inherent characteristics in TechKG.

## A.2 DISTANT SUPERVISION RELATION EXTRACTION

In this task, we select the "*PCNN+ATT*" Lin et al. (2016) model as the baseline method. Following two main factors are considered for this selection. First, it is a state-of-the-art RE method and many current RE methods are based on it. Second, there are available source codes for it, which is very important for building a RE system quickly.

In almost of all of current RE research, Freebase is used as the distant supervision KB and the three-year NYT corpus from 2005 to 2007 is used as the text corpus. In NYT dataset, there are about 281,270 training bags and 96,678 test bags. The size of TechRE is a little smaller than that of NYT. However, if taking the factor of sentences per bag into consideration, we can find that the parameter sizes of our datasets are far larger than NYT.

The average number of sentences per training bag in NYT corpus is about two, while this number is more 60 in TechRE. To evaluate whether this factor will affect the performance, we report the experimental results under different average number of sentences per training bag. There is also a huge difference between the proportion of *NA* relations in TechRE and NYT. To evaluate whether this factor will affect the performance, we also report the experimental results under different proportions of NA relations. During training, we iterate 100 rounds for each experiment. Finally, the *p@m* results are shown in Table 11, Figure 2, and Figure 3.

Table 11: Experimental results of RE.

|  |  | *p@300* | *p@300* | *p@300* | *mean* |
|---|---|---|---|---|---|
| NYT |  | 0.82 | 0.75 | 0.72 | 0.76 |
| TechRE(2 sentence per bag) | Metal | 0.17 | 0.195 | 0.21 | 0.192 |
|  | Computer | 0.25 | 0.26 | 0.23 | 0.247 |
|  | Electric | 0.1 | 0.16 | 0.143 | 0.134 |
|  | Traffic | 0.15 | 0.195 | 0.223 | 0.189 |
| TechRE(4 sentence per bag) | Metal | 0.07 | 0.15 | 0.15 | 0.123 |
|  | Computer | 0.16 | 0.185 | 0.173 | 0.173 |
|  | Electric | 0.13 | 0.115 | 0.12 | 0.122 |
|  | Traffic | 0.13 | 0.255 | 0.25 | 0.217 |
| TechRE(6 sentence per bag) | Metal | 0.23 | 0.19 | 0.19 | 0.203 |
|  | Computer | 0.13 | 0.205 | 0.19 | 0.175 |
|  | Electric | 0.07 | 0.095 | 0.117 | 0.094 |
|  | Traffic | 0.05 | 0.145 | 0.227 | 0.141 |
| TechRE(2 sent per bag, 90% NA rel) | Metal | 0.12 | 0.12 | 0.15 | 0.13 |
|  | Computer | 0.26 | 0.305 | 0.333 | 0.299 |
|  | Electric | 0.15 | 0.14 | 0.147 | 0.146 |
|  | Traffic | 0.08 | 0.15 | 0.167 | 0.132 |
| TechRE(2 sent per bag, 80% NA rel) | Metal | 0.07 | 0.085 | 0.073 | 0.076 |
|  | Computer | 0.23 | 0.19 | 0.183 | 0.201 |
|  | Electric | 0.12 | 0.095 | 0.087 | 0.101 |
|  | Traffic | 0.07 | 0.1 | 0.103 | 0.091 |
| TechRE(2 sent per bag, 60% NA rel) | Metal | 0.09 | 0.05 | 0.053 | 0.064 |
|  | Computer | 0.17 | 0.18 | 0.16 | 0.17 |
|  | Electric | 0.08 | 0.085 | 0.077 | 0.081 |
|  | Traffic | 0.05 | 0.055 | 0.07 | 0.058 |

From these results we can find following observations.

First, both the *average sentence per bag* and the *NA relation proportion* have effects on the performance of RE task. For some domains, the higher the *average sentence per bag*, the poorer performance would be obtained. But for other domains, there is an opposite conclusion. For the factor of the *NA relation proportion*, we find that when we reduce the proportion of *NA* relations,

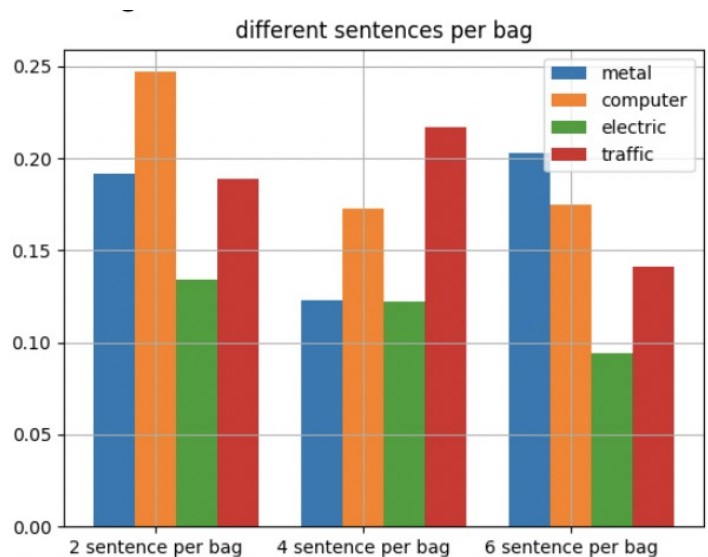

Figure 2: Experimental results of RE (part 1).

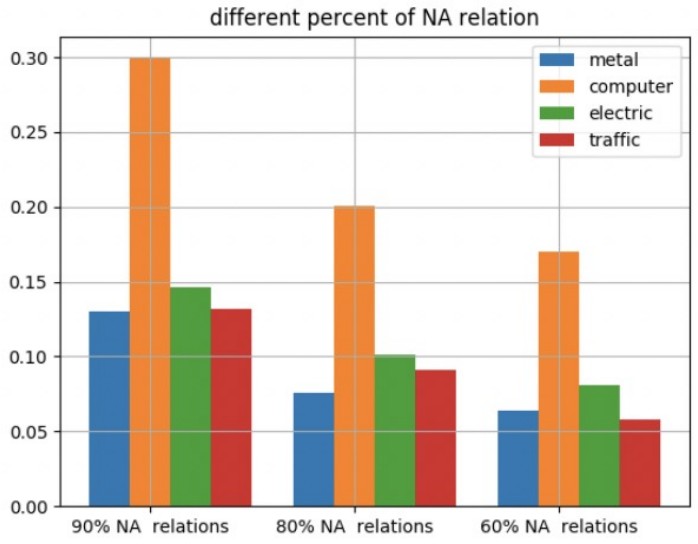

Figure 3: Experimental results of RE (part 2).

the performance decrease accordingly. Here not all of the experimental results are in line with our expectation. The reasons should be further studied.

Second, there is a big performance gap between NYT dataset and TechRE in both the *precision/recall* curves and the *p@m*: the performance on TechRE is far lower than that of on NYT. We find that in TechRE, there are almost no any cues in the training sentences that can imply a kind of relations. In NYT, however, there are words like "*caused*", "*because*", "*contain*", "*have*", etc. Obviously, these words are very helpful for the relation decision. In fact, we think TechRE is a more challenging dataset for the RE task: there are not always useful cues available.

Third, we find that there are also huge performance gaps between different kinds of domains. For example, under the setting of *2 sentences per bag*, the performance of *computer science* domain is far higher than the results of other domains.

## A.3    NAMED ENTITY RECOGNITION

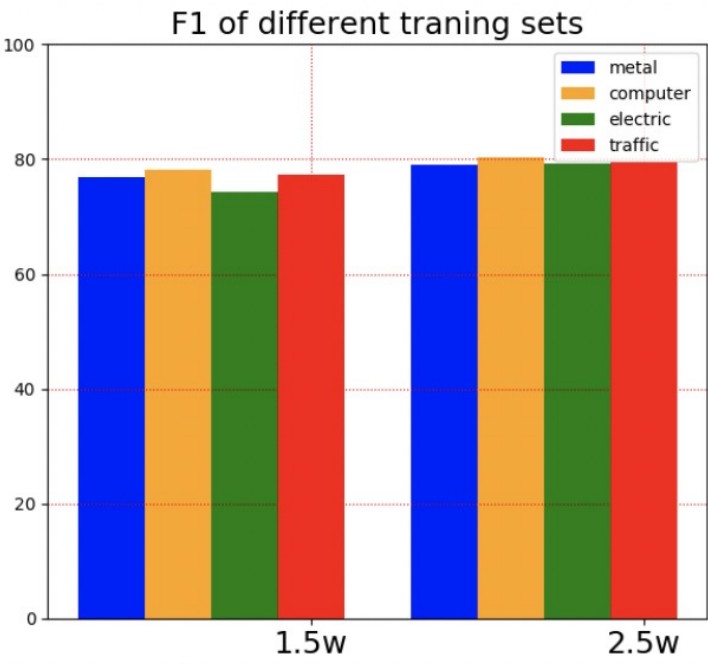

Figure 4:  F1 results of NER.

Table 12: F1 results of NER.

| | | F1 |
|---|---|---|
| CoNll2003 | | 90.94 |
| TechNER | Metal | 76.86 |
| | Computer | 74.36 |
| | Electric | 77.21 |
| | Traffic | 78.19 |
| TechNER (large) | Metal | 79.01 |
| | Computer | 80.33 |
| | Electric | 79.31 |
| | Traffic | 79.52 |

In this study, we select *LSTM-CRF* Lample et al. (2016), one of the state-of-the-art NER methods as baseline and evaluate its performance on recognizing domain terminologies.

In each domain of TechNER, the number of training number is far larger than CoNll2003. So in experiments, we randomly select 21,000 training samples for each selected domain. By this way, there is a similar sample size between CoNll2003 (also about 21,000) and TechNER. In experiments, we select 15,000 samples for training, 3,500 samples for both the development set and the test set. During training, we iterate 300 rounds for each experiment. Finally, the experimental results are shown in Table 12 and Figure 4, from which we can find following observations.

First, there is a big performance gap between CoNll2003 and TechNER. In fact, the terminology recognition task is a more challenging task than the traditional NER task. The main reason is that lots of useful features that work well in a traditional NER task cannot work here. For example, capitalizations, prefix and suffix feature, collocations, all of these features are very useful in a tradition NER task. But none of them can be used here. What's worse, as observed in the previous RE task, there are also almost no any cues in the training sentences that can imply the boundaries of a terminology.

Second, there is also an obvious performance difference between different kinds of domains. But compared with the previous KGE or RE tasks, the performance gap between different kinds of domains here is far smaller.

One may wonder whether we could obtain better performance by increasing the number of training samples. To answer this question, we select 25,000 training sentences for each domain and conduct the experiments again. Other experiment settings remain unchanged. The new results are denoted as *large* in Table 12. We can see that it is true that there is a minor improvement on the F1 values for each domain by increasing the number of training samples. But we find the performance gaps between different kinds of domains couldn't be eliminated by this way.

## B   DETAILED STATISTICS FOR TECHKG

Supposing there is a paper whose basic information is listed as:

Table 13: Basic Information of a Paper

| Title | Authors | Affiliation | Keywords (chn) | Keywords (eng) | domain | Published year | Published journal |
|-------|---------|-------------|----------------|----------------|--------|----------------|-------------------|
| $p$ | $a_1$, $a_2$,..., $a_m$ | $o_1$, $o_2$, ..., $o_s$ | $ck_1$, $ck_2$, ..., $ck_n$ | $ek_l$, $ek_2$, ..., $ek_l$ | $d_i$ | $y$ | $j$ |

Then the corresponding relations obtained are as listed.

Table 14: Samples of Extracted Relations

| relations | Extracted relations |
|-----------|---------------------|
| author_of | $<a_1$, author_of, $p>$, $<a_2$, author_of, $p>$, ..., $<a_m$, author_of, $p>$ |
| first_author | $<a_1$, first_author, $p>$ |
| second_author | $<a_2$, second_author, $p>$ |
| co_author | $<a_1$, co_author, $a_2>$, $<a_1$, co_author, $a_3>$,..., $<a_{m-1}$, co_author, $a_m>$ |
| research_interest | $<a_1$, research_interest, $ck_1>$, $<a_1$, research_interest, $ck_2>$,..., $<a_m$, research_interest, $ck_n>$ |
| affiliation | $<a_1$, affiliation, $o_1>$,..., $<a_1$, affiliation, $o_2>$, ..., $<a_m$, affiliation, $o_s>$ |
| belged_domain | $<ck_1$, belged_domain, $d_i>$, ..., $<ck_n$, belged_domain, $d_i>$ |
| co-occurrence_with | $<ck_1$, co-occurrence_with, $ck_2>$, ...,$<ck_1$, co-occurrence_with, $ck_n>$, ...,$<ck_{n-1}$, co-occurrence_with, $ck_n>$ |
| published_year | $<p$, published_year, $y>$ |
| contained_cky | $<p$, contained_cky, $ck_1$+ $ck_2$+...+ $ck_n>$ |
| contained_eky | $<p$, contained_eky, $ek_1$+ $ek_2$+...+ $ek_l>$ |
| other_author | $<a_3$, other_author, $p>$, $<a_4$, other_author, $p>$, ..., $<a_m$, other_author, $p>$, |
| all_authors | $<p$, all_authors, $a_1$+ $a_2$+...+ $a_m>$ |
| published_journal | $<p$, published_journal, $j>$ |
| translation_of | $<ck_1$, translation_of, $ek_1>$, $<ck_2$, translation_of, $ek_2>$, ..., $<ck_s$, translation_of, $ek_s>$. (Threshold requirement should be met.) |

Other statistics information about TechKG is shown in the following tables.

Table 15: Detailed Statistics of TechKG

| | Law | | MedicineRelated | | PoliticsRelated | | NatureScience Related | |
|---|---|---|---|---|---|---|---|---|
| | Ent# | Tri# | Ent# | Tri# | Ent# | Tri# | Ent# | Tri# |
| author_of | 52763 | 42219 | 1614774 | 4074249 | 123342 | 94895 | 790242 | 1120957 |
| first_author | 48051 | 32563 | 1297282 | 990282 | 112876 | 73466 | 632613 | 437183 |
| second_author | 14199 | 7917 | 1154040 | 890725 | 30285 | 16851 | 470032 | 320782 |
| co_author | 11312 | 11921 | 609337 | 5899008 | 25958 | 27953 | 326878 | 1147280 |
| research_inter | 61760 | 137344 | 1114364 | 12239418 | 120958 | 283638 | 883893 | 3920870 |
| affiliation | 5295 | 5770 | 583627 | 2514451 | 10602 | 9605 | 388660 | 1252779 |
| belged_domain | 101466 | 101466 | 841224 | 841224 | 296130 | 296130 | 688301 | 688301 |
| co-occurrence_with | 101293 | 594562 | 838379 | 8703065 | 295547 | 2922678 | 687361 | 3472226 |
| published_year | 147795 | 149318 | 2346223 | 2385304 | 835976 | 849274 | 710377 | 715292 |
| contained_cky | 180710 | 91544 | 3805573 | 1946078 | 928248 | 476598 | 1166195 | 588306 |
| contained_eky | 46719 | 23425 | 1441121 | 725911 | 97514 | 48927 | 777421 | 390670 |
| other_author | 2356 | 1740 | 1189796 | 2193664 | 6797 | 4591 | 372259 | 363014 |
| all_authors | 52282 | 32566 | 1883060 | 991031 | 120923 | 73499 | 807775 | 437306 |
| publi_journal | 147880 | 149268 | 2347191 | 2389266 | 836394 | 861521 | 710725 | 714789 |
| translation_of | 153835 | 93709 | 3295408 | 2142785 | 295830 | 181573 | 2229279 | 1418314 |
| Total | 475944 | 1475362 | 9308021 | 48928685 | 1870480 | 6221255 | 4378579 | 16988413 |

| | SportRelated | | PublishingRelated | | EnvironmentRelated | | FisheriesRelated | |
|---|---|---|---|---|---|---|---|---|
| | Ent# | Tri# | Ent# | Tri# | Ent# | Tri# | Ent# | Tri# |
| author_of | 87482 | 99025 | 73092 | 68834 | 192889 | 272905 | 41081 | 68587 |
| first_author | 75790 | 53981 | 64202 | 43271 | 141195 | 92267 | 28857 | 19450 |
| second_author | 40008 | 26217 | 27577 | 17103 | 109566 | 72367 | 23867 | 16258 |
| co_author | 26775 | 67321 | 20556 | 33123 | 93953 | 318459 | 20714 | 90368 |
| research_inter | 71352 | 318380 | 72235 | 221583 | 200140 | 961108 | 46784 | 230432 |
| affiliation | 22495 | 55028 | 8409 | 8987 | 48544 | 94239 | 27160 | 86679 |
| belged_domain | 87841 | 87841 | 138747 | 138747 | 163699 | 163698 | 49305 | 49304 |
| co-occurrence_with | 87648 | 754956 | 138468 | 1032810 | 163507 | 910402 | 49250 | 323270 |
| published_year | 158179 | 160267 | 260025 | 262332 | 193861 | 196357 | 66708 | 68089 |
| contained_cky | 195815 | 100011 | 302989 | 153703 | 287999 | 145679 | 93183 | 47365 |
| contained_eky | 70894 | 35550 | 51663 | 25916 | 129718 | 65010 | 28167 | 14122 |
| other_author | 22320 | 18836 | 11947 | 8460 | 105294 | 108288 | 26711 | 32881 |
| all_authors | 91601 | 54012 | 73250 | 43282 | 174118 | 92329 | 36878 | 19456 |
| publi_journal | 158231 | 159533 | 260121 | 263231 | 193928 | 195690 | 66705 | 68240 |
| translation_of | 201234 | 127884 | 157651 | 95069 | 392207 | 239751 | 99742 | 62492 |
| Total | 556452 | 2118866 | 705674 | 2416572 | 961988 | 3928619 | 278473 | 1197019 |

| | MetallurgicalIndustry | | PhysicsRelated | | History&Geography | | PowerEngineering | |
|---|---|---|---|---|---|---|---|---|
| | Ent# | Tri# | Ent# | Tri# | Ent# | Tri# | Ent# | Tri# |
| author_of | 303394 | 465580 | 129777 | 225201 | 63257 | 54465 | 130051 | 186652 |
| first_author | 236332 | 162211 | 90422 | 59140 | 55766 | 36680 | 93176 | 60050 |
| second_author | 173593 | 118676 | 77858 | 52303 | 16265 | 9623 | 72376 | 47147 |
| co_author | 128389 | 499336 | 69062 | 383499 | 15502 | 34318 | 65074 | 240627 |
| research_inter | 307348 | 1507161 | 160426 | 727998 | 79819 | 161376 | 153163 | 645826 |
| affiliation | 129811 | 385070 | 60320 | 173495 | 8296 | 10481 | 76361 | 195820 |
| belged_domain | 244813 | 244812 | 107366 | 107365 | 147371 | 147370 | 120266 | 120265 |
| co-occurrence_with | 244494 | 1391266 | 107217 | 394796 | 147128 | 956521 | 120139 | 560521 |
| published_year | 300589 | 304862 | 72691 | 73636 | 190612 | 192141 | 112125 | 113461 |
| contained_cky | 465548 | 235547 | 127151 | 64104 | 224319 | 113625 | 178792 | 90362 |
| contained_eky | 253867 | 127440 | 96534 | 48424 | 30352 | 15195 | 92333 | 46362 |
| other_author | 160783 | 184747 | 85364 | 113764 | 9639 | 8162 | 72243 | 79466 |

| | | | | | | | | |
|---|---|---|---|---|---|---|---|---|
| all_authors | 299221 | 162368 | 112354 | 59153 | 60325 | 36682 | 113688 | 60093 |
| publi_journal | 300694 | 304045 | 72694 | 72821 | 190705 | 191887 | 112151 | 113069 |
| translation_of | 738605 | 463472 | 320236 | 192795 | 150664 | 84112 | 309387 | 190816 |
| Total | 1602651 | 6556905 | 601142 | 2748591 | 580138 | 2052750 | 673566 | 2750582 |

| | MathRelated | | Military | | Machinery | | ArchitectureRelated | |
|---|---|---|---|---|---|---|---|---|
| | Ent# | Tri# | Ent# | Tri# | Ent# | Tri# | Ent# | Tri# |
| author_of | 85298 | 97359 | 17287 | 19647 | 317186 | 451634 | 378001 | 488228 |
| first_author | 70164 | 47958 | 12936 | 8013 | 245344 | 163764 | 305670 | 209767 |
| second_author | 48289 | 32372 | 8137 | 4860 | 193264 | 132597 | 209722 | 140807 |
| co_author | 34183 | 60632 | 7788 | 21704 | 143882 | 445428 | 147138 | 422327 |
| research_inter | 109831 | 318530 | 19267 | 56058 | 348790 | 1575060 | 394395 | 1726944 |
| affiliation | 51141 | 137190 | 6270 | 16530 | 90596 | 193492 | 105555 | 296834 |
| belged_domain | 84561 | 84560 | 45274 | 45274 | 271134 | 271133 | 380489 | 380488 |
| co-occurrence_with | 84334 | 303192 | 45040 | 215970 | 270817 | 1414790 | 380032 | 2586344 |
| published_year | 59760 | 60059 | 50490 | 50750 | 289942 | 292878 | 530158 | 536351 |
| contained_cky | 106065 | 53583 | 64307 | 32532 | 476547 | 240539 | 810061 | 409713 |
| contained_eky | 72836 | 36589 | 6474 | 3254 | 242015 | 121259 | 312200 | 156465 |
| other_author | 24575 | 17035 | 6265 | 6774 | 166201 | 155290 | 153569 | 137709 |
| all_authors | 84109 | 47980 | 14929 | 8016 | 308244 | 163825 | 382513 | 209896 |
| publi_journal | 59766 | 59912 | 50485 | 50582 | 290041 | 292305 | 530370 | 537169 |
| translation_of | 270735 | 167022 | 30459 | 17530 | 736666 | 449162 | 941452 | 585890 |
| Total | 468289 | 1524016 | 162185 | 557500 | 1620300 | 6363473 | 2321365 | 8825463 |

| | AgricultureRelated | | ElectricIndustry | | Forestry | | MaterialScience | |
|---|---|---|---|---|---|---|---|---|
| | Ent# | Tri# | Ent# | Tri# | Ent# | Tri# | Ent# | Tri# |
| author_of | 454171 | 875559 | 321536 | 465500 | 111149 | 163046 | 66045 | 97167 |
| first_author | 335916 | 235246 | 259346 | 173736 | 78978 | 52202 | 42734 | 26317 |
| second_author | 290594 | 207008 | 189222 | 132736 | 62782 | 41477 | 37712 | 24415 |
| co_author | 211507 | 1198644 | 135927 | 460054 | 55719 | 206904 | 39170 | 132590 |
| research_inter | 432607 | 2942049 | 359189 | 1579019 | 120707 | 564252 | 77469 | 340315 |
| affiliation | 200113 | 732908 | 125865 | 347644 | 30375 | 65086 | 46816 | 128112 |
| belged_domain | 371021 | 371021 | 324776 | 324776 | 113026 | 113026 | 49322 | 49322 |
| co-occurrence_with | 370559 | 3208824 | 324348 | 1875036 | 112852 | 660413 | 49271 | 197269 |
| published_year | 705112 | 718518 | 400724 | 406143 | 140191 | 141419 | 37363 | 37664 |
| contained_cky | 1018196 | 516386 | 581225 | 294232 | 208406 | 104973 | 62587 | 31489 |
| contained_eky | 399977 | 200569 | 284228 | 142586 | 91247 | 45698 | 45868 | 23018 |
| other_author | 315185 | 433346 | 152853 | 159046 | 65013 | 69377 | 44179 | 46435 |
| all_authors | 450364 | 235326 | 320083 | 173835 | 99092 | 52224 | 50837 | 26330 |
| publi_journal | 705422 | 722127 | 400873 | 405742 | 140251 | 141260 | 37348 | 37413 |
| translation_of | 1048945 | 684803 | 834585 | 520957 | 258882 | 161649 | 145708 | 90958 |
| Total | 2854050 | 13282530 | 1912996 | 7461378 | 654388 | 2583059 | 295413 | 1288937 |

| | ArtRelated | | PowerIndustry | | Mechanics | | Aerospace | |
|---|---|---|---|---|---|---|---|---|
| | Ent# | Tri# | Ent# | Tri# | Ent# | Tri# | Ent# | Tri# |
| author_of | 51753 | 39151 | 242128 | 334368 | 37258 | 47743 | 127097 | 168772 |
| first_author | 48269 | 31699 | 192127 | 129725 | 26885 | 17255 | 100993 | 67146 |
| second_author | 8962 | 5391 | 139685 | 94340 | 23155 | 15241 | 71143 | 48484 |
| co_author | 8391 | 14382 | 101355 | 320363 | 19463 | 45339 | 54175 | 151348 |
| research_inter | 47206 | 91295 | 249826 | 1152985 | 52518 | 178325 | 144054 | 554489 |
| affiliation | 1983 | 1699 | 111773 | 321825 | 25503 | 64698 | 39971 | 89938 |
| belged_domain | 125637 | 125637 | 214309 | 214309 | 35907 | 35907 | 128325 | 128325 |
| co-occurrence_with | 125292 | 1289926 | 214040 | 1246853 | 35894 | 127426 | 128172 | 577962 |

| | | | | | | | | |
|---|---|---|---|---|---|---|---|---|
| published_year | 254324 | 259120 | 274742 | 277813 | 19766 | 19811 | 120960 | 122302 |
| contained_cky | 272725 | 140982 | 421708 | 213552 | 36737 | 18448 | 175622 | 88603 |
| contained_eky | 13831 | 6935 | 201966 | 101368 | 31070 | 15581 | 89499 | 44824 |
| other_author | 2598 | 2070 | 112516 | 110354 | 18710 | 15249 | 58155 | 53148 |
| all_authors | 50464 | 31764 | 240252 | 129859 | 32463 | 17259 | 121975 | 67168 |
| publi_journal | 254455 | 258339 | 274840 | 278110 | 19746 | 19751 | 120996 | 121511 |
| translation_of | 98210 | 53865 | 588788 | 366697 | 112914 | 68574 | 308614 | 183263 |
| Total | 593371 | 2352315 | 1357767 | 5292736 | 193028 | 706631 | 661511 | 2467501 |

| | Bioscience | | AnimalHusbandry | | EnergyRelated | | MiningIndustry | |
|---|---|---|---|---|---|---|---|---|
| | Ent# | Tri# | Ent# | Tri# | Ent# | Tri# | Ent# | Tri# |
| author_of | 220383 | 374054 | 112335 | 199300 | 199497 | 295852 | 171235 | 211116 |
| first_author | 149691 | 98705 | 75338 | 49653 | 149836 | 101651 | 130207 | 84686 |
| second_author | 134009 | 90038 | 66178 | 44665 | 113533 | 75287 | 92642 | 59149 |
| co_author | 120131 | 565274 | 61213 | 312094 | 90241 | 355897 | 77261 | 206023 |
| research_inter | 241546 | 1298293 | 117775 | 661520 | 206738 | 1185271 | 200331 | 763546 |
| affiliation | 148883 | 492152 | 42939 | 106813 | 99350 | 350778 | 50093 | 96712 |
| belged_domain | 152943 | 152943 | 144486 | 144485 | 158772 | 158771 | 174923 | 174923 |
| co-occurrence_with | 152754 | 743139 | 144286 | 1165478 | 158635 | 1052945 | 174782 | 825785 |
| published_year | 136963 | 138050 | 269495 | 277489 | 171945 | 173624 | 161687 | 163021 |
| contained_cky | 231780 | 116802 | 341725 | 174647 | 283221 | 142712 | 288575 | 145345 |
| contained_eky | 154771 | 77745 | 83811 | 42076 | 159958 | 80159 | 135982 | 68140 |
| other_author | 150487 | 185321 | 78441 | 105005 | 112089 | 118949 | 77402 | 67287 |
| all_authors | 188665 | 98731 | 95842 | 49687 | 190843 | 101739 | 158881 | 84728 |
| publi_journal | 137006 | 137439 | 269574 | 280387 | 172004 | 172952 | 161736 | 162769 |
| translation_of | 501006 | 320686 | 234783 | 149316 | 485747 | 307204 | 426639 | 260175 |
| Total | 1016197 | 4889510 | 873713 | 3762703 | 1025518 | 4673947 | 943136 | 3373454 |

| | Astronomy | | CultureRelated | | LightIndustry | | SocialScience&Philosophy | |
|---|---|---|---|---|---|---|---|---|
| | Ent# | Tri# | Ent# | Tri# | Ent# | Tri# | Ent# | Tri# |
| author_of | 358797 | 632889 | 106430 | 85699 | 256322 | 352752 | 800417 | 763626 |
| first_author | 282323 | 203774 | 99818 | 66896 | 201018 | 140177 | 733745 | 536810 |
| second_author | 228664 | 165890 | 24425 | 15661 | 140552 | 96315 | 264352 | 165644 |
| co_author | 147335 | 742940 | 19338 | 28906 | 104138 | 335546 | 178099 | 292637 |
| research_inter | 402486 | 2243002 | 87407 | 187610 | 257723 | 1162402 | 769605 | 2636381 |
| affiliation | 103486 | 329883 | 5001 | 4938 | 90172 | 245136 | 163685 | 301436 |
| belged_domain | 311435 | 311434 | 210159 | 210159 | 272246 | 272245 | 794635 | 794635 |
| co-occurrence_with | 311083 | 1674665 | 209775 | 2967853 | 271629 | 1906011 | 793704 | 6250869 |
| published_year | 306976 | 310215 | 483035 | 505601 | 373662 | 378913 | 1351764 | 1363281 |
| contained_cky | 515453 | 259807 | 491900 | 254748 | 542107 | 275366 | 2144814 | 1084329 |
| contained_eky | 318143 | 159516 | 44204 | 22132 | 203655 | 102299 | 926805 | 465458 |
| other_author | 216826 | 263252 | 4236 | 3157 | 116655 | 116307 | 85617 | 61183 |
| all_authors | 381349 | 203848 | 106413 | 66954 | 250632 | 140330 | 839278 | 536933 |
| publi_journal | 307168 | 308563 | 483330 | 511041 | 373822 | 379262 | 1352491 | 1367087 |
| translation_of | 974101 | 610861 | 183399 | 108123 | 607602 | 377253 | 2364399 | 1530996 |
| Total | 1918791 | 8420905 | 1101328 | 5039721 | 1577973 | 6280999 | 5482685 | 18151694 |

| | Education | | TrafficTransportation | | IndustrialTechnology | | Finance | |
|---|---|---|---|---|---|---|---|---|
| | Ent# | Tri# | Ent# | Tri# | Ent# | Tri# | Ent# | Tri# |
| author_of | 309337 | 260059 | 270505 | 323174 | 132743 | 152571 | 366453 | 365594 |
| first_author | 279552 | 183452 | 223218 | 149307 | 94819 | 58153 | 323869 | 227102 |
| second_author | 94165 | 56786 | 140807 | 92900 | 69201 | 42856 | 150355 | 97077 |
| co_author | 75746 | 98772 | 102908 | 253687 | 67728 | 163072 | 101406 | 177472 |
| research_inter | 282404 | 807177 | 283403 | 1117265 | 166966 | 534338 | 321837 | 1135657 |

| | | | | | | | | |
|---|---|---|---|---|---|---|---|---|
| affiliation | 45382 | 51812 | 73780 | 167084 | 67516 | 137114 | 65015 | 107884 |
| belged_domain | 459339 | 459341 | 290242 | 290242 | 188271 | 188270 | 544014 | 544013 |
| co-occurrence_with | 458409 | 6487553 | 289732 | 1928432 | 187940 | 849757 | 542917 | 6446286 |
| published_year | 1387391 | 1435873 | 410641 | 415921 | 174508 | 176551 | 1761391 | 1785618 |
| contained_cky | 1759244 | 910876 | 609960 | 309770 | 253069 | 127808 | 2029112 | 1037511 |
| contained_eky | 265118 | 132834 | 224883 | 112727 | 87151 | 43654 | 311787 | 156461 |
| other_author | 29941 | 19824 | 97981 | 80988 | 60791 | 51567 | 58789 | 41426 |
| all_authors | 307335 | 183556 | 270693 | 149382 | 110228 | 58169 | 378969 | 227176 |
| publi_journal | 1388214 | 1450263 | 410810 | 415172 | 174555 | 175862 | 1762158 | 1795718 |
| translation_of | 712538 | 447052 | 665226 | 408883 | 317372 | 189344 | 890097 | 553170 |
| Total | 3439775 | 12985548 | 1708950 | 6215738 | 836742 | 2949164 | 4196406 | 14698754 |

| | ComputerScience | | Chemistry | | | | | |
|---|---|---|---|---|---|---|---|---|
| | Ent# | Tri# | Ent# | Tri# | | | | |
| author_of | 317782 | 443985 | 529270 | 921386 | | | | |
| first_author | 255881 | 169619 | 414458 | 294746 | | | | |
| second_author | 198351 | 140089 | 338716 | 240198 | | | | |
| co_author | 138764 | 391289 | 222034 | 1031983 | | | | |
| research_interest | 350332 | 1561345 | 500015 | 3019792 | | | | |
| affiliation | 133206 | 358802 | 247832 | 836399 | | | | |
| belged_domain | 294829 | 294828 | 368382 | 368382 | | | | |
| co-occurrence_with | 294450 | 1717881 | 367962 | 2363407 | | | | |
| published_year | 353439 | 356711 | 485128 | 493364 | | | | |
| contained_cky | 526722 | 266678 | 773325 | 392785 | | | | |
| contained_eky | 275255 | 138010 | 472653 | 238015 | | | | |
| other_author | 150728 | 134301 | 315932 | 386539 | | | | |
| all_authors | 318746 | 169666 | 545942 | 295031 | | | | |
| publi_journal | 353530 | 356221 | 485342 | 492794 | | | | |
| translation_of | 829241 | 520559 | 1295177 | 836286 | | | | |
| Total | 1818503 | 7020316 | 2734418 | 12211721 | | | | |

Table 16: Detailed cross-domain duplicate author names' statistics in TechKG

| domain | total # | dup # | dup ratio | dpa |
|---|---|---|---|---|
| Computer science | 148449 | 117756 | 79.32% | 7.14 |
| Mechanics | 20006 | 18244 | 91.19% | 9.82 |
| Power | 70066 | 56823 | 81.10% | 8.28 |
| Publishing-related | 29898 | 21528 | 72.00% | 10.48 |
| Forestry | 58993 | 43192 | 73.22% | 8.72 |
| Industrial technology | 74618 | 63579 | 85.21% | 8.41 |
| Math-related | 37454 | 30614 | 81.74% | 9.19 |
| Environment-related | 100746 | 83588 | 82.97% | 7.82 |
| Fisheries-related | 21646 | 17335 | 80.08% | 9.79 |
| Animal Husbandry | 62798 | 48417 | 77.10% | 8.39 |
| Architecture-related | 168623 | 115012 | 68.21% | 7.17 |
| Aerospace | 60002 | 47865 | 79.77% | 8.96 |
| Light Industry | 116481 | 83373 | 71.58% | 7.73 |
| Finance | 139736 | 98043 | 70.16% | 7.41 |
| Physics-related | 70654 | 57741 | 81.72% | 8.24 |
| Traffic transportation | 121514 | 85343 | 70.23% | 7.76 |
| History&Geography | 26600 | 18764 | 70.54% | 10.05 |
| Agriculture-related | 219169 | 161034 | 73.47% | 6.35 |
| Machinery | 153615 | 123370 | 80.31% | 7.11 |
| Chemistry | 235458 | 167041 | 70.94% | 6.43 |
| Law | 20248 | 14762 | 72.91% | 10.71 |
| Energy-related | 97985 | 70792 | 72.25% | 7.94 |
| Astronomy | 155138 | 113430 | 73.12% | 7.01 |
| Education | 127318 | 82920 | 65.13% | 7.66 |
| Power industry | 112684 | 79670 | 70.70% | 8.04 |
| Art-related | 20282 | 11917 | 58.76% | 11.42 |
| Material science | 39748 | 35744 | 89.93% | 9.19 |
| Metallurgical industry | 141458 | 102116 | 72.19% | 7.42 |
| Electric industry | 148079 | 110185 | 74.41% | 7.3 |
| Politics-related | 50026 | 34111 | 68.19% | 9.29 |
| Military | 9276 | 7475 | 80.58% | 10.94 |
| Medicine-related | 628999 | 258228 | 41.05% | 5.53 |
| Bioscience | 121779 | 109054 | 89.55% | 6.73 |
| Mining industry | 86681 | 66567 | 76.80% | 8.01 |
| Culture-related | 40363 | 21880 | 54.21% | 9.68 |
| Sport-related | 33668 | 21412 | 63.60% | 10.84 |
| Total | 1826217 | 654884 | 35.86% | 3.97 |

Table 17: Detailed distribution of domain numbers for cross-domain duplicated authors in TechKG

| domain | Ratio(%) | | | | | number | | | | |
|---|---|---|---|---|---|---|---|---|---|---|
| | [1,5) | [5,10) | [10,20) | [20,30) | [30,36) | [1,5) | [5,10) | [10,20) | [20,30) | [30,36) |
| ComputerScience | 58.05 | 21.51 | 14.13 | 5.12 | 1.18 | 68363 | 25334 | 16640 | 6029 | 1390 |
| Mechanics | 44.98 | 24.5 | 15.94 | 9.48 | 5.09 | 8207 | 4470 | 2909 | 1730 | 928 |
| Power | 53.36 | 20.78 | 15.64 | 7.85 | 2.37 | 30319 | 11807 | 8887 | 4463 | 1347 |
| PublishingRelated | 45.63 | 18.26 | 18.25 | 12.37 | 5.5 | 9823 | 3930 | 3929 | 2662 | 1184 |
| Forestry | 52.47 | 19.93 | 15.7 | 8.88 | 3.02 | 22665 | 8607 | 6781 | 3834 | 1305 |
| IndustrialTech | 49.81 | 24.55 | 16.03 | 7.46 | 2.15 | 31667 | 15611 | 10192 | 4741 | 1368 |
| MathRelated | 49.43 | 21.42 | 15.98 | 9.25 | 3.93 | 15131 | 6557 | 4893 | 2831 | 1202 |
| EnvironmentRel | 53.73 | 23.17 | 14.95 | 6.5 | 1.65 | 44913 | 19365 | 12497 | 5430 | 1383 |
| FisheriesRelated | 48.48 | 20.73 | 15.13 | 10.03 | 5.62 | 8404 | 3594 | 2623 | 1739 | 975 |
| AnimalHusbandry | 54.5 | 19.47 | 15.08 | 8.21 | 2.75 | 26385 | 9427 | 7300 | 3973 | 1332 |
| ArchitectureRel | 57.95 | 21.16 | 14.44 | 5.24 | 1.21 | 66652 | 24337 | 16604 | 6030 | 1389 |
| Aerospace | 48.99 | 22.21 | 16.92 | 9.04 | 2.83 | 23451 | 10631 | 8101 | 4326 | 1356 |
| LightIndustry | 55.47 | 21 | 15.33 | 6.54 | 1.66 | 46245 | 17505 | 12784 | 5455 | 1384 |
| Finance | 57.65 | 19.82 | 15.12 | 6 | 1.42 | 56518 | 19432 | 14821 | 5882 | 1390 |
| PhysicsRelated | 53.18 | 21.55 | 15.32 | 7.6 | 2.36 | 30705 | 12441 | 8845 | 4388 | 1362 |
| TrafficTrans | 54.79 | 21.42 | 15.66 | 6.52 | 1.62 | 46758 | 18279 | 13362 | 5562 | 1382 |
| History&Geography | 48.4 | 18.92 | 15.81 | 11 | 5.87 | 9082 | 3550 | 2967 | 2064 | 1101 |
| Agriculture-related | 63.93 | 19.65 | 11.73 | 3.83 | 0.86 | 102942 | 31637 | 18893 | 6172 | 1390 |
| Machinery | 57.47 | 22.64 | 13.89 | 4.87 | 1.13 | 70904 | 27930 | 17139 | 6007 | 1390 |
| Chemistry | 62.33 | 20.96 | 12.14 | 3.74 | 0.83 | 104116 | 35010 | 20282 | 6243 | 1390 |
| Law | 47.49 | 16.24 | 16.65 | 12.62 | 7 | 7010 | 2398 | 2458 | 1863 | 1033 |
| EnergyRelated | 55.13 | 19.83 | 15.85 | 7.25 | 1.93 | 39027 | 14038 | 11224 | 5134 | 1369 |
| Astronomy | 59.74 | 20.51 | 13.35 | 5.18 | 1.23 | 67760 | 23266 | 15140 | 5874 | 1390 |
| Education | 56.53 | 19.7 | 15.44 | 6.66 | 1.67 | 46877 | 16334 | 12803 | 5521 | 1385 |
| PowerIndustry | 52.94 | 21.73 | 16.61 | 6.98 | 1.74 | 42181 | 17312 | 13231 | 5561 | 1385 |
| ArtRelated | 43.85 | 16.59 | 17.73 | 13.75 | 8.07 | 5226 | 1977 | 2113 | 1639 | 962 |
| MaterialScience | 47.17 | 24.39 | 15.91 | 9.03 | 3.51 | 16859 | 8718 | 5687 | 3226 | 1254 |
| MetallurgicalIndustry | 56.08 | 22.13 | 14.75 | 5.67 | 1.36 | 57269 | 22602 | 15067 | 5791 | 1387 |
| ElectricIndustry | 57.03 | 21.63 | 14.66 | 5.42 | 1.26 | 62836 | 23832 | 16156 | 5971 | 1390 |
| PoliticsRelated | 51.39 | 17.62 | 16.68 | 10.44 | 3.87 | 17529 | 6011 | 5690 | 3560 | 1321 |
| Military | 49.18 | 14.42 | 15.48 | 11.71 | 9.22 | 3676 | 1078 | 1157 | 875 | 689 |
| MedicineRelated | 69.07 | 18.54 | 9.38 | 2.46 | 0.54 | 178357 | 47887 | 24231 | 6363 | 1390 |
| Bioscience | 63.43 | 18.09 | 12.12 | 5.09 | 1.27 | 69173 | 19724 | 13215 | 5554 | 1388 |
| MiningIndustry | 54.14 | 21.37 | 15.3 | 7.14 | 2.05 | 36037 | 14226 | 10188 | 4750 | 1366 |
| Culture-related | 51.6 | 15.95 | 16.17 | 11 | 5.29 | 11289 | 3489 | 3537 | 2407 | 1158 |
| Sport-related | 44.26 | 17.86 | 18.54 | 13.52 | 5.82 | 9478 | 3825 | 3969 | 2894 | 1246 |
| Total | 83.69 | 11.16 | 3.97 | 0.97 | 0.21 | 548067 | 73082 | 25976 | 6369 | 1390 |

Table 18: Extreme analysis of in-domain duplicate name issuefor authors in TechKG

| | TechKG | | | |
| | | Worse/better | | |
| domain | total # | dup # | dupRat(%) | apa |
| --- | --- | --- | --- | --- |
| Computer science | 95163 | 54333/52719 | 57.09/55.4 | 4.25/4.31 |
| Mechanics | 17889 | 10912/10375 | 61.00/58 | 3.65/3.72 |
| Power | 55702 | 32167/30760 | 57.75/55.22 | 3.77/3.84 |
| Publishing-related | 5759 | 1815/1678 | 31.52/29.14 | 2.47/2.5 |
| Forestry | 21918 | 14355/14051 | 65.49/64.11 | 3.37/3.4 |
| Industrial technology | 49038 | 25715/24828 | 52.4450.63 | 3.22/3.26 |
| Math-related | 33208 | 20391/19886 | 61.40/59.88 | 3.81/3.84 |
| Environment-related | 34376 | 20724/20294 | 60.29/59.04 | 3.26/3.28 |
| Fisheries-related | 18229 | 12657/12423 | 69.43/68.15 | 4.39/4.43 |
| Animal Husbandry | 31218 | 20881/20483 | 66.89/65.61 | 3.75/3.78 |
| Architecture-related | 73993 | 48273/47424 | 65.24/64.09 | 4.12/4.16 |
| Aerospace | 30907 | 16441/15567 | 53.20/50.37 | 3.26/3.31 |
| Light Industry | 62895 | 37217/36317 | 59.17/57.74 | 3.91/3.95 |
| Finance | 45630 | 21017/20185 | 46.06/44.24 | 3.07/3.11 |
| Physics-related | 44005 | 27282/26282 | 62.00/59.73 | 3.88/3.94 |
| Traffic transportation | 53354 | 29661/28843 | 55.59/54.06 | 3.44/3.48 |
| History&Geography | 5248 | 2214/2171 | 42.19/41.37 | 2.69/2.7 |
| Agriculture-related | 139950 | 101405/99843 | 72.46/71.34 | 5.09/5.14 |
| Machinery | 67320 | 37097/35464 | 55.11/52.68 | 3.53/3.58 |
| Chemistry | 175775 | 109847/106931 | 62.49/60.83 | 4.93/5 |
| Law | 3703 | 1074/1007 | 29.00/27.19 | 2.33/2.35 |
| Energy-related | 71127 | 48009/46934 | 67.50/65.99 | 5/5.06 |
| Astronomy | 73530 | 50681/49884 | 68.93/67.84 | 4.45/4.49 |
| Education | 30035 | 9905/9250 | 32.98/30.8 | 2.66/2.68 |
| Power industry | 80132 | 46692/44651 | 58.27/55.72 | 4.07/4.16 |
| Art-related | 1203 | 229/204 | 19.04/16.96 | 2.32/2.36 |
| Material science | 34796 | 21341/20403 | 61.33/58.64 | 3.66/3.72 |
| Social Science | 113015 | 51085/49083 | 45.20/43.43 | 3.23/3.27 |
| Metallurgical industry | 92637 | 56160/54065 | 60.62/58.36 | 4.38/4.46 |
| Nature Science | 282132 | 178220/175584 | 63.17/62.23 | 4.48/4.51 |
| Electric industry | 92230 | 51040/49055 | 55.34/53.19 | 4.19/4.26 |
| Politics-related | 7125 | 1801/1605 | 25.28/22.53 | 2.31/2.32 |
| Military | 4040 | 2603/2522 | 64.43/62.43 | 4.05/4.11 |
| Medicine-related | 369252 | 264850/254341 | 71.73/68.88 | 6.97/7.16 |
| Bioscience | 102912 | 73574/72176 | 71.49/70.13 | 4.7/4.74 |
| Mining industry | 35159 | 19645/19031 | 55.87/54.13 | 3.24/3.27 |
| Culture-related | 3043 | 931/851 | 30.59/27.97 | 2.33/2.35 |
| Sport-related | 15176 | 9635/9315 | 63.49/61.38 | 3.68/3.73 |

Table 19: Detailed cross-domain duplicate name statistics for keywords in TechKG

| domain | *Top-100% tf\*idf* | | | | *Top-90% tf\*idf* | | | |
|---|---|---|---|---|---|---|---|---|
| | total # | dup # | ratio(%) | *dpk* | total # | dup # | ratio(%) | *dpk* |
| Compute | 526348 | 300981 | 57.18% | 6.79 | 474907 | 249232 | 52.48% | 4.1 |
| Mechanics | 71687 | 44882 | 62.61% | 8.19 | 64954 | 38127 | 58.70% | 4.74 |
| Power | 210515 | 131515 | 62.47% | 8.92 | 189725 | 110675 | 58.33% | 5.25 |
| Publishing | 180589 | 123343 | 68.30% | 10.1 | 162713 | 105210 | 64.66% | 5.8 |
| Forestry | 199267 | 126190 | 63.33% | 9.34 | 180223 | 107068 | 59.41% | 5.35 |
| Industrial | 282980 | 206734 | 73.06% | 8.91 | 255301 | 178944 | 70.09% | 5.5 |
| Math | 162505 | 80477 | 49.52% | 8.05 | 146409 | 64232 | 43.87% | 4.16 |
| Environme | 273529 | 191278 | 69.93% | 8.6 | 248091 | 165728 | 66.80% | 5.2 |
| Fisheries | 76781 | 55605 | 72.42% | 10.5 | 69171 | 47948 | 69.32% | 5.82 |
| Animal | 216261 | 144946 | 67.02% | 8.04 | 196019 | 124489 | 63.51% | 4.7 |
| Architectur | 659210 | 342909 | 52.02% | 7.32 | 599571 | 283064 | 47.21% | 4.62 |
| Aerospace | 220928 | 141215 | 63.92% | 9.15 | 200530 | 120694 | 60.19% | 5.35 |
| LightIndu | 438951 | 268579 | 61.19% | 8.23 | 395126 | 224452 | 56.81% | 5.14 |
| Finance | 791894 | 465401 | 58.77% | 6.6 | 720474 | 393162 | 54.57% | 4.44 |
| Physics | 207825 | 107355 | 51.66% | 7.6 | 188154 | 87563 | 46.54% | 4.3 |
| Traffic | 489477 | 277050 | 56.60% | 7.95 | 441332 | 228785 | 51.84% | 4.93 |
| History& | 188186 | 120383 | 63.97% | 9.07 | 170467 | 102516 | 60.14% | 5.3 |
| Agriculture | 676833 | 396108 | 58.52% | 6.56 | 610980 | 329732 | 53.97% | 4.19 |
| Machinery | 486725 | 291766 | 59.94% | 7.1 | 438736 | 243665 | 55.54% | 4.43 |
| Chemistry | 711861 | 327319 | 45.98% | 6.95 | 646893 | 261512 | 40.43% | 4.34 |
| Law | 153659 | 89009 | 57.93% | 9.36 | 139365 | 74512 | 53.47% | 5.12 |
| Energy | 302269 | 162070 | 53.62% | 8.81 | 274507 | 134215 | 48.89% | 5.08 |
| Astronomy | 617698 | 286071 | 46.31% | 7.02 | 559855 | 228031 | 40.73% | 4.18 |
| Education | 677498 | 389173 | 57.44% | 7.1 | 615638 | 325615 | 52.89% | 4.6 |
| Power | 380101 | 204640 | 53.84% | 8.49 | 342605 | 167060 | 48.76% | 5.03 |
| Art-related | 144007 | 100390 | 69.71% | 9.78 | 130668 | 86964 | 66.55% | 5.67 |
| Material | 91489 | 62364 | 68.17% | 8.38 | 82881 | 53725 | 64.82% | 4.98 |
| Metal | 452839 | 235134 | 51.92% | 7.89 | 408569 | 190704 | 46.68% | 4.73 |
| Electronic | 565238 | 286802 | 50.74% | 7.15 | 515334 | 236652 | 45.92% | 4.31 |
| Politics | 393136 | 249428 | 63.45% | 7.78 | 354329 | 210204 | 59.32% | 4.92 |
| Military | 52641 | 45695 | 86.80% | 11.6 | 47481 | 40504 | 85.31% | 6.35 |
| Medicine* | 1545988 | 374140 | 24.20% | 6.57 | 1422954 | 251551 | 17.68% | 4.13 |
| Bioscience | 291019 | 204912 | 70.41% | 5.58 | 263812 | 177160 | 67.15% | 3.54 |
| Mining | 314728 | 180000 | 57.19% | 8.28 | 284098 | 149297 | 52.55% | 4.89 |
| Culture | 255354 | 168373 | 65.94% | 8.4 | 232608 | 145373 | 62.50% | 5.11 |
| Sport | 144830 | 86512 | 59.73% | 10.6 | 130877 | 72235 | 55.19% | 5.79 |
| Total | 8003868 | 1817771 | 22.71% | 4 | 8003293 | 1808536 | 22.60% | 3.32 |
| | *Top-80% tf\*idf* | | | | *Top-70% tf\*idf* | | | |
| domain | total # | dup # | ratio(%) | *dpk* | total # | dup # | ratio(%) | *dpk* |
| Compute | 432783 | 202643 | 46.82% | 3.35 | 402086 | 161525 | 40.17% | 3.14 |
| Mechanics | 57349 | 29984 | 52.28% | 3.68 | 50198 | 21615 | 43.06% | 3.49 |
| Power | 171732 | 90742 | 52.84% | 4.1 | 147571 | 62177 | 42.13% | 3.79 |
| Publishing | 144490 | 84285 | 58.33% | 4.48 | 126507 | 61255 | 48.42% | 4.14 |
| Forestry | 159579 | 84472 | 52.93% | 4.06 | 141923 | 62878 | 44.30% | 3.62 |
| Industrial | 232126 | 152772 | 65.81% | 4.32 | 200805 | 112225 | 55.89% | 3.93 |
| Math | 132352 | 47484 | 35.88% | 3.27 | 123901 | 36382 | 29.36% | 3.1 |
| Environme | 223614 | 137463 | 61.47% | 4.04 | 196297 | 100949 | 51.43% | 3.69 |
| Fisheries | 61822 | 39686 | 64.19% | 4.34 | 53762 | 30145 | 56.07% | 3.86 |
| Animal | 175868 | 100301 | 57.03% | 3.78 | 161030 | 82775 | 51.40% | 3.49 |
| Architectur | 528537 | 209321 | 39.60% | 3.74 | 490950 | 163085 | 33.22% | 3.52 |
| Aerospace | 177874 | 94729 | 53.26% | 4.11 | 154783 | 66294 | 42.83% | 3.83 |
| LightIndu | 351699 | 176036 | 50.05% | 4.11 | 312442 | 126896 | 40.61% | 3.85 |
| Finance | 667569 | 332178 | 49.76% | 3.69 | 619288 | 266974 | 43.11% | 3.48 |

| Physics | 166374 | 63948 | 38.44% | 3.45 | 153886 | 48161 | 31.30% | 3.25 |
|---|---|---|---|---|---|---|---|---|
| Traffic | 391979 | 176284 | 44.97% | 3.96 | 366150 | 138097 | 37.72% | 3.62 |
| History& | 154324 | 84683 | 54.87% | 4.16 | 135382 | 60282 | 44.53% | 3.89 |
| Agriculture | 555514 | 266312 | 47.94% | 3.51 | 517525 | 223289 | 43.15% | 3.3 |
| Machinery | 401522 | 203016 | 50.56% | 3.6 | 345031 | 141293 | 40.95% | 3.44 |
| Chemistry | 582921 | 192445 | 33.01% | 3.71 | 519895 | 130998 | 25.20% | 3.74 |
| Law | 123017 | 55986 | 45.51% | 3.82 | 107890 | 38195 | 35.40% | 3.51 |
| Energy | 242753 | 100449 | 41.38% | 3.88 | 222868 | 72442 | 32.50% | 3.63 |
| Astronomy | 517269 | 182781 | 35.34% | 3.43 | 488177 | 147766 | 30.27% | 3.21 |
| Education | 544084 | 247013 | 45.40% | 3.85 | 496295 | 190482 | 38.38% | 3.7 |
| Power | 311758 | 132881 | 42.62% | 3.93 | 266076 | 83308 | 31.31% | 3.83 |
| Art-related | 115208 | 70250 | 60.98% | 4.43 | 105339 | 55755 | 52.93% | 4.1 |
| Material | 73431 | 43548 | 59.30% | 3.95 | 64075 | 31637 | 49.37% | 3.72 |
| Metal | 369741 | 149003 | 40.30% | 3.78 | 327094 | 99341 | 30.37% | 3.64 |
| Electronic | 461533 | 179554 | 38.90% | 3.48 | 429959 | 138569 | 32.23% | 3.3 |
| Politics | 317923 | 170370 | 53.59% | 4.02 | 296393 | 139571 | 47.09% | 3.79 |
| Military | 42421 | 33484 | 78.93% | 4.82 | 37223 | 27337 | 73.44% | 4.04 |
| Medicine* | 1357307 | 193647 | 14.27% | 3.7 | 1357307 | 179792 | 13.25% | 3.42 |
| Bioscience | 233339 | 137138 | 58.77% | 3.22 | 231246 | 132610 | 57.35% | 2.99 |
| Mining | 253300 | 116607 | 46.04% | 3.8 | 229435 | 85991 | 37.48% | 3.52 |
| Culture | 210510 | 120672 | 57.32% | 4.12 | 188244 | 91374 | 48.54% | 3.96 |
| Sport | 116087 | 54913 | 47.30% | 4.41 | 104234 | 39707 | 38.09% | 4.04 |
| Total | 7978805 | 1676176 | 21.01% | 2.84 | 7883539 | 1363444 | 17.29% | 2.68 |
| | *Top-60% tf*idf* | | | | *Top-50% tf*idf* | | | |
| domain | total # | dup # | ratio(%) | *dpk* | total # | dup # | ratio(%) | *dpk* |
| Compute | 330618 | 91361 | 27.63% | 3.46 | 330618 | 86579 | 26.19% | 3.39 |
| Mechanics | 49233 | 17425 | 35.39% | 3.45 | 40677 | 11492 | 28.25% | 3.73 |
| Power | 127891 | 41333 | 32.32% | 4.02 | 125292 | 37402 | 29.85% | 3.94 |
| Publishing | 108383 | 38477 | 35.50% | 4.25 | 96591 | 28518 | 29.52% | 4.38 |
| Forestry | 120953 | 36503 | 30.18% | 3.87 | 100416 | 24788 | 24.69% | 4.2 |
| Industrial | 169796 | 73709 | 43.41% | 4.16 | 163843 | 60740 | 37.07% | 4.18 |
| Math | 105123 | 23238 | 22.11% | 3.19 | 105123 | 22369 | 21.28% | 3.12 |
| Environme | 164196 | 56997 | 34.71% | 4.12 | 162204 | 50640 | 31.22% | 4.11 |
| Fisheries | 46247 | 19162 | 41.43% | 3.95 | 39154 | 15122 | 38.62% | 3.99 |
| Animal | 148401 | 60725 | 40.92% | 3.59 | 125118 | 47054 | 37.61% | 3.71 |
| Architectur | 411955 | 86091 | 20.90% | 4.06 | 411955 | 80319 | 19.50% | 4.01 |
| Aerospace | 139229 | 45166 | 32.44% | 3.91 | 116244 | 30815 | 26.51% | 4.14 |
| LightIndu | 284236 | 87248 | 30.70% | 4.02 | 234687 | 57906 | 24.67% | 4.4 |
| Finance | 519083 | 170558 | 32.86% | 3.71 | 519083 | 160350 | 30.89% | 3.63 |
| Physics | 130950 | 27847 | 21.27% | 3.57 | 130950 | 27139 | 20.72% | 3.49 |
| Traffic | 334847 | 87609 | 26.16% | 3.91 | 280680 | 59434 | 21.18% | 4.2 |
| History& | 113698 | 36864 | 32.42% | 4.16 | 94730 | 29792 | 31.45% | 4.28 |
| Agriculture | 426907 | 132544 | 31.05% | 3.63 | 426907 | 126279 | 29.58% | 3.59 |
| Machinery | 345031 | 103216 | 29.91% | 3.65 | 274464 | 71234 | 25.95% | 3.94 |
| Chemistry | 519895 | 112394 | 21.62% | 3.71 | 519895 | 106713 | 20.53% | 3.65 |
| Law | 98168 | 26158 | 26.65% | 3.57 | 81650 | 18476 | 22.63% | 3.7 |
| Energy | 181397 | 36103 | 19.90% | 4.33 | 177900 | 31577 | 17.75% | 4.18 |
| Astronomy | 418406 | 74062 | 17.70% | 3.72 | 418406 | 69803 | 16.68% | 3.69 |
| Education | 413007 | 128199 | 31.04% | 3.85 | 413007 | 122616 | 29.69% | 3.73 |
| Power | 228137 | 45596 | 19.99% | 4.45 | 225262 | 40841 | 18.13% | 4.38 |
| Art-related | 91793 | 38101 | 41.51% | 4.21 | 72759 | 28137 | 38.67% | 4.39 |
| Material | 54982 | 22371 | 40.69% | 3.9 | 53657 | 20451 | 38.11% | 3.82 |
| Metal | 279246 | 55826 | 19.99% | 4.16 | 279246 | 52324 | 18.74% | 4.12 |
| Electronic | 365397 | 76880 | 21.04% | 3.72 | 365397 | 73182 | 20.03% | 3.65 |
| Politics | 266360 | 103506 | 38.86% | 3.83 | 220783 | 80911 | 36.65% | 3.89 |
| Military | 31705 | 20459 | 64.53% | 3.68 | 27319 | 15218 | 55.70% | 3.54 |

| domain | total # | dup # | ratio(%) | dpk | total # | dup # | ratio(%) | dpk |
|---|---|---|---|---|---|---|---|---|
| Medicine* | 1357307 | 150415 | 11.08% | 3.41 | 1357307 | 137004 | 10.09% | 3.41 |
| Bioscience | 178399 | 84968 | 47.63% | 3.21 | 178399 | 83517 | 46.81% | 3.16 |
| Mining | 209757 | 49421 | 23.56% | 3.97 | 171103 | 32061 | 18.74% | 4.35 |
| Culture | 169734 | 70008 | 41.25% | 3.96 | 141133 | 57728 | 40.90% | 3.95 |
| Sport | 86908 | 22425 | 25.80% | 4.26 | 73779 | 14738 | 19.98% | 4.37 |
| Total | 7494900 | 820490 | 10.95% | 2.87 | 7234018 | 691549 | 9.56% | 2.91 |
| | *Top-40% tf*idf* | | | | *Top-30% tf*idf* | | | |
| domain | total # | dup # | ratio(%) | dpk | total # | dup # | ratio(%) | dpk |
| Compute | 330618 | 85447 | 25.84% | 3.35 | 160492 | 77130 | 48.06% | 3.25 |
| Mechanics | 40677 | 11381 | 27.98% | 3.7 | 21928 | 8787 | 40.07% | 3.6 |
| Power | 125292 | 36903 | 29.45% | 3.9 | 66560 | 28818 | 43.30% | 3.77 |
| Publishing | 80443 | 21413 | 26.62% | 4.43 | 80443 | 20771 | 25.82% | 4.33 |
| Forestry | 100416 | 24569 | 24.47% | 4.1 | 100416 | 23925 | 23.83% | 3.92 |
| Industrial | 129906 | 45490 | 35.02% | 4.31 | 129906 | 44178 | 34.01% | 4.17 |
| Math | 65727 | 22136 | 33.68% | 3.08 | 52563 | 13028 | 24.79% | 2.99 |
| Environme | 125698 | 38074 | 30.29% | 4.28 | 125698 | 36534 | 29.06% | 4.14 |
| Fisheries | 39154 | 14930 | 38.13% | 3.91 | 23036 | 12253 | 53.19% | 3.76 |
| Animal | 125118 | 46644 | 37.28% | 3.62 | 65635 | 39093 | 59.56% | 3.44 |
| Architectur | 411955 | 78870 | 19.15% | 3.96 | 411955 | 76538 | 18.58% | 3.83 |
| Aerospace | 116244 | 30364 | 26.12% | 4.1 | 116244 | 29082 | 25.02% | 4.01 |
| LightIndu | 234687 | 56958 | 24.27% | 4.32 | 234687 | 54453 | 23.20% | 4.17 |
| Finance | 519083 | 157255 | 30.29% | 3.56 | 237796 | 120491 | 50.67% | 3.53 |
| Physics | 130950 | 26836 | 20.49% | 3.45 | 130950 | 25750 | 19.66% | 3.34 |
| Traffic | 280680 | 58669 | 20.90% | 4.15 | 280680 | 57342 | 20.43% | 4.01 |
| History& | 90999 | 26010 | 28.58% | 4.18 | 90999 | 25580 | 28.11% | 4.07 |
| Agriculture | 426907 | 124177 | 29.09% | 3.51 | 204111 | 108968 | 53.39% | 3.39 |
| Machinery | 274464 | 70529 | 25.70% | 3.9 | 274464 | 68873 | 25.09% | 3.76 |
| Chemistry | 519895 | 103942 | 19.99% | 3.61 | 218944 | 68892 | 31.47% | 3.6 |
| Law | 81650 | 18312 | 22.43% | 3.62 | 81650 | 17740 | 21.73% | 3.54 |
| Energy | 177900 | 31246 | 17.56% | 4.13 | 177900 | 30129 | 16.94% | 4.02 |
| Astronomy | 418406 | 68308 | 16.33% | 3.64 | 418406 | 66308 | 15.85% | 3.53 |
| Education | 413007 | 120116 | 29.08% | 3.63 | 413007 | 114874 | 27.81% | 3.54 |
| Power | 225262 | 40380 | 17.93% | 4.33 | 225262 | 39313 | 17.45% | 4.18 |
| Art-related | 69365 | 24796 | 35.75% | 4.3 | 69365 | 24508 | 35.33% | 4.18 |
| Material | 37175 | 20308 | 54.63% | 3.77 | 30025 | 13479 | 44.89% | 3.83 |
| Metal | 279246 | 51588 | 18.47% | 4.07 | 279246 | 49229 | 17.63% | 3.94 |
| Electronic | 365397 | 72171 | 19.75% | 3.6 | 365397 | 70338 | 19.25% | 3.49 |
| Politics | 220783 | 80104 | 36.28% | 3.78 | 220783 | 78053 | 35.35% | 3.67 |
| Military | 21811 | 10929 | 50.11% | 3.49 | 15800 | 7495 | 47.44% | 3.66 |
| Medicine* | 621737 | 111247 | 17.89% | 3.31 | 570727 | 76733 | 13.44% | 3.32 |
| Bioscience | 117523 | 75619 | 64.34% | 3.06 | 98653 | 55237 | 55.99% | 3.07 |
| Mining | 171103 | 31557 | 18.44% | 4.32 | 171103 | 30962 | 18.10% | 4.18 |
| Culture | 141133 | 57227 | 40.55% | 3.83 | 77019 | 56576 | 73.46% | 3.69 |
| Sport | 73779 | 14484 | 19.63% | 4.28 | 73779 | 13955 | 18.91% | 4.18 |
| Total | 6357230 | 662029 | 10.41% | 2.88 | 5224408 | 594194 | 11.37% | 2.84 |
| | *Top-20% tf*idf* | | | | *Top-10% tf*idf* | | | |
| domain | total # | dup # | ratio(%) | dpk | total # | dup # | ratio | dpk |
| Compute | 107083 | 41106 | 38.39% | 3.35 | 52679 | 23121 | 43.89% | 2.94 |
| Mechanics | 14378 | 4455 | 30.98% | 3.9 | 13109 | 2782 | 21.22% | 3.54 |
| Power | 42362 | 12931 | 30.52% | 3.98 | 42362 | 10213 | 24.11% | 3.46 |
| Publishing | 80443 | 18323 | 22.78% | 4.06 | 18899 | 5757 | 30.46% | 3.55 |
| Forestry | 100416 | 20812 | 20.73% | 3.71 | 20338 | 8676 | 42.66% | 3.35 |
| Industrial | 56644 | 25755 | 45.47% | 4.15 | 32172 | 9659 | 30.02% | 3.96 |
| Math | 41059 | 7187 | 17.50% | 3.07 | 16902 | 4213 | 24.93% | 2.72 |
| Environme | 55104 | 32477 | 58.94% | 3.9 | 28190 | 12582 | 44.63% | 3.61 |
| Fisheries | 15467 | 5983 | 38.68% | 3.9 | 7703 | 3498 | 45.41% | 3.61 |

| | | | | | | | | |
|---|---|---|---|---|---|---|---|---|
| Animal | 44513 | 20591 | 46.26% | 3.49 | 21635 | 11337 | 52.40% | 3.2 |
| Architectur | 132244 | 62937 | 47.59% | 3.58 | 82356 | 26109 | 31.70% | 3.25 |
| Aerospace | 44489 | 18194 | 40.90% | 3.91 | 28764 | 7763 | 26.99% | 3.71 |
| LightIndu | 87820 | 44482 | 50.65% | 3.95 | 54445 | 17972 | 33.01% | 3.57 |
| Finance | 175871 | 77142 | 43.86% | 3.43 | 80437 | 34048 | 42.33% | 3.01 |
| Physics | 46272 | 13871 | 29.98% | 3.16 | 37124 | 8311 | 22.39% | 2.9 |
| Traffic | 280680 | 52391 | 18.67% | 3.72 | 54608 | 18863 | 34.54% | 3.44 |
| History& | 90999 | 22250 | 24.45% | 3.81 | 19364 | 8568 | 44.25% | 3.38 |
| Agriculture | 135481 | 56395 | 41.63% | 3.44 | 68732 | 31123 | 45.28% | 3.1 |
| Machinery | 98038 | 53510 | 54.58% | 3.63 | 48717 | 20694 | 42.48% | 3.5 |
| Chemistry | 186654 | 50161 | 26.87% | 3.54 | 73432 | 26187 | 35.66% | 3.22 |
| Law | 81650 | 15430 | 18.90% | 3.34 | 15890 | 6279 | 39.52% | 2.87 |
| Energy | 177900 | 27563 | 15.49% | 3.79 | 35818 | 11903 | 33.23% | 3.29 |
| Astronomy | 124643 | 37559 | 30.13% | 3.41 | 62318 | 22069 | 35.41% | 3.1 |
| Education | 136691 | 68097 | 49.82% | 3.45 | 68097 | 29567 | 43.42% | 3.01 |
| Power | 225262 | 34860 | 15.48% | 3.92 | 44511 | 12622 | 28.36% | 3.65 |
| Art-related | 29950 | 18012 | 60.14% | 3.97 | 14451 | 7378 | 51.06% | 3.51 |
| Material | 19590 | 6944 | 35.45% | 4.23 | 19590 | 6032 | 30.79% | 3.52 |
| Metal | 90972 | 41917 | 46.08% | 3.76 | 56900 | 16731 | 29.40% | 3.48 |
| Electronic | 117859 | 46219 | 39.22% | 3.35 | 57358 | 23912 | 41.69% | 3 |
| Politics | 79945 | 52325 | 65.45% | 3.51 | 39743 | 24345 | 61.26% | 3.03 |
| Military | 15767 | 6912 | 43.84% | 3.4 | 6397 | 3002 | 46.93% | 3.03 |
| Medicine* | 309239 | 50200 | 16.23% | 3.23 | 155681 | 21741 | 13.97% | 2.9 |
| Bioscience | 68539 | 31521 | 45.99% | 3.26 | 29217 | 19270 | 65.95% | 3.01 |
| Mining | 171103 | 28667 | 16.75% | 3.93 | 31666 | 12476 | 39.40% | 3.6 |
| Culture | 53358 | 35120 | 65.82% | 3.61 | 25639 | 17156 | 66.91% | 3.11 |
| Sport | 73779 | 12220 | 16.56% | 3.94 | 16694 | 4220 | 25.28% | 3.4 |
| Total | 2870181 | 412436 | 14.37% | 2.8 | 1153819 | 202060 | 17.51% | 2.62 |

Table 20: Detailed domain number distribution for cross-domain duplicated keywords (number) in TechKG

| | Top-100% tf*idf | | | | | Top-90% tf*idf | | | | |
|---|---|---|---|---|---|---|---|---|---|---|
| Domain | [1,5) | [5,10) | [10,20) | [20,30) | [30,36) | [1,5) | [5,10) | [10,20) | [20,30) | [30,36) |
| Computer | 186965 | 53625 | 41309 | 16040 | 3042 | 201254 | 36226 | 10940 | 808 | 4 |
| Mechanis | 24266 | 8820 | 7283 | 3534 | 979 | 27409 | 8245 | 2283 | 190 | 0 |
| Power | 64179 | 26711 | 25718 | 12110 | 2797 | 73532 | 26719 | 9619 | 801 | 4 |
| Publising | 46729 | 31100 | 29587 | 12979 | 2948 | 60402 | 34533 | 9643 | 628 | 4 |
| Forestry | 58850 | 26166 | 25239 | 12966 | 2969 | 68857 | 28781 | 8605 | 821 | 4 |
| Industrial | 93162 | 49005 | 44678 | 16837 | 3052 | 112137 | 49408 | 16343 | 1052 | 4 |
| Math | 46020 | 13698 | 12174 | 6464 | 2121 | 51431 | 9724 | 2806 | 271 | 0 |
| Environe | 92871 | 42404 | 37411 | 15556 | 3036 | 109188 | 42687 | 12923 | 926 | 4 |
| Fisheries | 23981 | 11379 | 10732 | 7045 | 2468 | 27924 | 14544 | 5046 | 430 | 4 |
| Animal | 80131 | 28010 | 22576 | 11333 | 2896 | 89877 | 26986 | 7067 | 555 | 4 |
| Architecr | 192828 | 70557 | 57793 | 18657 | 3074 | 211252 | 51890 | 18812 | 1106 | 4 |
| Aerospace | 65786 | 30169 | 28935 | 13399 | 2926 | 78316 | 31262 | 10265 | 847 | 4 |
| LightIndu | 132099 | 61835 | 53342 | 18236 | 3067 | 152784 | 51711 | 18857 | 1096 | 4 |
| Finance | 279397 | 101068 | 63473 | 18388 | 3075 | 294830 | 78229 | 19165 | 934 | 4 |
| Physics | 62223 | 19698 | 16196 | 7274 | 1964 | 68644 | 14412 | 4160 | 347 | 0 |
| Traffic | 143772 | 60503 | 51692 | 18017 | 3066 | 162858 | 47698 | 17147 | 1078 | 4 |
| History& | 53643 | 29899 | 24076 | 9972 | 2793 | 64692 | 30231 | 7284 | 305 | 4 |
| Agricue | 249082 | 71561 | 54018 | 18373 | 3074 | 261533 | 50740 | 16382 | 1073 | 4 |
| Machiner | 171816 | 57746 | 43250 | 15906 | 3048 | 187147 | 42503 | 13105 | 906 | 4 |
| Chemistry | 196673 | 63006 | 47461 | 17114 | 3065 | 202152 | 44169 | 14208 | 979 | 4 |
| Law | 41814 | 18400 | 17311 | 8871 | 2613 | 49505 | 19384 | 5267 | 352 | 4 |
| Energy | 80125 | 31626 | 32619 | 14712 | 2988 | 92417 | 29639 | 11191 | 964 | 4 |
| Astronoy | 172653 | 52307 | 41621 | 16432 | 3058 | 181904 | 34059 | 11158 | 906 | 4 |

| Domain | [1,5) | [5,10) | [10,20) | [20,30) | [30,36) | [1,5) | [5,10) | [10,20) | [20,30) | [30,36) |
|---|---|---|---|---|---|---|---|---|---|---|
| Education | 219439 | 88219 | 59976 | 18463 | 3076 | 240547 | 66863 | 17246 | 955 | 4 |
| Power | 101378 | 43924 | 40005 | 16295 | 3038 | 117158 | 35801 | 13058 | 1039 | 4 |
| Art- | 39538 | 26654 | 22101 | 9391 | 2706 | 50313 | 29400 | 6959 | 288 | 4 |
| Material | 32894 | 12718 | 10319 | 4878 | 1555 | 36766 | 12941 | 3768 | 249 | 1 |
| Metal | 126442 | 48233 | 41187 | 16217 | 3055 | 140198 | 36777 | 12740 | 985 | 4 |
| Electronic | 170228 | 53877 | 42948 | 16699 | 3050 | 186580 | 37361 | 11772 | 935 | 4 |
| Politics | 128661 | 61094 | 42208 | 14438 | 3027 | 146014 | 51200 | 12424 | 562 | 4 |
| Military | 16503 | 9423 | 10789 | 6638 | 2342 | 20057 | 14777 | 5316 | 350 | 4 |
| Medicine | 235742 | 66734 | 50723 | 17867 | 3074 | 197769 | 40031 | 12813 | 934 | 4 |
| Bioscie | 149574 | 27733 | 17390 | 7759 | 2456 | 152119 | 20227 | 4566 | 248 | 0 |
| Mining | 94202 | 35394 | 33282 | 14191 | 2931 | 106863 | 30312 | 11194 | 925 | 3 |
| Culture | 80185 | 42999 | 30772 | 11507 | 2910 | 96446 | 39512 | 9069 | 342 | 4 |
| Sport | 32544 | 19804 | 20979 | 10316 | 2869 | 41189 | 23612 | 6999 | 431 | 4 |
| Total | 1508200 | 201467 | 85567 | 19461 | 3076 | 1604384 | 174265 | 28770 | 1113 | 4 |
|  |  |  |  |  |  |  |  |  |  |  |
| *Top-80% tf*idf* |  |  |  |  | | *Top-70% tf*idf* |  |  |  |  |
| Domain | [1,5) | [5,10) | [10,20) | [20,30) | [30,36) | [1,5) | [5,10) | [10,20) | [20,30) | [30,36) |
| Computer | 178551 | 18490 | 5311 | 290 | 1 | 144694 | 13198 | 3471 | 162 | 0 |
| Mechanis | 25445 | 3456 | 1010 | 73 | 0 | 18657 | 2276 | 632 | 50 | 0 |
| Power | 73520 | 12170 | 4750 | 301 | 1 | 51769 | 7367 | 2877 | 164 | 0 |
| Publising | 63339 | 16731 | 3983 | 231 | 1 | 46943 | 11947 | 2251 | 114 | 0 |
| Forestry | 68862 | 11681 | 3626 | 302 | 1 | 53059 | 7655 | 2017 | 147 | 0 |
| Industrial | 119043 | 25138 | 8195 | 395 | 1 | 90643 | 16247 | 5113 | 222 | 0 |
| Math | 42315 | 3983 | 1083 | 103 | 0 | 32918 | 2686 | 716 | 62 | 0 |
| Environe | 111366 | 19714 | 6020 | 362 | 1 | 83606 | 13507 | 3639 | 197 | 0 |
| Fisheries | 30416 | 7402 | 1748 | 119 | 1 | 24240 | 4905 | 943 | 57 | 0 |
| Animal | 83788 | 13368 | 2949 | 195 | 1 | 70944 | 9922 | 1801 | 108 | 0 |
| Architecr | 174971 | 25065 | 8875 | 409 | 1 | 138368 | 18672 | 5819 | 226 | 0 |
| Aerospace | 76395 | 13512 | 4514 | 307 | 1 | 54559 | 8682 | 2890 | 163 | 0 |
| LightIndu | 140034 | 26722 | 8871 | 408 | 1 | 102813 | 18263 | 5597 | 223 | 0 |
| Finance | 278183 | 45069 | 8578 | 347 | 1 | 228419 | 33115 | 5247 | 193 | 0 |
| Physics | 55639 | 6368 | 1819 | 122 | 0 | 42653 | 4277 | 1154 | 77 | 0 |
| Traffic | 143385 | 24236 | 8254 | 408 | 1 | 115716 | 16821 | 5337 | 223 | 0 |
| History& | 66317 | 15232 | 3029 | 104 | 1 | 47757 | 10821 | 1657 | 47 | 0 |
| Agricue | 229104 | 29071 | 7730 | 406 | 1 | 196048 | 21981 | 5034 | 226 | 0 |
| Machiner | 173589 | 22321 | 6770 | 335 | 1 | 121333 | 15311 | 4461 | 188 | 0 |
| Chemistry | 162082 | 22878 | 7127 | 357 | 1 | 109643 | 16445 | 4708 | 202 | 0 |
| Law | 46273 | 7780 | 1813 | 119 | 1 | 32487 | 4788 | 862 | 58 | 0 |
| Energy | 84042 | 10704 | 5343 | 359 | 1 | 61357 | 7321 | 3562 | 202 | 0 |
| Astronoy | 159056 | 17989 | 5389 | 346 | 1 | 130638 | 13404 | 3525 | 199 | 0 |
| Education | 201937 | 37268 | 7451 | 356 | 1 | 158275 | 27549 | 4467 | 191 | 0 |
| Power | 109614 | 16374 | 6500 | 392 | 1 | 68112 | 10768 | 4216 | 212 | 0 |
| Art-relatd | 52387 | 14955 | 2819 | 88 | 1 | 42654 | 11469 | 1592 | 40 | 0 |
| Material | 35674 | 6042 | 1741 | 91 | 0 | 26533 | 3976 | 1074 | 54 | 0 |
| Metal | 125018 | 17215 | 6397 | 372 | 1 | 83499 | 11493 | 4139 | 210 | 0 |
| Electronic | 155845 | 17881 | 5491 | 336 | 1 | 121852 | 12853 | 3676 | 188 | 0 |
| Politics | 135302 | 29686 | 5176 | 205 | 1 | 113926 | 22562 | 2975 | 108 | 0 |
| Military | 23253 | 8484 | 1633 | 113 | 1 | 21793 | 4725 | 761 | 58 | 0 |
| Medicine | 163118 | 24004 | 6159 | 365 | 1 | 155809 | 19548 | 4232 | 203 | 0 |
| Bioscie | 122389 | 12387 | 2264 | 98 | 0 | 121259 | 9798 | 1495 | 58 | 0 |
| Mining | 98533 | 12142 | 5575 | 356 | 1 | 73768 | 8295 | 3726 | 202 | 0 |
| Culture | 94265 | 22559 | 3730 | 117 | 1 | 72183 | 17062 | 2076 | 53 | 0 |
| Sport | 41589 | 10381 | 2786 | 156 | 1 | 30878 | 7209 | 1547 | 73 | 0 |
| Total | 1574965 | 87502 | 13294 | 414 | 1 | 1292004 | 62879 | 8333 | 228 | 0 |
| *Top-60% tf*idf* |  |  |  |  | | *Top-50% tf*idf* |  |  |  |  |
| Domain | [1,5) | [5,10) | [10,20) | [20,30) | [30,36) | [1,5) | [5,10) | [10,20) | [20,30) | [30,36) |
| Computer | 79805 | 9156 | 2317 | 83 | 0 | 76488 | 8019 | 2007 | 65 | 0 |

| Domain | | | | | | | | | | |
|---|---|---|---|---|---|---|---|---|---|
| Mechanis | 15188 | 1721 | 483 | 33 | 0 | 9798 | 1291 | 374 | 29 | 0 |
| Power | 33787 | 5401 | 2058 | 87 | 0 | 30880 | 4693 | 1758 | 71 | 0 |
| Publising | 28979 | 8075 | 1362 | 61 | 0 | 21149 | 6322 | 1012 | 35 | 0 |
| Forestry | 29909 | 5295 | 1223 | 76 | 0 | 19655 | 4215 | 870 | 48 | 0 |
| Industrial | 58174 | 11821 | 3591 | 123 | 0 | 47958 | 9722 | 2972 | 88 | 0 |
| Math | 21008 | 1691 | 505 | 34 | 0 | 20421 | 1491 | 425 | 32 | 0 |
| Environe | 44751 | 9801 | 2344 | 101 | 0 | 39948 | 8673 | 1945 | 74 | 0 |
| Fisheries | 15329 | 3279 | 531 | 23 | 0 | 12074 | 2638 | 398 | 12 | 0 |
| Animal | 51843 | 7630 | 1199 | 53 | 0 | 39883 | 6249 | 890 | 32 | 0 |
| Architecr | 68930 | 13068 | 3967 | 126 | 0 | 64933 | 11915 | 3381 | 90 | 0 |
| Aerospace | 37008 | 6027 | 2045 | 86 | 0 | 24803 | 4379 | 1572 | 61 | 0 |
| LightIndu | 69236 | 13980 | 3906 | 126 | 0 | 44072 | 10734 | 3010 | 90 | 0 |
| Finance | 143460 | 23667 | 3331 | 100 | 0 | 137240 | 20365 | 2677 | 68 | 0 |
| Physics | 24143 | 2882 | 773 | 49 | 0 | 23797 | 2628 | 667 | 47 | 0 |
| Traffic | 71213 | 12521 | 3754 | 121 | 0 | 46868 | 9485 | 2995 | 86 | 0 |
| History& | 28320 | 7530 | 986 | 28 | 0 | 22632 | 6429 | 717 | 14 | 0 |
| Agricue | 113301 | 15837 | 3280 | 126 | 0 | 109111 | 14299 | 2779 | 90 | 0 |
| Machiner | 87234 | 12539 | 3334 | 109 | 0 | 58550 | 9905 | 2701 | 78 | 0 |
| Chemistry | 94955 | 13800 | 3522 | 117 | 0 | 91084 | 12508 | 3035 | 86 | 0 |
| Law | 22323 | 3286 | 520 | 29 | 0 | 15700 | 2397 | 364 | 15 | 0 |
| Energy | 28257 | 5274 | 2460 | 112 | 0 | 25178 | 4305 | 2010 | 84 | 0 |
| Astronoy | 62282 | 9335 | 2334 | 111 | 0 | 59190 | 8559 | 1970 | 84 | 0 |
| Education | 105898 | 19411 | 2786 | 104 | 0 | 103757 | 16529 | 2258 | 72 | 0 |
| Power | 34834 | 7709 | 2939 | 114 | 0 | 31581 | 6636 | 2539 | 85 | 0 |
| Art- | 28729 | 8414 | 932 | 26 | 0 | 20813 | 6642 | 668 | 14 | 0 |
| Material | 18579 | 2998 | 764 | 30 | 0 | 17190 | 2600 | 636 | 25 | 0 |
| Metal | 44478 | 8357 | 2875 | 116 | 0 | 42049 | 7648 | 2543 | 84 | 0 |
| Electronic | 65234 | 9041 | 2507 | 98 | 0 | 62923 | 8005 | 2178 | 76 | 0 |
| Politics | 85095 | 16523 | 1824 | 64 | 0 | 66576 | 12962 | 1337 | 36 | 0 |
| Military | 17081 | 2950 | 393 | 35 | 0 | 12959 | 2007 | 234 | 18 | 0 |
| Medicine | 131366 | 15874 | 3059 | 116 | 0 | 120185 | 14188 | 2547 | 84 | 0 |
| Bioscie | 76524 | 7439 | 979 | 26 | 0 | 75875 | 6812 | 810 | 20 | 0 |
| Mining | 40310 | 6269 | 2732 | 110 | 0 | 25213 | 4657 | 2110 | 81 | 0 |
| Culture | 55891 | 12826 | 1259 | 32 | 0 | 46514 | 10290 | 910 | 14 | 0 |
| Sport | 16779 | 4683 | 918 | 45 | 0 | 10939 | 3157 | 616 | 26 | 0 |
| Total | 768736 | 45996 | 5632 | 126 | 0 | 648186 | 38705 | 4568 | 90 | 0 |
| | | | | | | | | | | |
| | *Top-40% tf\*idf* | | | | | *Top-30% tf\*idf* | | | | |
| Domain | [1,5) | [5,10) | [10,20) | [20,30) | [30,36) | [1,5) | [5,10) | [10,20) | [20,30) | [30,36) |
| Computer | 75925 | 7586 | 1876 | 60 | 0 | 69126 | 6404 | 1553 | 47 | 0 |
| Mechanis | 9745 | 1251 | 359 | 26 | 0 | 7556 | 954 | 255 | 22 | 0 |
| Power | 30613 | 4551 | 1675 | 64 | 0 | 24130 | 3349 | 1288 | 51 | 0 |
| Publising | 15831 | 4789 | 762 | 31 | 0 | 15614 | 4447 | 683 | 27 | 0 |
| Forestry | 19729 | 4011 | 787 | 42 | 0 | 19627 | 3578 | 685 | 35 | 0 |
| Industrial | 35478 | 7478 | 2456 | 78 | 0 | 35010 | 6964 | 2140 | 64 | 0 |
| Math | 20279 | 1435 | 390 | 32 | 0 | 11981 | 779 | 242 | 26 | 0 |
| Environe | 29457 | 7026 | 1528 | 63 | 0 | 28776 | 6382 | 1323 | 53 | 0 |
| Fisheries | 12067 | 2496 | 357 | 10 | 0 | 10020 | 1961 | 263 | 9 | 0 |
| Animal | 39947 | 5879 | 791 | 27 | 0 | 34036 | 4456 | 581 | 20 | 0 |
| Architecr | 64296 | 11385 | 3109 | 80 | 0 | 63318 | 10457 | 2697 | 66 | 0 |
| Aerospace | 24561 | 4240 | 1509 | 54 | 0 | 23800 | 3897 | 1341 | 44 | 0 |
| LightIndu | 43877 | 10242 | 2759 | 80 | 0 | 42737 | 9238 | 2412 | 66 | 0 |
| Finance | 136361 | 18477 | 2356 | 61 | 0 | 104157 | 14501 | 1785 | 48 | 0 |
| Physics | 23648 | 2509 | 637 | 42 | 0 | 22919 | 2234 | 562 | 35 | 0 |
| Traffic | 46640 | 9150 | 2801 | 78 | 0 | 46332 | 8535 | 2411 | 64 | 0 |
| History& | 20045 | 5407 | 546 | 12 | 0 | 20150 | 4931 | 487 | 12 | 0 |
| Agricue | 108219 | 13356 | 2522 | 80 | 0 | 96095 | 10768 | 2039 | 66 | 0 |
| Machiner | 58368 | 9549 | 2542 | 70 | 0 | 57914 | 8710 | 2193 | 56 | 0 |

| Domain | [1,5) | [5,10) | [10,20) | [20,30) | [30,36) | [1,5) | [5,10) | [10,20) | [20,30) | [30,36) |
|---|---|---|---|---|---|---|---|---|---|---|
| Chemistry | 89130 | 11920 | 2815 | 77 | 0 | 58814 | 8057 | 1963 | 58 | 0 |
| Law | 15794 | 2191 | 312 | 15 | 0 | 15417 | 2029 | 279 | 15 | 0 |
| Energy | 25067 | 4208 | 1896 | 75 | 0 | 24445 | 3919 | 1704 | 61 | 0 |
| Astronoy | 58331 | 8106 | 1796 | 75 | 0 | 57316 | 7374 | 1556 | 62 | 0 |
| Education | 103116 | 14946 | 1990 | 64 | 0 | 99890 | 13209 | 1722 | 53 | 0 |
| Power | 31466 | 6444 | 2393 | 77 | 0 | 31093 | 6049 | 2107 | 64 | 0 |
| Art- | 18573 | 5691 | 522 | 10 | 0 | 18909 | 5134 | 455 | 10 | 0 |
| Material | 17211 | 2479 | 596 | 22 | 0 | 11191 | 1831 | 440 | 17 | 0 |
| Metal | 41713 | 7400 | 2400 | 75 | 0 | 40324 | 6746 | 2097 | 62 | 0 |
| Electronic | 62425 | 7611 | 2067 | 68 | 0 | 61557 | 6919 | 1808 | 54 | 0 |
| Politics | 67276 | 11661 | 1135 | 32 | 0 | 66580 | 10466 | 978 | 29 | 0 |
| Military | 9356 | 1422 | 138 | 13 | 0 | 6265 | 1116 | 105 | 9 | 0 |
| Medicine | 98540 | 10722 | 1913 | 72 | 0 | 67893 | 7445 | 1339 | 56 | 0 |
| Bioscie | 69108 | 5859 | 639 | 13 | 0 | 50304 | 4462 | 461 | 10 | 0 |
| Mining | 24942 | 4537 | 2006 | 72 | 0 | 24802 | 4314 | 1787 | 59 | 0 |
| Culture | 47127 | 9330 | 758 | 12 | 0 | 47713 | 8221 | 630 | 12 | 0 |
| Sport | 10911 | 3018 | 531 | 24 | 0 | 10672 | 2782 | 479 | 22 | 0 |
| Total | 622623 | 35235 | 4091 | 80 | 0 | 560547 | 30150 | 3431 | 66 | 0 |

| | Top-20% tf*idf | | | | | Top-10% tf*idf | | | | |
|---|---|---|---|---|---|---|---|---|---|---|
| Domain | [1,5) | [5,10) | [10,20) | [20,30) | [30,36) | [1,5) | [5,10) | [10,20) | [20,30) | [30,36) |
| Computer | 36424 | 3762 | 901 | 19 | 0 | 21463 | 1372 | 278 | 8 | 0 |
| Mechanis | 3664 | 621 | 158 | 12 | 0 | 2425 | 278 | 74 | 5 | 0 |
| Power | 10496 | 1763 | 650 | 22 | 0 | 8876 | 1014 | 314 | 9 | 0 |
| Publising | 14417 | 3419 | 475 | 12 | 0 | 4894 | 786 | 76 | 1 | 0 |
| Forestry | 17561 | 2754 | 480 | 17 | 0 | 7693 | 897 | 82 | 4 | 0 |
| Industrial | 20221 | 4171 | 1334 | 29 | 0 | 7839 | 1406 | 405 | 9 | 0 |
| Math | 6536 | 491 | 147 | 13 | 0 | 3975 | 181 | 52 | 5 | 0 |
| Environe | 26552 | 4987 | 912 | 26 | 0 | 10725 | 1649 | 201 | 7 | 0 |
| Fisheries | 4784 | 1076 | 119 | 4 | 0 | 2909 | 553 | 34 | 2 | 0 |
| Animal | 17813 | 2499 | 271 | 8 | 0 | 10217 | 1062 | 57 | 1 | 0 |
| Architecr | 53639 | 7506 | 1761 | 31 | 0 | 23276 | 2333 | 491 | 9 | 0 |
| Aerospace | 14920 | 2428 | 824 | 22 | 0 | 6562 | 936 | 258 | 7 | 0 |
| LightIndu | 35884 | 6942 | 1625 | 31 | 0 | 15382 | 2154 | 427 | 9 | 0 |
| Finance | 67528 | 8587 | 1007 | 20 | 0 | 31595 | 2269 | 180 | 4 | 0 |
| Physics | 12507 | 1079 | 269 | 16 | 0 | 7761 | 441 | 102 | 7 | 0 |
| Traffic | 43929 | 6815 | 1616 | 31 | 0 | 16376 | 2019 | 459 | 9 | 0 |
| History& | 18194 | 3729 | 321 | 6 | 0 | 7553 | 970 | 45 | 0 | 0 |
| Agricue | 49578 | 5796 | 990 | 31 | 0 | 28649 | 2224 | 241 | 9 | 0 |
| Machiner | 45609 | 6416 | 1457 | 28 | 0 | 18007 | 2222 | 456 | 9 | 0 |
| Chemistry | 43218 | 5652 | 1263 | 28 | 0 | 23545 | 2230 | 403 | 9 | 0 |
| Law | 13703 | 1523 | 197 | 7 | 0 | 5951 | 305 | 23 | 0 | 0 |
| Energy | 22832 | 3428 | 1273 | 30 | 0 | 10521 | 1030 | 343 | 9 | 0 |
| Astronoy | 32857 | 3885 | 788 | 29 | 0 | 20116 | 1710 | 235 | 8 | 0 |
| Education | 59670 | 7501 | 903 | 23 | 0 | 27506 | 1887 | 171 | 3 | 0 |
| Power | 28327 | 5032 | 1470 | 31 | 0 | 10648 | 1532 | 433 | 9 | 0 |
| Art- | 14347 | 3384 | 278 | 3 | 0 | 6396 | 939 | 43 | 0 | 0 |
| Material | 5450 | 1205 | 281 | 8 | 0 | 5279 | 638 | 110 | 5 | 0 |
| Metal | 35017 | 5391 | 1478 | 31 | 0 | 14518 | 1761 | 443 | 9 | 0 |
| Electronic | 40886 | 4230 | 1080 | 23 | 0 | 22102 | 1453 | 349 | 8 | 0 |
| Politics | 45394 | 6343 | 575 | 13 | 0 | 22615 | 1631 | 97 | 2 | 0 |
| Military | 6013 | 821 | 72 | 6 | 0 | 2765 | 227 | 10 | 0 | 0 |
| Medicine | 44932 | 4463 | 781 | 24 | 0 | 20291 | 1311 | 133 | 6 | 0 |
| Bioscie | 28305 | 2947 | 266 | 3 | 0 | 17897 | 1310 | 62 | 1 | 0 |
| Mining | 23563 | 3752 | 1324 | 28 | 0 | 10720 | 1329 | 418 | 9 | 0 |
| Culture | 29856 | 4932 | 330 | 2 | 0 | 15754 | 1362 | 40 | 0 | 0 |
| Sport | 9663 | 2207 | 341 | 9 | 0 | 3660 | 508 | 50 | 2 | 0 |
| Total | 390134 | 20104 | 2167 | 31 | 0 | 194842 | 6612 | 597 | 9 | 0 |

Table 21: Detailed triplet statistics under different tf*idf setting in TechKG

| | Top-100% tf*idf | | | | | | | Top-90% tf*idf | | | | | | |
| | 1-1 | 1-n | | m-1 | | m-n | | 1-1 | 1-n | | m-1 | | m-n | |
| Domain | % | % | ñ | % | | % | ū(m/n) | % | % | ñ | % | | % | ū(m/n) |
|---|---|---|---|---|---|---|---|---|---|---|---|---|---|---|
| Computer | 15.55 | 0 | \ | 18.78 | 11.05 | 65.68 | 1.19 | 16.22 | 0 | \ | 18.38 | 10.1 | 65.39 | 1.21 |
| Mechanis | 16.85 | 0 | \ | 15.3 | 7.34 | 67.85 | 1.05 | 17.3 | 0 | \ | 15.75 | 7.03 | 66.95 | 1.06 |
| Power | 14.05 | 0 | \ | 16.51 | 9.32 | 69.44 | 1.12 | 14.5 | 0 | \ | 16.43 | 8.7 | 69.06 | 1.13 |
| Publising | 13.48 | 0 | \ | 30 | 26.7 | 56.52 | 1.2 | 14.55 | 0 | \ | 28.6 | 23.27 | 56.85 | 1.23 |
| Forestry | 14.07 | 0 | \ | 18.95 | 12.47 | 66.97 | 1.2 | 14.51 | 0 | \ | 19.45 | 12.07 | 66.04 | 1.22 |
| Industrial | 14.17 | 0 | \ | 21.76 | 11.73 | 64.07 | 1.09 | 14.72 | 0 | \ | 22.13 | 11.09 | 63.15 | 1.1 |
| Math | 21.03 | 0 | \ | 18.71 | 9.06 | 60.26 | 1.11 | 21.76 | 0 | \ | 19.31 | 8.7 | 58.93 | 1.12 |
| Environe | 13.77 | 0 | \ | 18.34 | 10.08 | 67.88 | 1.23 | 14.21 | 0 | \ | 18.76 | 9.79 | 67.04 | 1.25 |
| Fisheries | 11.94 | 0 | \ | 18.5 | 16 | 69.56 | 1.08 | 12.26 | 0 | \ | 18.88 | 15.55 | 68.86 | 1.09 |
| Animal | 11.02 | 0 | \ | 21.18 | 20.8 | 67.8 | 1.16 | 11.37 | 0 | \ | 21.2 | 19.87 | 67.43 | 1.17 |
| Architecr | 15.4 | 0 | \ | 20.45 | 13.38 | 64.15 | 1.26 | 16.12 | 0 | \ | 19.29 | 11.78 | 64.59 | 1.28 |
| Aerospace | 15.51 | 0 | \ | 19.78 | 10.41 | 64.71 | 1.2 | 16.39 | 0 | \ | 18.39 | 8.87 | 65.21 | 1.22 |
| LightIndu | 14.22 | 0 | \ | 20.17 | 14.59 | 65.61 | 1.23 | 14.65 | 0 | \ | 20.31 | 13.93 | 65.04 | 1.25 |
| Finance | 13.7 | 0 | \ | 30.27 | 35.02 | 56.03 | 1.26 | 14.43 | 0 | \ | 28.09 | 30.49 | 57.48 | 1.28 |
| Physics | 13.21 | 0 | \ | 13.3 | 7.9 | 73.5 | 1.13 | 13.4 | 0 | \ | 13.51 | 7.61 | 73.08 | 1.14 |
| Traffic | 15.75 | 0 | \ | 21.95 | 13.55 | 62.3 | 1.25 | 16.22 | 0 | \ | 22.54 | 13.17 | 61.24 | 1.27 |
| History& | 10.81 | 0.4 | 2.2 | 32.12 | 18.34 | 56.67 | 0.99 | 11.68 | 0.4 | 2.2 | 31.65 | 16.36 | 56.23 | 0.99 |
| Agricue | 7.17 | 5.1 | 1.5 | 16.97 | 15.71 | 70.73 | 1.23 | 12.72 | 0 | \ | 15.78 | 13.85 | 71.51 | 1.26 |
| Machiner | 15.29 | 0 | \ | 18.12 | 9.89 | 66.6 | 1.24 | 15.7 | 0 | \ | 18.54 | 9.58 | 65.76 | 1.27 |
| Chemistry | 14.38 | 0 | \ | 15.48 | 10.98 | 70.14 | 1.27 | 14.71 | 0 | \ | 15.51 | 10.53 | 69.79 | 1.3 |
| Law | 14.65 | 0.1 | 1.8 | 34.4 | 18.31 | 50.83 | 1.02 | 15.77 | 0.1 | 1.8 | 34.74 | 16.78 | 49.36 | 1.02 |
| Energy | 13.44 | 0 | \ | 14.61 | 9.82 | 71.95 | 1.21 | 13.81 | 0 | \ | 14.37 | 9.15 | 71.82 | 1.23 |
| Astronoy | 14.6 | 0 | \ | 15.44 | 11.84 | 69.95 | 1.27 | 14.97 | 0 | \ | 15.55 | 11.29 | 69.48 | 1.3 |
| Education | 13.03 | 0 | \ | 29.62 | 29.99 | 57.35 | 1.06 | 13.46 | 0 | \ | 28.6 | 27.61 | 57.95 | 1.06 |
| Power | 15.29 | 0 | \ | 18.8 | 11.26 | 65.91 | 1.21 | 15.76 | 0 | \ | 19.31 | 10.92 | 64.93 | 1.23 |
| Art- | 8.79 | 0.1 | 2.4 | 31.78 | 28.52 | 59.34 | 1.04 | 9.33 | 0.1 | 2.4 | 31.37 | 26.22 | 59.2 | 1.05 |
| Material | 13.28 | 0 | \ | 13.6 | 6.95 | 73.12 | 1.08 | 13.47 | 0 | \ | 13.79 | 6.7 | 72.74 | 1.08 |
| Metal | 15.05 | 0 | \ | 17.31 | 10.85 | 67.64 | 1.24 | 15.48 | 0 | \ | 17.45 | 10.34 | 67.07 | 1.26 |
| Electronic | 15.13 | 0 | \ | 19.35 | 12.32 | 65.53 | 1.2 | 15.84 | 0 | \ | 18.42 | 10.94 | 65.74 | 1.21 |
| Politics | 11.62 | 0.1 | 1.6 | 36.31 | 32.26 | 52 | 1.05 | 12.48 | 0.1 | 1.6 | 34.47 | 28.19 | 52.97 | 1.06 |
| Military | 11.84 | 0 | \ | 27.75 | 31.07 | 60.42 | 1.05 | 12.43 | 0 | \ | 28.86 | 30.14 | 58.71 | 1.05 |
| Medicine | 11.85 | 0 | \ | 15.33 | 17.49 | 72.83 | 1.44 | 12.08 | 0 | \ | 14.53 | 16.09 | 73.39 | 1.47 |
| Bioscie | 12.51 | 0 | \ | 12.63 | 8.03 | 74.86 | 1.11 | 12.78 | 0 | \ | 12.41 | 7.51 | 74.81 | 1.12 |
| Mining | 16.5 | 0 | \ | 19.12 | 9.83 | 64.38 | 1.19 | 17.2 | 0 | \ | 18.7 | 8.92 | 64.1 | 1.21 |
| Culture | 7.63 | 0.1 | 1.6 | 29.01 | 26.37 | 63.29 | 1.03 | 8.1 | 0.1 | 1.6 | 27.48 | 23.26 | 64.35 | 1.03 |
| Sport | 14.94 | 0 | \ | 23.03 | 16.91 | 62.02 | 1.3 | 15.54 | 0 | \ | 23.86 | 16.5 | 60.6 | 1.33 |
| Total | 13.61 | 0.3 | 1.5 | 20.64 | 14.99 | 65.47 | 1.21 | 14.36 | 0 | 1.9 | 19.93 | 13.69 | 65.7 | 1.23 |

| | Top-80% tf*idf | | | | | | | Top-70% tf*idf | | | | | | |
| | 1-1 | 1-n | | m-1 | | m-n | | 1-1 | 1-n | | m-1 | | m-n | |
| Domain | % | % | ñ | % | | % | ū(m/n) | % | % | ñ | % | | % | ū(m/n) |
|---|---|---|---|---|---|---|---|---|---|---|---|---|---|---|
| Compute | 16.45 | 0 | \ | 18.65 | 9.87 | 64.9 | 1.23 | 16.57 | 0 | \ | 18.82 | 9.71 | 64.61 | 1.24 |
| Mechanis | 17.56 | 0 | \ | 16.14 | 6.74 | 66.3 | 1.08 | 17.75 | 0 | \ | 16.44 | 6.49 | 65.8 | 1.09 |
| Power | 14.75 | 0 | \ | 16.52 | 8.33 | 68.73 | 1.14 | 15.04 | 0 | \ | 16.87 | 8.02 | 68.09 | 1.16 |
| Publising | 15.25 | 0 | \ | 29.53 | 22.67 | 55.21 | 1.26 | 15.98 | 0 | \ | 30.48 | 22.08 | 53.53 | 1.29 |
| Forestry | 14.9 | 0 | \ | 19.87 | 11.69 | 65.23 | 1.25 | 15.22 | 0 | \ | 20.3 | 11.38 | 64.47 | 1.27 |
| Industrial | 15.13 | 0 | \ | 22.45 | 10.65 | 62.42 | 1.1 | 15.69 | 0 | \ | 23.05 | 10.17 | 61.26 | 1.12 |
| Math | 22.23 | 0 | \ | 19.71 | 8.43 | 58.07 | 1.14 | 22.42 | 0 | \ | 19.89 | 8.29 | 57.69 | 1.15 |
| Environe | 14.55 | 0 | \ | 19.08 | 9.52 | 66.38 | 1.27 | 15.02 | 0 | \ | 18.59 | 8.75 | 66.39 | 1.3 |
| Fisheries | 12.68 | 0 | \ | 18.58 | 14.49 | 68.73 | 1.1 | 12.92 | 0 | \ | 18.55 | 13.77 | 68.53 | 1.11 |
| Animal | 11.59 | 0 | \ | 21.55 | 19.46 | 66.85 | 1.19 | 11.73 | 0 | \ | 21.81 | 19.18 | 66.46 | 1.2 |
| Architecr | 16.84 | 0 | \ | 18.17 | 10.32 | 64.99 | 1.31 | 17.04 | 0 | \ | 18.29 | 10.1 | 64.67 | 1.33 |
| Aerospace | 16.8 | 0 | \ | 18.85 | 8.55 | 64.34 | 1.25 | 17.16 | 0 | \ | 19.28 | 8.24 | 63.56 | 1.28 |
| LightIndu | 15.19 | 0 | \ | 19.99 | 12.91 | 64.82 | 1.27 | 15.73 | 0 | \ | 19.46 | 11.88 | 64.81 | 1.3 |

| | | | | | | | | | | | | | | |
|---|---|---|---|---|---|---|---|---|---|---|---|---|---|---|
| Finance | 14.75 | 0 | \ | 27.61 | 29.08 | 57.64 | 1.3 | 14.88 | 0 | \ | 27.84 | 28.8 | 57.28 | 1.32 |
| Physics | 13.52 | 0 | \ | 13.67 | 7.32 | 72.81 | 1.16 | 13.59 | 0 | \ | 13.75 | 7.15 | 72.66 | 1.17 |
| Traffic | 16.56 | 0 | \ | 23.06 | 12.83 | 60.38 | 1.3 | 16.81 | 0 | \ | 23.44 | 12.65 | 59.75 | 1.32 |
| History& | 12.23 | 0.5 | 2.2 | 32.83 | 15.97 | 54.47 | 0.99 | 12.91 | 0.5 | 2.2 | 34.25 | 15.52 | 52.33 | 0.99 |
| Agricue | 12.85 | 0 | \ | 15.93 | 13.58 | 71.22 | 1.28 | 12.91 | 0 | \ | 15.98 | 13.38 | 71.11 | 1.29 |
| Machiner | 15.9 | 0 | \ | 18.77 | 9.34 | 65.33 | 1.29 | 16.3 | 0 | \ | 18.14 | 8.48 | 65.55 | 1.33 |
| Chemistry | 14.83 | 0 | \ | 15.65 | 10.29 | 69.53 | 1.32 | 14.97 | 0 | \ | 15.21 | 9.67 | 69.82 | 1.35 |
| Law | 16.61 | 0.1 | 1.8 | 36.33 | 16.28 | 46.92 | 1.03 | 17.45 | 0.2 | 1.8 | 37.31 | 15.48 | 45.08 | 1.04 |
| Energy | 14.01 | 0 | \ | 14.55 | 8.85 | 71.44 | 1.25 | 14.13 | 0 | \ | 14.66 | 8.67 | 71.22 | 1.26 |
| Astronoy | 15.34 | 0 | \ | 14.44 | 9.99 | 70.22 | 1.32 | 15.4 | 0 | \ | 14.53 | 9.84 | 70.07 | 1.33 |
| Education | 13.99 | 0 | \ | 27.37 | 25.03 | 58.65 | 1.08 | 14.4 | 0 | \ | 26.13 | 22.94 | 59.46 | 1.09 |
| Power | 16.03 | 0 | \ | 19.62 | 10.68 | 64.35 | 1.25 | 16.33 | 0 | \ | 19.99 | 10.35 | 63.68 | 1.29 |
| Art- | 10.04 | 0.1 | 2.4 | 30.83 | 23.81 | 59.03 | 1.06 | 10.38 | 0.1 | 2.4 | 31.56 | 23.47 | 57.96 | 1.06 |
| Material | 13.54 | 0 | \ | 13.94 | 6.47 | 72.53 | 1.1 | 13.58 | 0 | \ | 14.09 | 6.27 | 72.33 | 1.11 |
| Metal | 15.73 | 0 | \ | 17.3 | 9.83 | 66.97 | 1.28 | 15.92 | 0 | \ | 17.49 | 9.56 | 66.6 | 1.31 |
| Electronic | 16.12 | 0 | \ | 18.74 | 10.63 | 65.14 | 1.24 | 16.24 | 0 | \ | 18.92 | 10.47 | 64.84 | 1.25 |
| Politics | 12.94 | 0.1 | 1.6 | 34.5 | 26.91 | 52.47 | 1.07 | 13.3 | 0.1 | 1.6 | 34.04 | 25.66 | 52.58 | 1.07 |
| Military | 13.15 | 0 | \ | 30 | 29.21 | 56.85 | 1.06 | 13.94 | 0 | \ | 31.12 | 28.28 | 54.94 | 1.07 |
| Medicine | 12.14 | 0 | \ | 14.28 | 15.63 | 73.58 | 1.48 | 12.14 | 0 | \ | 14.28 | 15.63 | 73.58 | 1.48 |
| Bioscie | 12.78 | 0 | \ | 12.48 | 7.29 | 74.75 | 1.13 | 12.8 | 0 | \ | 12.5 | | 74.71 | 1.13 |
| Mining | 17.58 | 0 | \ | 19.13 | 8.61 | 63.28 | 1.24 | 17.88 | 0 | \ | 19.47 | | 62.64 | 1.26 |
| Culture | 8.35 | 0.1 | 1.6 | 27.88 | 22.71 | 63.7 | 1.04 | 8.59 | 0.1 | 1.6 | 28.32 | | 63.01 | 1.04 |
| Sport | 16.29 | 0 | \ | 24.11 | 15.61 | 59.6 | 1.37 | 16.86 | 0 | \ | 24.48 | | 58.66 | 1.41 |
| Total | 14.6 | 0 | 1.9 | 19.83 | 13.13 | 65.56 | 1.25 | 14.77 | 0 | 1.9 | 19.78 | | 65.44 | 1.27 |

| | *Top-60% tf\*idf* | | | | | | | *Top-50% tf\*idf* | | | | | | |
|---|---|---|---|---|---|---|---|---|---|---|---|---|---|---|
| | 1-1 | 1-n | | m-1 | | m-n | | 1-1 | 1-n | | m-1 | | m-n | |
| Domain | % | % | ñ | % | | % | ū(m/n) | % | % | ñ | % | | % | ū(m/n) |
| Compute | 16.75 | 0 | \ | 19.16 | 9.38 | 64.09 | 1.28 | 16.75 | 0 | \ | 19.16 | 9.38 | 64.09 | 1.28 |
| Mechanis | 17.84 | 0 | \ | 16.54 | 6.45 | 65.63 | 1.09 | 18.09 | 0 | \ | 16.9 | 6.17 | 65.02 | 1.11 |
| Power | 15.2 | 0 | \ | 17.14 | 7.8 | 67.67 | 1.18 | 15.29 | 0 | \ | 17.22 | 7.76 | 67.49 | 1.18 |
| Publising | 17.34 | 0 | \ | 29.54 | 19.48 | 53.12 | 1.33 | 18.42 | 0 | \ | 28.38 | 17.44 | 53.2 | 1.37 |
| Forestry | 15.47 | 0 | \ | 20.75 | 11.06 | 63.77 | 1.3 | 15.63 | 0 | \ | 21.17 | 10.79 | 63.2 | 1.33 |
| Industrial | 16.14 | 0 | \ | 23.63 | 9.78 | 60.23 | 1.14 | 16.37 | 0 | \ | 23.93 | 9.69 | 59.7 | 1.14 |
| Math | 22.77 | 0 | \ | 20.26 | 7.99 | 56.96 | 1.18 | 22.77 | 0 | \ | 20.26 | 7.99 | 56.96 | 1.18 |
| Environe | 15.32 | 0 | \ | 18.82 | 8.41 | 65.86 | 1.33 | 15.38 | 0 | \ | 18.89 | 8.39 | 65.72 | 1.33 |
| Fisheries | 13.17 | 0 | \ | 18.91 | 13.42 | 67.92 | 1.13 | 13.29 | 0 | \ | 19.18 | 13.14 | 67.53 | 1.15 |
| Animal | 11.82 | 0 | \ | 22.02 | 18.95 | 66.17 | 1.21 | 11.92 | 0 | \ | 22.38 | 18.58 | 65.69 | 1.23 |
| Architecr | 17.33 | 0 | \ | 18.66 | 9.77 | 64.01 | 1.37 | 17.33 | 0 | \ | 18.66 | 9.77 | 64.01 | 1.37 |
| Aerospace | 17.47 | 0 | \ | 19.65 | 8.04 | 62.88 | 1.3 | 17.86 | 0 | \ | 20.16 | 7.77 | 61.99 | 1.34 |
| LightIndu | 15.97 | 0 | \ | 19.71 | 11.66 | 64.32 | 1.32 | 16.32 | 0 | \ | 20.14 | 11.32 | 63.54 | 1.36 |
| Finance | 15.18 | 0 | \ | 27.65 | 27.43 | 57.17 | 1.37 | 15.18 | 0 | \ | 27.65 | 27.43 | 57.17 | 1.37 |
| Physics | 13.62 | 0 | \ | 13.91 | 6.89 | 72.47 | 1.19 | 13.62 | 0 | \ | 13.91 | 6.89 | 72.47 | 1.19 |
| Traffic | 17.01 | 0 | \ | 23.82 | 12.45 | 59.16 | 1.34 | 17.35 | 0 | \ | 24.43 | 12.12 | 58.22 | 1.39 |
| History& | 13.63 | 0.6 | 2.2 | 35.86 | 15.01 | 49.96 | 1 | 14.08 | 0.6 | 2.2 | 36.58 | 14.43 | 48.76 | 1.02 |
| Agricue | 13.04 | 0 | \ | 15.54 | 12.46 | 71.42 | 1.33 | 13.04 | 0 | \ | 15.54 | 12.46 | 71.42 | 1.33 |
| Machiner | 16.3 | 0 | \ | 18.14 | 8.48 | 65.55 | 1.33 | 16.53 | 0 | \ | 18.53 | 8.15 | 64.94 | 1.38 |
| Chemistry | 14.97 | 0 | \ | 15.21 | 9.67 | 69.82 | 1.35 | 14.97 | 0 | \ | 15.21 | 9.67 | 69.82 | 1.35 |
| Law | 17.86 | 0.2 | 1.8 | 38.36 | 15.23 | 43.62 | 1.05 | 18.24 | 0.2 | 1.8 | 39.83 | 14.86 | 41.76 | 1.07 |
| Energy | 14.2 | 0 | \ | 14.82 | 8.35 | 70.98 | 1.3 | 14.31 | 0 | \ | 14.95 | 8.32 | 70.74 | 1.31 |
| Astronoy | 15.51 | 0 | \ | 14.48 | 9.35 | 70.01 | 1.37 | 15.51 | 0 | \ | 14.48 | 9.35 | 70.01 | 1.37 |
| Education | 14.78 | 0 | \ | 25.54 | 21.36 | 59.68 | 1.11 | 14.78 | 0 | \ | 25.54 | 21.36 | 59.68 | 1.11 |
| Power | 16.53 | 0 | \ | 20.3 | 10.08 | 63.17 | 1.32 | 16.61 | 0 | \ | 20.41 | 10.06 | 62.98 | 1.32 |
| Art- | 10.87 | 0.1 | 2.4 | 32.59 | 23.02 | 56.43 | 1.08 | 11.43 | 0.1 | 2.4 | 33.87 | 22.42 | 54.58 | 1.1 |
| Material | 13.61 | 0 | \ | 14.13 | 6.05 | 72.26 | 1.12 | 13.65 | 0 | \ | 14.19 | 6.02 | 72.16 | 1.12 |
| Metal | 16.12 | 0 | \ | 17.45 | 9.13 | 66.43 | 1.34 | 16.12 | 0 | \ | 17.45 | 9.13 | 66.43 | 1.34 |
| Electronic | 16.4 | 0 | \ | 19.24 | 10.16 | 64.36 | 1.29 | 16.4 | 0 | \ | 19.24 | 10.16 | 64.36 | 1.29 |
| Politics | 13.57 | 0.1 | 1.6 | 34.55 | 25.23 | 51.79 | 1.09 | 14.11 | 0.1 | 1.6 | 34.75 | 23.88 | 51.04 | 1.12 |
| Military | 14.77 | 0 | \ | 32.22 | 27.33 | 53.01 | 1.08 | 15.54 | 0 | \ | 33.29 | 26.55 | 51.18 | 1.09 |

| Domain | 1-1 % | 1-n % | 1-n ñ | m-1 % | m-1 | m-n % | m-n ū(m/n) | 1-1 % | 1-n % | 1-n ñ | m-1 % | m-1 | m-n % | m-n ū(m/n) |
|---|---|---|---|---|---|---|---|---|---|---|---|---|---|---|
| Medicine | 12.14 | 0 | \ | 14.28 | 15.63 | 73.58 | 1.48 | 12.14 | 0 | \ | 14.28 | 15.63 | 73.58 | 1.48 |
| Bioscie | 12.74 | 0 | \ | 12.61 | 6.97 | 74.66 | 1.16 | 12.74 | 0 | \ | 12.61 | 6.97 | 74.66 | 1.16 |
| Mining | 18.1 | 0 | \ | 19.71 | 8.19 | 62.19 | 1.27 | 18.48 | 0 | \ | 20.22 | 7.88 | 61.3 | 1.32 |
| Culture | 8.78 | 0.1 | 1.6 | 28.73 | 21.88 | 62.42 | 1.05 | 9.01 | 0.1 | 1.6 | 29.42 | 21.4 | 61.49 | 1.07 |
| Sport | 17.64 | 0 | \ | 24.74 | 14.2 | 57.62 | 1.47 | 18.43 | 0 | \ | 24.95 | 13.39 | 56.62 | 1.52 |
| Total | 14.91 | 0 | 1.9 | 19.8 | 12.39 | 65.28 | 1.29 | 14.99 | 0 | 1.9 | 19.84 | 12.27 | 65.17 | 1.3 |

| | Top-40% tf*idf | | | | | | | Top-30% tf*idf | | | | | | |
|---|---|---|---|---|---|---|---|---|---|---|---|---|---|---|
| | 1-1 | 1-n | | m-1 | | m-n | | 1-1 | 1-n | | m-1 | | m-n | |
| Domain | % | % | ñ | % | | % | ū(m/n) | % | % | ñ | % | | % | ū(m/n) |
| Compute | 16.75 | 0 | \ | 19.16 | 9.38 | 64.09 | 1.28 | 16.86 | 0 | \ | 19.67 | 8.75 | 63.48 | 1.42 |
| Mechanis | 18.09 | 0 | \ | 16.9 | 6.17 | 65.02 | 1.11 | 18.76 | 0 | \ | 16.28 | 5.2 | 64.96 | 1.19 |
| Power | 15.29 | 0 | \ | 17.22 | 7.76 | 67.49 | 1.18 | 15.46 | 0 | \ | 17.5 | 7.14 | 67.05 | 1.26 |
| Publising | 19.67 | 0 | \ | 28.31 | 16.05 | 52.01 | 1.42 | 19.67 | 0 | \ | 28.31 | 16.05 | 52.01 | 1.42 |
| Forestry | 15.63 | 0 | \ | 21.17 | 10.79 | 63.2 | 1.33 | 15.63 | 0 | \ | 21.17 | 10.79 | 63.2 | 1.33 |
| Industrial | 17.01 | 0 | \ | 24.74 | 9.26 | 58.24 | 1.17 | 17.01 | 0 | \ | 24.74 | 9.26 | 58.24 | 1.17 |
| Math | 23.11 | 0 | \ | 20.36 | 7.37 | 56.53 | 1.26 | 24.02 | 0 | \ | 21.08 | 7.11 | 54.91 | 1.3 |
| Environe | 15.73 | 0 | \ | 18.95 | 7.91 | 65.32 | 1.38 | 15.73 | 0 | \ | 18.95 | 7.91 | 65.32 | 1.38 |
| Fisheries | 13.29 | 0 | \ | 19.18 | 13.14 | 67.53 | 1.15 | 13.4 | 0 | \ | 19.18 | 12.26 | 67.42 | 1.2 |
| Animal | 11.92 | 0 | \ | 22.38 | 18.58 | 65.69 | 1.23 | 11.9 | 0 | \ | 22.83 | 17.59 | 65.27 | 1.3 |
| Architecr | 17.33 | 0 | \ | 18.66 | 9.77 | 64.01 | 1.37 | 17.33 | 0 | \ | 18.66 | 9.77 | 64.01 | 1.37 |
| Aerospace | 17.86 | 0 | \ | 20.16 | 7.77 | 61.99 | 1.34 | 17.86 | 0 | \ | 20.16 | 7.77 | 61.99 | 1.34 |
| LightIndu | 16.32 | 0 | \ | 20.14 | 11.32 | 63.54 | 1.36 | 16.32 | 0 | \ | 20.14 | 11.32 | 63.54 | 1.36 |
| Finance | 15.18 | 0 | \ | 27.65 | 27.43 | 57.17 | 1.37 | 16.33 | 0 | \ | 24.76 | 21.34 | 58.91 | 1.56 |
| Physics | 13.62 | 0 | \ | 13.91 | 6.89 | 72.47 | 1.19 | 13.62 | 0 | \ | 13.91 | 6.89 | 72.47 | 1.19 |
| Traffic | 17.35 | 0 | \ | 24.43 | 12.12 | 58.22 | 1.39 | 17.35 | 0 | \ | 24.43 | 12.12 | 58.22 | 1.39 |
| History& | 14.5 | 0.6 | 2.2 | 37.46 | 14.28 | 47.44 | 1.01 | 14.5 | 0.6 | 2.2 | 37.46 | 14.28 | 47.44 | 1.01 |
| Agricue | 13.04 | 0 | \ | 15.54 | 12.46 | 71.42 | 1.33 | 12.85 | 0 | \ | 15.27 | 11.4 | 71.88 | 1.42 |
| Machiner | 16.53 | 0 | \ | 18.53 | 8.15 | 64.94 | 1.38 | 16.53 | 0 | \ | 18.53 | 8.15 | 64.94 | 1.38 |
| Chemistry | 14.97 | 0 | \ | 15.21 | 9.67 | 69.82 | 1.35 | 15.19 | 0 | \ | 14.65 | 8.22 | 70.16 | 1.49 |
| Law | 18.24 | 0.2 | 1.8 | 39.83 | 14.86 | 41.76 | 1.07 | 18.24 | 0.2 | 1.8 | 39.83 | 14.86 | 41.76 | 1.07 |
| Energy | 14.31 | 0 | \ | 14.95 | 8.32 | 70.74 | 1.31 | 14.31 | 0 | \ | 14.95 | 8.32 | 70.74 | 1.31 |
| Astronoy | 15.51 | 0 | \ | 14.48 | 9.35 | 70.01 | 1.37 | 15.51 | 0 | \ | 14.48 | 9.35 | 70.01 | 1.37 |
| Education | 14.78 | 0 | \ | 25.54 | 21.36 | 59.68 | 1.11 | 14.78 | 0 | \ | 25.54 | 21.36 | 59.68 | 1.11 |
| Power | 16.61 | 0 | \ | 20.41 | 10.06 | 62.98 | 1.32 | 16.61 | 0 | \ | 20.41 | 10.06 | 62.98 | 1.32 |
| Art- | 11.76 | 0.1 | 2.4 | 34.71 | 22.31 | 53.4 | 1.11 | 11.76 | 0.1 | 2.4 | 34.71 | 22.31 | 53.4 | 1.11 |
| Material | 13.58 | 0 | \ | 14.29 | 5.77 | 72.14 | 1.16 | 13.83 | 0 | \ | 14.5 | 5.56 | 71.67 | 1.17 |
| Metal | 16.12 | 0 | \ | 17.45 | 9.13 | 66.43 | 1.34 | 16.12 | 0 | \ | 17.45 | 9.13 | 66.43 | 1.34 |
| Electronic | 16.4 | 0 | \ | 19.24 | 10.16 | 64.36 | 1.29 | 16.4 | 0 | \ | 19.24 | 10.16 | 64.36 | 1.29 |
| Politics | 14.11 | 0.1 | 1.6 | 34.75 | 23.88 | 51.04 | 1.12 | 14.11 | 0.1 | 1.6 | 34.75 | 23.88 | 51.04 | 1.12 |
| Military | 16.46 | 0 | \ | 34.52 | 25.59 | 49.02 | 1.11 | 17.3 | 0 | \ | 35.84 | 24.7 | 46.86 | 1.16 |
| Medicine | 11.96 | 0 | \ | 13.86 | 14.22 | 74.18 | 1.63 | 12.14 | 0 | \ | 13.26 | 13.33 | 74.6 | 1.64 |
| Bioscie | 12.75 | 0 | \ | 12.59 | 6.54 | 74.66 | 1.2 | 12.89 | 0 | \ | 12.76 | 6.39 | 74.36 | 1.21 |
| Mining | 18.48 | 0 | \ | 20.22 | 7.88 | 61.3 | 1.32 | 18.48 | 0 | \ | 20.22 | 7.88 | 61.3 | 1.32 |
| Culture | 9.01 | 0.1 | 1.6 | 29.42 | 21.4 | 61.49 | 1.07 | 8.88 | 0.1 | 1.6 | 29.65 | 20.59 | 61.39 | 1.17 |
| Sport | 18.43 | 0 | \ | 24.95 | 13.39 | 56.62 | 1.52 | 18.43 | 0 | \ | 24.95 | 13.39 | 56.62 | 1.52 |
| Total | 14.99 | 0 | 1.9 | 19.87 | 12 | 65.13 | 1.32 | 15.15 | 0 | 1.9 | 19.65 | 11.48 | 65.19 | 1.36 |

| | Top-20% tf*idf | | | | | | | Top-10% tf*idf | | | | | | |
|---|---|---|---|---|---|---|---|---|---|---|---|---|---|---|
| | 1-1 | 1-n | | m-1 | | m-n | | 1-1 | 1-n | | m-1 | | m-n | |
| Domain | % | % | ñ | % | | % | ū(m/n) | % | % | ñ | % | | % | ū(m/n) |
| Compute | 17.52 | 0 | \ | 20.37 | 8.41 | 62.12 | 1.49 | 18.08 | 0 | \ | 21.09 | 8.13 | 60.82 | 1.57 |
| Mechanis | 19.42 | 0 | \ | 16.86 | 4.93 | 63.71 | 1.23 | 19.78 | 0 | \ | 17.19 | 4.88 | 63.03 | 1.24 |
| Power | 16.33 | 0 | \ | 18.31 | 6.8 | 65.36 | 1.31 | 16.33 | 0 | \ | 18.31 | 6.8 | 65.36 | 1.31 |
| Publising | 19.67 | 0 | \ | 28.31 | 16.05 | 52.01 | 1.42 | 23.61 | 0 | \ | 32.29 | 14.16 | 44.1 | 1.8 |
| Forestry | 15.63 | 0 | \ | 21.17 | 10.79 | 63.2 | 1.33 | 16.36 | 0 | \ | 22.78 | 9.64 | 60.86 | 1.52 |
| Industrial | 18.14 | 0 | \ | 26.14 | 8.51 | 55.72 | 1.26 | 19.79 | 0 | \ | 27.81 | 8.08 | 52.4 | 1.3 |
| Math | 24.63 | 0 | \ | 21.57 | 6.91 | 53.79 | 1.33 | 25.29 | 0 | \ | 22.17 | 6.58 | 52.54 | 1.43 |
| Environe | 15.63 | 0 | \ | 19.23 | 7.51 | 65.14 | 1.49 | 16.76 | 0 | \ | 20.47 | 7.2 | 62.76 | 1.55 |

| | | | | | | | | | | | | | |
|---|---|---|---|---|---|---|---|---|---|---|---|---|---|
| Fisheries | 13.88 | 0 | \ | 19.88 | 11.91 | 66.24 | 1.24 | 14.33 | 0 | \ | 20.55 | 11.59 | 65.13 | 1.28 |
| Animal | 12.4 | 0 | \ | 23.75 | 17.1 | 63.85 | 1.34 | 13.08 | 0 | \ | 25.08 | 16.63 | 61.83 | 1.4 |
| Architecr | 17.58 | 0 | \ | 18.53 | 8.5 | 63.88 | 1.59 | 19.09 | 0 | \ | 19.14 | 7.87 | 61.78 | 1.66 |
| Aerospace | 18.68 | 0 | \ | 20.95 | 6.99 | 60.37 | 1.53 | 19.85 | 0 | \ | 22.09 | 6.76 | 58.06 | 1.6 |
| LightIndu | 16.36 | 0 | \ | 20.39 | 10.46 | 63.25 | 1.51 | 18.04 | 0 | \ | 20.59 | 9.34 | 61.37 | 1.57 |
| Finance | 17.17 | 0 | \ | 25.87 | 20.95 | 56.96 | 1.64 | 19.23 | 0 | \ | 28.39 | 20.13 | 52.38 | 1.84 |
| Physics | 13.82 | 0 | \ | 14.42 | 6.06 | 71.76 | 1.34 | 14.08 | 0 | \ | 14.71 | 5.94 | 71.21 | 1.36 |
| Traffic | 17.35 | 0 | \ | 24.43 | 12.12 | 58.22 | 1.39 | 18.87 | 0 | \ | 27.34 | 10.95 | 53.79 | 1.67 |
| History& | 14.5 | 0.6 | 2.2 | 37.46 | 14.28 | 47.44 | 1.01 | 17.45 | 0.8 | 2.2 | 44.81 | 12.81 | 36.94 | 1.22 |
| Agricue | 13.37 | 0 | \ | 15.41 | 10.72 | 71.22 | 1.47 | 13.85 | 0 | \ | 16 | 10.41 | 70.15 | 1.53 |
| Machiner | 16.61 | 0 | \ | 18.88 | 7.45 | 64.51 | 1.56 | 17.87 | 0 | \ | 20.27 | 7.16 | 61.86 | 1.64 |
| Chemistry | 15.37 | 0 | \ | 14.8 | 8.09 | 69.83 | 1.52 | 15.78 | 0 | \ | 15.3 | 7.73 | 68.92 | 1.62 |
| Law | 18.24 | 0.2 | 1.8 | 39.83 | 14.86 | 41.76 | 1.07 | 19.9 | 0.2 | 1.8 | 46.54 | 13.57 | 33.35 | 1.39 |
| Energy | 14.31 | 0 | \ | 14.95 | 8.32 | 70.74 | 1.31 | 14.67 | 0 | \ | 15.83 | 7.4 | 69.5 | 1.48 |
| Astronoy | 15.86 | 0 | \ | 14.44 | 7.76 | 69.7 | 1.65 | 16.36 | 0 | \ | 14.48 | 7.25 | 69.16 | 1.76 |
| Education | 15.42 | 0 | \ | 26.76 | 19.88 | 57.83 | 1.27 | 17.22 | 0 | \ | 29.51 | 19.29 | 53.27 | 1.38 |
| Power | 16.61 | 0 | \ | 20.41 | 10.06 | 62.98 | 1.32 | 17.5 | 0 | \ | 22.17 | 9.04 | 60.33 | 1.55 |
| Art- | 12.54 | 0.1 | 2.4 | 36.68 | 21.16 | 50.64 | 1.27 | 15.02 | 0.2 | 2.4 | 42.42 | 20.34 | 42.39 | 1.46 |
| Material | 14.24 | 0 | \ | 14.94 | 5.32 | 70.82 | 1.2 | 14.24 | 0 | \ | 14.94 | 5.32 | 70.82 | 1.2 |
| Metal | 15.85 | 0 | \ | 17.74 | 8.4 | 66.41 | 1.5 | 17.07 | 0 | \ | 16.78 | 7.18 | 66.16 | 1.55 |
| Electronic | 16.81 | 0 | \ | 20.06 | 9.08 | 63.13 | 1.5 | 17.68 | 0 | \ | 20.98 | 8.68 | 61.34 | 1.61 |
| Politics | 14.91 | 0.1 | 1.6 | 36.78 | 22.14 | 48.2 | 1.3 | 17.03 | 0.1 | 1.6 | 41.16 | 21.42 | 41.67 | 1.5 |
| Military | 17.33 | 0 | \ | 35.9 | 24.69 | 46.77 | 1.16 | 19.18 | 0 | \ | 38.15 | 23.16 | 42.66 | 1.26 |
| Medicine | 12.19 | 0 | \ | 13.32 | 12.92 | 74.49 | 1.73 | 12.63 | 0 | \ | 13.34 | 12.22 | 74.03 | 1.81 |
| Bioscie | 13.02 | 0 | \ | 12.92 | 6.18 | 74.07 | 1.24 | 13.29 | 0 | \ | 13.24 | 5.92 | 73.48 | 1.28 |
| Mining | 18.48 | 0 | \ | 20.22 | 7.88 | 61.3 | 1.32 | 19.69 | 0 | \ | 21.65 | 6.85 | 58.66 | 1.54 |
| Culture | 9.75 | 0.1 | 1.6 | 31.99 | 20.08 | 58.17 | 1.23 | 11.46 | 0.1 | 1.6 | 37.02 | 19.51 | 51.41 | 1.37 |
| Sport | 18.43 | 0 | \ | 24.95 | 13.39 | 56.62 | 1.52 | 20.25 | 0 | \ | 26.13 | 11.42 | 53.61 | 1.81 |
| Total | 15.43 | 0 | 1.9 | 19.88 | 10.73 | 64.68 | 1.46 | 16.32 | 0 | 1.9 | 20.74 | 10.14 | 62.93 | 1.58 |

Table 22: Question patterns for TechQA.

| Question Type | Patterns |
|---|---|
| Who | *Who published a paper titled B in journal C?* |
| | *With whom A published a paper titled B in journal C?* |
| | *With whom A published a paper titled B?* |
| | *With whom A published a paper in journal C?* |
| When | *When A published a paper titled B in journal C?* |
| | *When A published a paper titled B with C in journal D?* |
| | *When A published a paper titled B?* |
| | *When A published a paper in journal C?* |
| | *When A published a paper titled B with C?* |
| What | *What is/are the research interests of A?* |
| | *What are the papers' titles that A published with B in journal C?* |
| | *What are the papers' titles that A published with B?* |
| | *What are the papers' titles that A published in journal C?* |
| Where | *Where A published a paper titled B with C?* |
| | *Where A published a paper titled B?* |
| | *Where A published a paper with C?* |

Table 23: Completed and detailed statistics for TechRE, TechNER, and TechQA.

| Domain | TechRE | | | TechNER | | TechQA |
| --- | --- | --- | --- | --- | --- | --- |
| | Total Training bags | Average sentences per bag | Average entities per sentence | Total Training samples | Average entities per sentence | Total training samples |
| Computer science | 198446 | 34.59 | 4.27 | 1576964 | 3.23 | 43951 |
| Mechanics | 8696 | 8.1 | 3 | 73027 | 2.1 | 5852 |
| Power | 67952 | 21.76 | 4.09 | 381936 | 3.07 | 34505 |
| Publishing-related | 127046 | 32.85 | 4.28 | 479795 | 2.06 | 5699 |
| Forestry | 100212 | 19.54 | 4.21 | 523874 | 3.4 | 23943 |
| Industrial technology | 52761 | 22.63 | 3.53 | 463889 | 2.35 | 18122 |
| Math-related | 23379 | 9.63 | 3.36 | 151508 | 2.71 | 10389 |
| Environment-related | 121595 | 28.87 | 4.38 | 752569 | 3.23 | 34153 |
| Fisheries-related | 48935 | 11.19 | 3.71 | 241742 | 2.88 | 12604 |
| Animal Husbandry | 190973 | 16.8 | 4.22 | 975686 | 3.59 | 52841 |
| Architecture-related | 528891 | 34.87 | 5.45 | 2183504 | 4.59 | 50214 |
| Aerospace | 54492 | 21.89 | 3.76 | 404377 | 2.64 | 16876 |
| Light Industry | 325758 | 22.42 | 4.6 | 1428152 | 3.73 | 39812 |
| Finance | 1270481 | 55.09 | 5.92 | 5291369 | 2.63 | 31205 |
| Physics-related | 31429 | 10.37 | 3.29 | 28838 | 2.89 | 55042 |
| Traffic transportation | 352681 | 34.76 | 5.03 | 1622627 | 4.06 | 31308 |
| History&Geography | 90942 | 9.52 | 3.35 | 673715 | 2.7 | 6240 |
| Agriculture-related | 692626 | 29.16 | 4.99 | 2153733 | 2.52 | 159034 |
| Machinery | 198786 | 28.18 | 4.4 | 1151598 | 3.37 | 45605 |
| Chemistry | 468372 | 31.92 | 5.3 | 2067044 | 4.64 | 119719 |
| Law | 88832 | 22.11 | 4.18 | 623381 | 3.77 | 3487 |
| Energy-related | 219147 | 30.02 | 4.9 | 861678 | 4 | 44308 |
| Astronomy | 286018 | 24.26 | 4.57 | 1695248 | 4.1 | 90876 |
| Education | 1553391 | 47.56 | 6.15 | 4485715 | 2.9 | 22852 |
| Power industry | 206058 | 36.68 | 5.06 | 1048190 | 4.12 | 40081 |
| Art-related | 190234 | 18.83 | 4.1 | 1079659 | 3.27 | 3038 |
| Material science | 21809 | 18.55 | 3.93 | 160414 | 2.88 | 14794 |
| Social Science | 1709737 | 54.96 | 6.39 | 6160722 | 3.27 | 72049 |
| Metallurgical industry | 244066 | 27.66 | 4.77 | 1237446 | 4.03 | 58396 |
| Nature Science | 491841 | 31.56 | 4.59 | 1738890 | 1.78 | 138960 |
| Electric industry | 244667 | 29.23 | 4.48 | 1655899 | 3.78 | 60805 |
| Politics-related | 677707 | 57.58 | 6.09 | 3915464 | 5.57 | 6898 |
| Military | 15939 | 13.32 | 3.16 | 170399 | 2.45 | 3240 |
| Medicine-related | 2527051 | 61.96 | 6.48 | 10489452 | 3.51 | 1004798 |
| Bioscience | 102197 | 16.63 | 3.77 | 794692 | 3.29 | 80121 |
| Mining industry | 124676 | 24.27 | 4.54 | 597232 | 3.78 | 23042 |
| Culture-related | 420254 | 10.72 | 3.69 | 1235663 | 1.7 | 8129 |
| Sport-related | 133844 | 34.09 | 4.7 | 739997 | 3.54 | 12184 |
| Total | 14211921 | 42.8 | 5.53 | 213051613 | 3.46 | 2485172 |

Table 24: Detailed Statistics of TechKG10

| | Law | | Medicine-Related | | Politics-related | | Nature Science-related | |
|---|---|---|---|---|---|---|---|---|
| | Ent# | Tri# | Ent# | Tri# | Ent# | Tri# | Ent# | Tri# |
| author_of | 10251 | 9530 | 1329892 | 3757657 | 21505 | 19477 | 553358 | 904583 |
| first_author | 7206 | 4609 | 1031978 | 796863 | 14760 | 8974 | 404553 | 278297 |
| second_author | 6169 | 3640 | 1037238 | 816297 | 12634 | 7491 | 400714 | 281965 |
| co_author | 4225 | 6096 | 486748 | 5579998 | 8875 | 13312 | 249807 | 1022342 |
| research_inter | 15311 | 51815 | 577178 | 9265515 | 33864 | 119106 | 324168 | 2107432 |
| affiliation | 1667 | 1970 | 356185 | 1561538 | 1748 | 1776 | 240127 | 784487 |
| belged_domainl | 1539 | 11539 | 113385 | 113385 | 36011 | 36011 | 77109 | 77109 |
| co-occurrence_with | 11517 | 172108 | 113358 | 5028608 | 36002 | 1288488 | 76856 | 1040701 |
| published_year | 4360 | 4638 | 237459 | 244717 | 7097 | 8669 | 191152 | 192200 |
| contained_cky | 23 | 14 | 3132 | 2190 | 327 | 233 | 381 | 227 |
| contained_eky | 0 | 0 | 295 | 209 | 0 | 0 | 10 | 5 |
| other_author | 1583 | 1282 | 1125980 | 2144915 | 4050 | 3024 | 342786 | 344343 |
| all_authors | 302 | 161 | 4337 | 2467 | 1405 | 770 | 2000 | 1156 |
| publi_journal | 6672 | 6955 | 789237 | 807641 | 15855 | 24738 | 303987 | 305581 |
| translation_of | 481 | 490 | 39543 | 70955 | 1269 | 1359 | 19094 | 25968 |
| Total | 25694 | 274847 | 1506141 | 30192955 | 67814 | 1533428 | 662349 | 7366396 |
| | Sport-related | | Publishing-related | | Environment-related | | Fisheries-related | |
| | Ent# | Tri# | Ent# | Tri# | Ent# | Tri# | Ent# | Tri# |
| author_of | 39120 | 56092 | 21727 | 26093 | 125736 | 211453 | 32525 | 60739 |
| first_author | 28935 | 19948 | 15397 | 10040 | 84717 | 56501 | 21322 | 14444 |
| second_author | 28275 | 19177 | 14662 | 9483 | 82485 | 56988 | 20741 | 14440 |
| co_author | 17562 | 55108 | 9829 | 20679 | 62143 | 264020 | 17574 | 84976 |
| research_inter | 26622 | 168610 | 20102 | 95749 | 75213 | 484048 | 20481 | 118011 |
| affiliation | 12339 | 29668 | 2155 | 2430 | 24029 | 43878 | 16320 | 50628 |
| belged_domainl | 0897 | 10897 | 13836 | 13836 | 16176 | 16175 | 5817 | 5816 |
| co-occurrence_with | 10872 | 242830 | 13809 | 279237 | 16093 | 228449 | 5798 | 109735 |
| published_year | 9986 | 10217 | 475 | 575 | 49745 | 50429 | 11033 | 11346 |
| contained_cky | 78 | 59 | 44 | 29 | 68 | 46 | 21 | 15 |
| contained_eky | 0 | 0 | 0 | 0 | 2 | 1 | 0 | 0 |
| other_author | 19268 | 16976 | 8693 | 6570 | 89629 | 97981 | 25186 | 31857 |
| all_authors | 484 | 270 | 410 | 223 | 362 | 213 | 47 | 30 |
| publi_journal | 20805 | 21293 | 12968 | 13365 | 63232 | 63845 | 13889 | 14075 |
| translation_of | 1112 | 1470 | 1648 | 1726 | 2846 | 3348 | 349 | 382 |
| Total | 53594 | 652615 | 39915 | 480035 | 147979 | 1577375 | 40687 | 516494 |
| | Metallurgical industry | | Physics-related | | History&Geography | | Power engineering | |
| | Ent# | Tri# | Ent# | Tri# | Ent# | Tri# | Ent# | Tri# |
| author_of | 202478 | 376503 | 102493 | 198790 | 13604 | 16398 | 156256 | 258057 |
| first_author | 145841 | 101664 | 67635 | 44707 | 7882 | 4824 | 112852 | 77453 |
| second_author | 141701 | 100706 | 65924 | 45505 | 7691 | 4768 | 110428 | 77572 |
| co_author | 93547 | 441527 | 53753 | 354155 | 8022 | 25212 | 71474 | 276238 |
| research_inter | 124060 | 879810 | 63555 | 317196 | 19950 | 48261 | 95582 | 634360 |
| affiliation | 76176 | 225998 | 38019 | 106026 | 2391 | 3260 | 62508 | 190150 |
| belged_domain | 33049 | 33048 | 14822 | 14821 | 16711 | 16710 | 25895 | 25895 |
| co-occurrence_with | 32931 | 494016 | 14555 | 86215 | 16663 | 264060 | 25804 | 396951 |
| published_year | 28657 | 29297 | 49899 | 50330 | 4543 | 4764 | 87388 | 88805 |
| contained_cky | 138 | 84 | 82 | 48 | 29 | 22 | 86 | 53 |
| contained_eky | 0 | 0 | 2 | 1 | 0 | 0 | 2 | 1 |
| other_author | 144883 | 174187 | 77987 | 108584 | 7517 | 6806 | 100774 | 103083 |

| | | | | | | | | |
|---|---|---|---|---|---|---|---|---|
| all_authors | 1044 | 622 | 129 | 77 | 113 | 61 | 1027 | 573 |
| publi_journal | 110816 | 112490 | 47952 | 47978 | 6846 | 6981 | 82985 | 84539 |
| translation_of | 5869 | 7180 | 2218 | 2394 | 275 | 271 | 3724 | 4352 |
| Total | 248236 | 2977132 | 122410 | 1376827 | 35077 | 402398 | 193004 | 2218082 |

| | Math-related | | Military | | Machinery | | Architecture-related | |
|---|---|---|---|---|---|---|---|---|
| | Ent# | Tri# | Ent# | Tri# | Ent# | Tri# | Ent# | Tri# |
| author_of | 50457 | 64322 | 7189 | 11380 | 208939 | 350641 | 216669 | 345047 |
| first_author | 36614 | 24230 | 4257 | 2710 | 150981 | 104146 | 157830 | 110878 |
| second_author | 35835 | 24732 | 4148 | 2654 | 149407 | 107603 | 156400 | 110076 |
| co_author | 23990 | 48284 | 4244 | 17599 | 89431 | 353023 | 93073 | 342245 |
| research_inter | 29960 | 113592 | 5906 | 20623 | 115023 | 757508 | 140007 | 945441 |
| affiliation | 25131 | 79075 | 2980 | 8077 | 50380 | 104554 | 60345 | 162820 |
| belged_domain | 9424 | 9423 | 3624 | 3624 | 29674 | 29673 | 52173 | 52172 |
| co-occurrence_with | 9318 | 67567 | 3585 | 35216 | 29531 | 425569 | 52082 | 1000942 |
| published_year | 27933 | 28091 | 3408 | 3467 | 84763 | 85741 | 29059 | 29332 |
| contained_cky | 45 | 26 | 7 | 6 | 99 | 62 | 164 | 99 |
| contained_eky | 0 | 0 | 11 | 10 | 4 | 2 | 7 | 6 |
| other_author | 21609 | 15366 | 5006 | 6016 | 140709 | 138908 | 131650 | 124143 |
| all_authors | 278 | 180 | 3 | 2 | 537 | 308 | 1499 | 870 |
| publi_journal | 25139 | 25229 | 3402 | 3405 | 115594 | 116343 | 119422 | 121800 |
| translation_of | 1029 | 1640 | 236 | 231 | 5124 | 5833 | 5857 | 7066 |
| Total | 65557 | 501757 | 11975 | 115020 | 247582 | 2579914 | 283294 | 3352937 |

| | Agriculture-related | | Electric industry | | Forestry | | Material science | |
|---|---|---|---|---|---|---|---|---|
| | Ent# | Tri# | Ent# | Tri# | Ent# | Tri# | Ent# | Tri# |
| author_of | 367390 | 792108 | 212926 | 367206 | 77392 | 132902 | 55966 | 87833 |
| first_author | 261789 | 186313 | 155437 | 106513 | 51430 | 34903 | 34542 | 21506 |
| second_author | 257138 | 187965 | 153910 | 112630 | 50427 | 34875 | 33157 | 21964 |
| co_author | 169794 | 1113293 | 90428 | 387883 | 38493 | 176327 | 32759 | 121330 |
| research_inter | 202184 | 1872518 | 123461 | 802381 | 48104 | 285167 | 36972 | 165228 |
| affiliation | 129799 | 445648 | 73760 | 208663 | 16711 | 32580 | 30109 | 78613 |
| belged_domain | 47621 | 47621 | 40001 | 40001 | 12486 | 12486 | 6570 | 6570 |
| co-occurrence_with | 47574 | 1414785 | 39921 | 655854 | 12421 | 185877 | 6462 | 46256 |
| published_year | 83455 | 84388 | 83793 | 85321 | 40146 | 40464 | 23019 | 23098 |
| contained_cky | 257 | 181 | 229 | 140 | 32 | 22 | 33 | 26 |
| contained_eky | 2 | 1 | 21 | 13 | 0 | 0 | 0 | 0 |
| other_author | 293233 | 417871 | 135632 | 148081 | 55749 | 63134 | 41114 | 44363 |
| all_authors | 448 | 275 | 1052 | 663 | 131 | 67 | 69 | 52 |
| publi_journal | 185995 | 188968 | 118211 | 119924 | 35667 | 35861 | 23272 | 23297 |
| translation_of | 9247 | 13648 | 10807 | 12282 | 843 | 971 | 1484 | 1633 |
| Total | 432620 | 6765583 | 264637 | 3047555 | 93130 | 1035636 | 65759 | 641769 |

| | Art-related | | Power Industry | | Mechanics | | Aerospace | |
|---|---|---|---|---|---|---|---|---|
| | Ent# | Tri# | Ent# | Tri# | Ent# | Tri# | Ent# | Tri# |
| author_of | 4171 | 3978 | 92715 | 154519 | 29192 | 40268 | 77208 | 123739 |
| first_author | 2111 | 1273 | 59953 | 39363 | 19759 | 12847 | 54999 | 37038 |
| second_author | 2005 | 1220 | 58746 | 39558 | 19306 | 13102 | 54055 | 38465 |
| co_author | 2908 | 9172 | 50214 | 218643 | 15071 | 39334 | 35253 | 122117 |
| research_inter | 13016 | 35047 | 61792 | 294510 | 16752 | 58734 | 43249 | 209388 |
| affiliation | 265 | 258 | 44098 | 117659 | 14122 | 36477 | 23854 | 56031 |
| belged_domain | 14033 | 14033 | 15261 | 15260 | 3461 | 3461 | 12136 | 12136 |
| co-occurrence_with | 14030 | 455486 | 15021 | 133627 | 3395 | 19252 | 12001 | 118672 |
| published_year | 1767 | 2623 | 32003 | 32290 | 14558 | 14546 | 16775 | 17020 |
| contained_cky | 269 | 209 | 35 | 19 | 2 | 1 | 40 | 26 |

| | | | | | | | | |
|---|---|---|---|---|---|---|---|---|
| contained_eky | 0 | 0 | 0 | 0 | 0 | 0 | 0 | 0 |
| other_author | 1713 | 1494 | 66117 | 75609 | 17115 | 14321 | 50411 | 48242 |
| all_authors | 338 | 206 | 248 | 134 | 33 | 27 | 130 | 72 |
| publi_journal | 3199 | 4095 | 44098 | 44392 | 10571 | 10565 | 40894 | 41050 |
| translation_of | 144 | 140 | 1724 | 1887 | 422 | 585 | 927 | 958 |
| Total | 22483 | 529234 | 114666 | 1167470 | 34549 | 263520 | 93319 | 824954 |

| | Bioscience | | Animal Hus-bandry | | Energy-related | | Mining Industry | |
|---|---|---|---|---|---|---|---|---|
| | Ent# | Tri# | Ent# | Tri# | Ent# | Tri# | Ent# | Tri# |
| author_of | 186930 | 339841 | 89700 | 178260 | 143319 | 246040 | 101033 | 151265 |
| first_author | 120526 | 79421 | 57976 | 38999 | 97315 | 66082 | 69664 | 46699 |
| second_author | 118260 | 80497 | 56369 | 39339 | 96722 | 65572 | 67909 | 45768 |
| co_author | 103173 | 532748 | 47741 | 286241 | 74251 | 330765 | 48093 | 162094 |
| research_inter | 116001 | 693164 | 57385 | 378149 | 92585 | 734045 | 65591 | 337779 |
| affiliation | 93517 | 296214 | 26312 | 59685 | 64177 | 241310 | 24135 | 46338 |
| belged_domain | 17459 | 17459 | 18040 | 18039 | 19601 | 19600 | 19216 | 19216 |
| co-occurrence_with | 17379 | 210251 | 18028 | 445930 | 19527 | 379263 | 19115 | 227004 |
| published_year | 69801 | 70082 | 40104 | 42168 | 57474 | 58093 | 37938 | 38336 |
| contained_cky | 93 | 56 | 116 | 77 | 31 | 23 | 15 | 10 |
| contained_eky | 2 | 1 | 11 | 10 | 0 | 0 | 14 | 13 |
| other_author | 142871 | 179933 | 71255 | 99945 | 105059 | 114421 | 63771 | 58803 |
| all_authors | 220 | 161 | 169 | 93 | 341 | 188 | 413 | 223 |
| publi_journal | 81627 | 81757 | 41938 | 44295 | 68590 | 69005 | 53721 | 54195 |
| translation_of | 4966 | 6389 | 1832 | 2106 | 2749 | 3493 | 1434 | 1577 |
| Total | 216087 | 2587974 | 111528 | 1633336 | 172372 | 2327900 | 126860 | 1189320 |

| | Astronomy | | Culture-related | | Light Industry | | Social Science & Philosophy-related | |
|---|---|---|---|---|---|---|---|---|
| | Ent# | Tri# | Ent# | Tri# | Ent# | Tri# | Ent# | Tri# |
| author_of | 261764 | 537706 | 12507 | 12166 | 156382 | 262230 | 250650 | 295782 |
| first_author | 196980 | 143830 | 8452 | 5344 | 111030 | 77888 | 188027 | 126276 |
| second_author | 192263 | 144006 | 7322 | 4572 | 108961 | 77797 | 180403 | 119382 |
| co_author | 108822 | 671164 | 6038 | 16275 | 70892 | 281892 | 99977 | 199917 |
| research_inter | 142647 | 1223155 | 22498 | 70632 | 98356 | 647939 | 221173 | 1430544 |
| affiliation | 65533 | 211438 | 1005 | 1112 | 50855 | 133089 | 72759 | 143853 |
| belged_domain | 38159 | 38158 | 24306 | 24306 | 35396 | 35395 | 106131 | 106131 |
| co-occurrence_with | 38070 | 550541 | 24294 | 1338615 | 35273 | 652721 | 106042 | 3209705 |
| published_year | 33294 | 33581 | 8864 | 13914 | 25522 | 26117 | 2102 | 2399 |
| contained_cky | 188 | 116 | 427 | 327 | 162 | 105 | 282 | 194 |
| contained_eky | 9 | 6 | 0 | 0 | 4 | 2 | 11 | 7 |
| other_author | 197176 | 249897 | 2815 | 2265 | 101604 | 106589 | 66827 | 50134 |
| all_authors | 517 | 337 | 1289 | 770 | 1348 | 830 | 6744 | 3843 |
| publi_journal | 150371 | 150845 | 11924 | 22092 | 84539 | 86691 | 152525 | 158459 |
| translation_of | 6515 | 8016 | 1225 | 1204 | 4972 | 5814 | 20690 | 37479 |
| Total | 310418 | 3962796 | 43321 | 1513594 | 202297 | 2395099 | 402514 | 5884105 |

| | Education | | Traffic Transportation | | Industrial Technology | | Finance | |
|---|---|---|---|---|---|---|---|---|
| | Ent# | Tri# | Ent# | Tri# | Ent# | Tri# | Ent# | Tri# |
| author_of | 76464 | 80324 | 145027 | 215219 | 81973 | 110204 | 131243 | 165185 |
| first_author | 53592 | 34260 | 105535 | 71575 | 50893 | 31919 | 97488 | 67219 |
| second_author | 50198 | 31895 | 103962 | 72269 | 50234 | 32639 | 94155 | 64920 |
| co_author | 30647 | 51924 | 62605 | 195227 | 44528 | 129752 | 50958 | 115497 |
| research_inter | 75003 | 359370 | 92660 | 549797 | 48267 | 160936 | 101293 | 595651 |
| affiliation | 11306 | 14084 | 40018 | 89779 | 34034 | 68532 | 27167 | 46638 |
| belged_domain | 61161 | 61162 | 36162 | 36162 | 13151 | 13150 | 71913 | 71912 |

| | | | | | | | | |
|---|---|---|---|---|---|---|---|---|
| co-occurrence_with | 61148 | 3769343 | 36031 | 684482 | 12903 | 124291 | 71903 | 3304046 |
| published_year | 112011 | 15824 | 85806 | 88376 | 34538 | 35107 | 5562 | 6733 |
| contained_cky | 1663 | 1249 | 130 | 90 | 49 | 27 | 590 | 393 |
| contained_eky | 0 | 0 | 2 | 1 | 2 | 1 | 2 | 1 |
| other_author | 20260 | 14171 | 82256 | 71396 | 51233 | 45651 | 44480 | 33057 |
| all_authors | 3575 | 2241 | 1130 | 628 | 148 | 85 | 2926 | 1645 |
| publi_journal | 53197 | 76178 | 84185 | 85901 | 32994 | 33252 | 79535 | 87870 |
| translation_of | 5071 | 6902 | 4487 | 5025 | 1071 | 1138 | 5415 | 6601 |
| Total | 157465 | 4518927 | 192028 | 2165927 | 101779 | 786684 | 220977 | 4567368 |
| | Computer science | | Chemistry | | | | | |
| | Ent# | Tri# | Ent# | Tri# | | | | |
| author_of | 216180 | 348282 | 397169 | 795356 | | | | |
| first_author | 159209 | 109004 | 295670 | 212199 | | | | |
| second_author | 158493 | 116608 | 291230 | 211992 | | | | |
| co_author | 89562 | 314808 | 173179 | 943688 | | | | |
| research_interest | 117962 | 793333 | 212420 | 1881048 | | | | |
| affiliation | 75704 | 216821 | 153608 | 527827 | | | | |
| belged_domain | 34291 | 34290 | 43944 | 43944 | | | | |
| co-occurrence_with | 34230 | 574754 | 43861 | 892135 | | | | |
| published_year | 102943 | 103891 | 113915 | 116198 | | | | |
| contained_cky | 138 | 90 | 310 | 206 | | | | |
| contained_eky | 21 | 15 | 4 | 2 | | | | |
| other_author | 131788 | 122691 | 293307 | 371261 | | | | |
| all_authors | 515 | 289 | 2394 | 1516 | | | | |
| publi_journal | 126325 | 127415 | 221285 | 224906 | | | | |
| translation_of | 9415 | 11325 | 11978 | 17717 | | | | |
| Total | 262255 | 2873616 | 462361 | 6239995 | | | | |

Table 25: Detailed cross-domain duplicate author names' statistics in TechKG10

| domain | total # | dup # | dup ratio | *dpa* |
|---|---|---|---|---|
| Computer science | 89387 | 70549 | 78.93% | 6.86 |
| Mechanics | 15048 | 13532 | 89.93% | 8.77 |
| Power | 50039 | 39503 | 78.94% | 7.52 |
| Publishing-related | 9904 | 6993 | 70.61% | 11.54 |
| Forestry | 38681 | 27515 | 71.13% | 8.08 |
| Industrial technology | 44696 | 38142 | 85.34% | 7.97 |
| Math-related | 23373 | 18479 | 79.06% | 8.59 |
| Environment-related | 62044 | 51705 | 83.34% | 7.4 |
| Fisheries-related | 17437 | 13597 | 77.98% | 8.66 |
| Animal Husbandry | 47762 | 36731 | 76.90% | 7.4 |
| Architecture-related | 93125 | 64107 | 68.84% | 7.08 |
| Aerospace | 35023 | 28305 | 80.82% | 8.49 |
| Light Industry | 70769 | 51534 | 72.82% | 7.31 |
| Finance | 51907 | 37670 | 72.57% | 8.39 |
| Physics-related | 53626 | 42760 | 79.74% | 7.36 |
| Traffic transportation | 62965 | 44654 | 70.92% | 7.79 |
| History&Geography | 7698 | 5608 | 72.85% | 11.02 |
| Agriculture-related | 169808 | 120635 | 71.04% | 5.67 |
| Machinery | 89371 | 72548 | 81.18% | 6.9 |
| Chemistry | 172855 | 119500 | 69.13% | 5.84 |
| Law | 4805 | 3403 | 70.82% | 12.36 |
| Energy-related | 73979 | 50612 | 68.41% | 7.18 |
| Astronomy | 108545 | 77331 | 71.24% | 6.45 |
| Education | 33939 | 23803 | 70.13% | 9.68 |
| Power industry | 71542 | 48331 | 67.56% | 7.71 |
| Art-related | 2544 | 1466 | 57.63% | 13.53 |
| Material science | 32722 | 29122 | 89.00% | 8.04 |
| Metallurgical industry | 93600 | 66686 | 71.25% | 6.97 |
| Electric industry | 90557 | 68113 | 75.22% | 6.93 |
| Politics-related | 10281 | 7089 | 68.95% | 12.14 |
| Military | 4142 | 3728 | 90.00% | 9.41 |
| Medicine-related | 486052 | 185193 | 38.10% | 5.01 |
| Bioscience | 102819 | 91762 | 89.25% | 5.82 |
| Mining industry | 48482 | 37025 | 76.37% | 7.78 |
| Culture-related | 5891 | 3442 | 58.43% | 12.71 |
| Sport-related | 17414 | 10714 | 61.53% | 10.89 |
| Total | 1206302 | 425357 | 35.26% | 3.79 |

Table 26: Detailed distribution of domain numbers for cross-domain duplicated authors in TechKG10.

| domain | Ratio(%) | | | | | number | | | | |
|---|---|---|---|---|---|---|---|---|---|---|
| | [1,5) | [5,10) | [10,20) | [20,30) | [30,36) | [1,5) | [5,10) | [10,20) | [20,30) | [30,36) |
| Computer science | 60.16% | 20.55% | 13.60% | 4.88% | 0.82% | 42439 | 14496 | 9595 | 3443 | 576 |
| Mechanics | 50.58% | 23.50% | 14.03% | 8.68% | 3.21% | 6845 | 3180 | 1898 | 1175 | 434 |
| Power | 57.74% | 19.85% | 14.14% | 6.82% | 1.44% | 22810 | 7842 | 5585 | 2696 | 570 |
| Publishing-related | 42.49% | 17.19% | 17.99% | 15.54% | 6.79% | 2971 | 1202 | 1258 | 1087 | 475 |
| Forestry | 56.41% | 18.96% | 14.13% | 8.46% | 2.04% | 15522 | 5217 | 3887 | 2328 | 561 |
| Industrial technology | 52.84% | 23.87% | 14.47% | 7.31% | 1.50% | 20154 | 9106 | 5521 | 2789 | 572 |
| Math-related | 53.00% | 20.70% | 14.37% | 9.07% | 2.86% | 9794 | 3826 | 2655 | 1676 | 528 |
| Environment-related | 56.85% | 22.02% | 13.93% | 6.09% | 1.11% | 29392 | 11385 | 7205 | 3147 | 576 |
| Fisheries-related | 53.78% | 20.09% | 13.66% | 8.94% | 3.53% | 7312 | 2732 | 1857 | 1216 | 480 |
| Animal Husbandry | 60.07% | 18.46% | 13.01% | 6.90% | 1.55% | 22066 | 6780 | 4779 | 2536 | 570 |
| Architecture-related | 58.50% | 20.72% | 14.55% | 5.34% | 0.90% | 37502 | 13281 | 9325 | 3423 | 576 |
| Aerospace | 52.00% | 21.38% | 15.73% | 8.85% | 2.03% | 14719 | 6053 | 4453 | 2504 | 576 |
| Light Industry | 58.24% | 20.06% | 14.43% | 6.15% | 1.12% | 30012 | 10339 | 7436 | 3171 | 576 |
| Finance | 51.76% | 20.18% | 18.01% | 8.51% | 1.53% | 19498 | 7603 | 6786 | 3207 | 576 |
| Physics-related | 58.64% | 20.10% | 13.47% | 6.45% | 1.34% | 25073 | 8595 | 5758 | 2759 | 575 |
| Traffic transportation | 54.88% | 20.84% | 15.96% | 7.03% | 1.29% | 24506 | 9304 | 7129 | 3140 | 575 |
| History& Geography | 44.01% | 20.11% | 15.60% | 13.11% | 7.17% | 2468 | 1128 | 875 | 735 | 402 |
| Agriculture-related | 69.20% | 17.79% | 9.63% | 2.91% | 0.48% | 83475 | 21463 | 11613 | 3508 | 576 |
| Machinery | 59.02% | 21.96% | 13.56% | 4.68% | 0.79% | 42815 | 15930 | 9834 | 3393 | 576 |
| Chemistry | 66.69% | 19.59% | 10.28% | 2.96% | 0.48% | 79700 | 23405 | 12287 | 3532 | 576 |
| Law | 41.96% | 15.93% | 16.28% | 15.28% | 10.55% | 1428 | 542 | 554 | 520 | 359 |
| Energy-related | 59.46% | 19.03% | 14.30% | 6.07% | 1.14% | 30092 | 9632 | 7238 | 3074 | 576 |
| Astronomy | 63.51% | 19.35% | 12.01% | 4.39% | 0.74% | 49115 | 14960 | 9288 | 3392 | 576 |
| Education | 45.75% | 19.75% | 20.35% | 11.73% | 2.42% | 10890 | 4702 | 4844 | 2792 | 575 |
| Power industry | 54.75% | 21.26% | 16.09% | 6.71% | 1.19% | 26461 | 10275 | 7776 | 3243 | 576 |
| Art-related | 36.49% | 17.39% | 16.78% | 16.71% | 12.62% | 535 | 255 | 246 | 245 | 185 |
| Material science | 53.84% | 22.78% | 14.01% | 7.48% | 1.88% | 15680 | 6635 | 4080 | 2179 | 548 |
| Metallurgical industry | 58.89% | 21.60% | 13.61% | 5.04% | 0.86% | 39271 | 14404 | 9076 | 3360 | 575 |
| Electric industry | 59.58% | 20.77% | 13.79% | 5.02% | 0.85% | 40581 | 14146 | 9394 | 3416 | 576 |
| Politics-related | 40.37% | 16.18% | 18.90% | 17.24% | 7.31% | 2862 | 1147 | 1340 | 1222 | 518 |
| Military | 57.54% | 13.44% | 11.61% | 10.25% | 7.16% | 2145 | 501 | 433 | 382 | 267 |
| Medicine-related | 73.55% | 16.69% | 7.53% | 1.92% | 0.31% | 136207 | 30911 | 13947 | 3552 | 576 |
| Bioscience | 69.52% | 16.46% | 9.77% | 3.63% | 0.63% | 63795 | 15101 | 8961 | 3329 | 576 |
| Mining industry | 56.14% | 20.35% | 14.53% | 7.44% | 1.54% | 20787 | 7535 | 5378 | 2755 | 570 |
| Culture-related | 41.02% | 14.70% | 17.05% | 16.44% | 10.78% | 1412 | 506 | 587 | 566 | 371 |
| Sport-related | 43.71% | 17.37% | 19.18% | 14.87% | 4.87% | 4683 | 1861 | 2055 | 1593 | 522 |
| Total | 85.45% | 10.17% | 3.41% | 0.84% | 0.14% | 363450 | 43259 | 14520 | 3552 | 576 |

Table 27: Extreme analysis of in-domain duplicate name issuefor authors in TechKG10.

| | | Worse/better | | |
|---|---|---|---|---|
| domain | total # | dup # | dupRat(%) | apa |
| Computer science | 69808 | 32725/31793 | 46.88/45.54 | 3.72/3.76 |
| Mechanics | 12928 | 5496/5073 | 42.51/39.24 | 3.07/3.14 |
| Power | 40670 | 17666/16582 | 43.44/40.77 | 3.37/3.44 |
| Publishing-related | 1950 | 306/301 | 15.69/15.44 | 2.31/2.32 |
| Forestry | 15497 | 7060/6715 | 45.56/43.33 | 2.98/3.02 |
| Industrial technology | 31374 | 10550/10033 | 33.63/31.98 | 2.81/2.84 |
| Math-related | 22409 | 10807/10526 | 48.23/46.97 | 3.29/3.31 |
| Environment-related | 22176 | 8690/8519 | 39.19/38.42 | 2.86/2.87 |
| Fisheries-related | 14774 | 7772/7511 | 52.61/50.84 | 4.04/4.1 |
| Animal Husbandry | 24208 | 11906/11490 | 49.18/47.46 | 3.35/3.39 |
| Architecture-related | 55937 | 28077/27354 | 50.19/48.90 | 3.66/3.69 |
| Aerospace | 22530 | 9754/8905 | 43.29/39.53 | 2.96/3.01 |
| Light Industry | 46657 | 20748/19864 | 44.47/42.57 | 3.42/3.48 |
| Finance | 25231 | 8824/8414 | 34.97/33.35 | 2.74/2.77 |
| Physics-related | 34738 | 16445/15787 | 47.34/45.45 | 3.38/3.43 |
| Traffic transportation | 37361 | 15044/14237 | 40.27/38.11 | 3.08/3.13 |
| History&Geography | 2169 | 512/504 | 23.61/23.24 | 2.37/2.38 |
| Agriculture-related | 117754 | 70229/68518 | 59.64/58.19 | 4.46/4.51 |
| Machinery | 47033 | 18185/17183 | 38.66/36.53 | 3.12/3.16 |
| Chemistry | 140435 | 75276/72807 | 53.60/51.84 | 4.35/4.43 |
| Law | 1535 | 188/164 | 12.25/10.68 | 2.1/2.11 |
| Energy-related | 58931 | 34054/32973 | 57.79/55.95 | 4.53/4.61 |
| Astronomy | 60186 | 34788/34098 | 57.80/56.65 | 3.96/3.99 |
| Education | 10297 | 1801/1591 | 17.49/15.45 | 2.42/2.45 |
| Power industry | 57696 | 27753/26442 | 48.10/45.83 | 3.72/3.79 |
| Art-related | 228 | 7/6 | 3.07/2.63 | 2.14/2.17 |
| Material science | 27841 | 11921/11271 | 42.82/40.48 | 3.13/3.17 |
| Social Science | 66788 | 23748/22618 | 35.56/33.87 | 2.91/2.94 |
| Metallurgical industry | 70179 | 34181/32597 | 48.71/46.45 | 3.89/3.97 |
| Nature Science | 221725 | 120452/118129 | 54.32/53.28 | 3.96/4 |
| Electric industry | 68297 | 31671/30266 | 46.37/44.32 | 3.69/3.75 |
| Politics-related | 1598 | 158/112 | 9.89/7.01 | 2.13/2.06 |
| Military | 2648 | 1167/1083 | 44.07/40.90 | 3.33/3.43 |
| Medicine-related | 306737 | 187416/177696 | 61.10/57.93 | 5.86/6.04 |
| Bioscience | 84376 | 47907/46723 | 56.78/55.37 | 3.95/4 |
| Mining industry | 22355 | 8401/7766 | 37.58/34.74 | 2.96/3.02 |
| Culture-related | 896 | 114/82 | 12.72/9.15 | 2.2/2.24 |
| Sport-related | 11257 | 5383/5139 | 47.82/45.65 | 3.22/3.27 |

Table 28: Detailed cross-domain duplicate name statistics for keywords in TechKG10. (It should be noted that in TechKG10, all of its keywords rank in top-10% tf*idf values).

| domain | total # | dup # | ratio | *dpk* |
|---|---|---|---|---|
| Computer science | 34219 | 15800 | 46.17% | 3.05 |
| Mechanics | 3350 | 1575 | 47.01% | 3.65 |
| Power | 14892 | 7265 | 48.78% | 3.56 |
| Publishing-related | 13671 | 5384 | 39.38% | 3.61 |
| Forestry | 12359 | 6318 | 51.12% | 3.35 |
| Industrial technology | 12722 | 7577 | 59.56% | 4.08 |
| Math-related | 9193 | 2227 | 24.22% | 2.87 |
| Environment-related | 16027 | 8902 | 55.54% | 3.67 |
| Fisheries-related | 5639 | 2587 | 45.88% | 3.65 |
| Animal Husbandry | 17929 | 9238 | 51.53% | 3.18 |
| Architecture-related | 52283 | 20046 | 38.34% | 3.29 |
| Aerospace | 11971 | 5460 | 45.61% | 3.8 |
| Light Industry | 35354 | 14113 | 39.92% | 3.61 |
| Finance | 71987 | 31860 | 44.26% | 3.01 |
| Physics-related | 14462 | 5350 | 36.99% | 2.95 |
| Traffic transportation | 36161 | 14771 | 40.85% | 3.43 |
| History&Geography | 16478 | 8205 | 49.79% | 3.4 |
| Agriculture-related | 48122 | 22655 | 47.08% | 3.12 |
| Machinery | 29515 | 14879 | 50.41% | 3.58 |
| Chemistry | 44546 | 17759 | 39.87% | 3.3 |
| Law | 11388 | 5560 | 48.82% | 2.93 |
| Energy-related | 19517 | 7708 | 39.49% | 3.4 |
| Astronomy | 38313 | 14503 | 37.85% | 3.17 |
| Education | 61306 | 27910 | 45.53% | 3.04 |
| Power industry | 25821 | 9375 | 36.31% | 3.7 |
| Art-related | 13858 | 7283 | 52.55% | 3.52 |
| Material science | 6457 | 3794 | 58.76% | 3.62 |
| Metallurgical industry | 33023 | 11992 | 36.31% | 3.53 |
| Electric industry | 39901 | 17472 | 43.79% | 3.05 |
| Politics-related | 35874 | 23125 | 64.46% | 3.05 |
| Military | 3482 | 2307 | 66.26% | 3.25 |
| Medicine-related | 116379 | 16020 | 13.77% | 2.94 |
| Bioscience | 17462 | 11972 | 68.56% | 3.09 |
| Mining industry | 19104 | 8906 | 46.62% | 3.67 |
| Culture-related | 24144 | 16722 | 69.26% | 3.13 |
| Sport-related | 10788 | 3514 | 32.57% | 3.46 |
| Total | 722298 | 154735 | 21.42% | 2.65 |

Table 29: Detailed domain numbers distribution for cross-domain duplicated keywords in TechKG10.

| | number | | | | | ratio | | | | |
|---|---|---|---|---|---|---|---|---|---|---|
| Domain | [1,5) | [5,10) | [10,20) | [20,30) | [30,36) | [1,5) | [5,10) | [10,20) | [20,30) | [30,36) |
| Computer | 12181 | 3399 | 214 | 6 | 0 | 77.09% | 21.51% | 1.35% | 0.04% | 0.00% |
| Mechanis | 1034 | 491 | 46 | 4 | 0 | 65.65% | 31.17% | 2.92% | 0.25% | 0.00% |
| Power | 5013 | 2003 | 242 | 7 | 0 | 69.00% | 27.57% | 3.33% | 0.10% | 0.00% |
| Publising | 3345 | 1963 | 75 | 1 | 0 | 62.13% | 36.46% | 1.39% | 0.02% | 0.00% |
| Forestry | 4431 | 1814 | 70 | 3 | 0 | 70.13% | 28.71% | 1.11% | 0.05% | 0.00% |
| Industrial | 4401 | 2839 | 330 | 7 | 0 | 58.08% | 37.47% | 4.36% | 0.09% | 0.00% |
| Math | 1868 | 322 | 33 | 4 | 0 | 83.88% | 14.46% | 1.48% | 0.18% | 0.00% |
| Environe | 5511 | 3223 | 163 | 5 | 0 | 61.91% | 36.21% | 1.83% | 0.06% | 0.00% |
| Fisheries | 1584 | 972 | 30 | 1 | 0 | 61.23% | 37.57% | 1.16% | 0.04% | 0.00% |
| Animal | 6539 | 2649 | 49 | 1 | 0 | 70.78% | 28.68% | 0.53% | 0.01% | 0.00% |
| Architecr | 14559 | 5088 | 392 | 7 | 0 | 72.63% | 25.38% | 1.96% | 0.03% | 0.00% |
| Aerospace | 3424 | 1840 | 190 | 6 | 0 | 62.71% | 33.70% | 3.48% | 0.11% | 0.00% |
| LightIndu | 9231 | 4528 | 347 | 7 | 0 | 65.41% | 32.08% | 2.46% | 0.05% | 0.00% |
| Finance | 24445 | 7242 | 169 | 4 | 0 | 76.73% | 22.73% | 0.53% | 0.01% | 0.00% |
| Physics | 4287 | 986 | 72 | 5 | 0 | 80.13% | 18.43% | 1.35% | 0.09% | 0.00% |
| Traffic | 10294 | 4112 | 358 | 7 | 0 | 69.69% | 27.84% | 2.42% | 0.05% | 0.00% |
| History& | 5393 | 2768 | 44 | 0 | 0 | 65.73% | 33.74% | 0.54% | 0.00% | 0.00% |
| Agricue | 16756 | 5699 | 193 | 7 | 0 | 73.96% | 25.16% | 0.85% | 0.03% | 0.00% |
| Machiner | 9721 | 4797 | 354 | 7 | 0 | 65.33% | 32.24% | 2.38% | 0.05% | 0.00% |
| Chemistry | 12702 | 4736 | 314 | 7 | 0 | 71.52% | 26.67% | 1.77% | 0.04% | 0.00% |
| Law | 4298 | 1239 | 23 | 0 | 0 | 77.30% | 22.28% | 0.41% | 0.00% | 0.00% |
| Energy | 5699 | 1740 | 262 | 7 | 0 | 73.94% | 22.57% | 3.40% | 0.09% | 0.00% |
| Astronoy | 10655 | 3666 | 176 | 6 | 0 | 73.47% | 25.28% | 1.21% | 0.04% | 0.00% |
| Education | 21090 | 6655 | 162 | 3 | 0 | 75.56% | 23.84% | 0.58% | 0.01% | 0.00% |
| Power | 6142 | 2887 | 339 | 7 | 0 | 65.51% | 30.79% | 3.62% | 0.07% | 0.00% |
| Art- | 4628 | 2613 | 42 | 0 | 0 | 63.55% | 35.88% | 0.58% | 0.00% | 0.00% |
| Material | 2418 | 1297 | 76 | 3 | 0 | 63.73% | 34.19% | 2.00% | 0.08% | 0.00% |
| Metal | 8241 | 3397 | 347 | 7 | 0 | 68.72% | 28.33% | 2.89% | 0.06% | 0.00% |
| Electronic | 13511 | 3682 | 273 | 6 | 0 | 77.33% | 21.07% | 1.56% | 0.03% | 0.00% |
| Politics | 17257 | 5772 | 94 | 2 | 0 | 74.62% | 24.96% | 0.41% | 0.01% | 0.00% |
| Military | 1576 | 721 | 10 | 0 | 0 | 68.31% | 31.25% | 0.43% | 0.00% | 0.00% |
| Medicine | 12515 | 3386 | 114 | 5 | 0 | 78.12% | 21.14% | 0.71% | 0.03% | 0.00% |
| Bioscie | 8777 | 3146 | 49 | 0 | 0 | 73.31% | 26.28% | 0.41% | 0.00% | 0.00% |
| Mining | 5970 | 2612 | 317 | 7 | 0 | 67.03% | 29.33% | 3.56% | 0.08% | 0.00% |
| Culture | 12061 | 4621 | 40 | 0 | 0 | 72.13% | 27.63% | 0.24% | 0.00% | 0.00% |
| Sport | 2392 | 1075 | 45 | 2 | 0 | 68.07% | 30.59% | 1.28% | 0.06% | 0.00% |
| Total | 131970 | 22284 | 474 | 7 | 0 | 85.29% | 14.40% | 0.31% | 0.00% | 0.00% |

Table 30: Detailed triplet statistics for TechKG10.

| Domain | 1-1 % | 1-n % | ñ | m-1 % | | m-n % | ū(m/n) |
|---|---|---|---|---|---|---|---|
| Computer science | 0.01% | 0.00% | \ | 17.49% | 6.65 | 82.50% | 1.94 |
| Mechanics | 0.00% | 0.00% | \ | 20.92% | 5.18 | 79.08% | 1.64 |
| Power | 0.01% | 0.00% | \ | 14.80% | 5.32 | 85.19% | 1.57 |
| Publishing-related | 0.41% | 0.00% | \ | 11.73% | 5.49 | 87.86% | 1.6 |
| Forestry | 0.01% | 0.00% | \ | 15.41% | 6.47 | 84.58% | 1.71 |
| Industrial technology | 0.01% | 0.00% | \ | 18.71% | 4.85 | 81.27% | 1.57 |
| Math-related | 0.01% | 0.00% | \ | 25.36% | 5.43 | 74.63% | 1.69 |
| Environment-related | 0.02% | 0.00% | \ | 15.68% | 5.77 | 84.31% | 1.78 |
| Fisheries-related | 0.00% | 0.00% | \ | 11.72% | 5.79 | 88.28% | 1.55 |
| Animal Husbandry | 0.01% | 0.00% | \ | 11.32% | 6.51 | 88.67% | 1.59 |
| Architecture-related | 0.03% | 0.00% | \ | 12.86% | 5.99 | 87.11% | 1.82 |
| Aerospace | 0.01% | 0.00% | \ | 17.78% | 5.58 | 82.21% | 1.84 |
| Light Industry | 0.04% | 0.00% | \ | 12.93% | 6.03 | 87.03% | 1.75 |
| Finance | 0.04% | 0.00% | \ | 7.41% | 6.18 | 92.55% | 1.68 |
| Physics-related | 0.01% | 0.00% | \ | 14.94% | 5.86 | 85.05% | 1.61 |
| Traffic transportation | 0.03% | 0.00% | \ | 16.59% | 6.9 | 83.38% | 1.81 |
| History&Geography | 0.09% | 0.00% | \ | 9.46% | 7.51 | 90.46% | 1.34 |
| Agriculture-related | 0.01% | 0.00% | \ | 10.28% | 6.5 | 89.72% | 1.82 |
| Machinery | 0.01% | 0.00% | \ | 17.42% | 6.47 | 82.57% | 1.95 |
| Chemistry | 0.00% | 0.00% | \ | 12.99% | 6.67 | 87.00% | 1.92 |
| Law | 1.39% | 0.47% | 1.96 | 10.99% | 9.6 | 87.16% | 1.46 |
| Energy-related | 0.01% | 0.00% | \ | 12.11% | 5.75 | 87.88% | 1.67 |
| Astronomy | 0.00% | 0.00% | \ | 13.09% | 6.84 | 86.91% | 1.97 |
| Education | 0.39% | 0.00% | \ | 5.32% | 7.27 | 94.29% | 1.54 |
| Power industry | 0.03% | 0.00% | \ | 16.17% | 6.64 | 83.80% | 1.8 |
| Art-related | 0.55% | 0.28% | 2.8 | 3.50% | 64.33 | 95.67% | 1.3 |
| Material science | 0.00% | 0.00% | \ | 15.29% | 4.69 | 84.71% | 1.49 |
| Metallurgical industry | 0.02% | 0.00% | \ | 12.91% | 5.7 | 87.07% | 1.79 |
| Electric industry | 0.00% | 0.00% | \ | 15.66% | 6.42 | 84.33% | 1.87 |
| Politics-related | 0.55% | 0.20% | 1.75 | 3.70% | 8.6 | 95.55% | 1.42 |
| Military | 0.01% | 0.00% | 2 | 14.00% | 6.08 | 85.99% | 1.35 |
| Medicine-related | 0.01% | 0.00% | \ | 9.21% | 8.53 | 90.78% | 2.21 |
| Bioscience | 0.00% | 0.00% | \ | 12.97% | 5.26 | 87.02% | 1.58 |
| Mining industry | 0.02% | 0.00% | \ | 17.30% | 5.89 | 82.68% | 1.74 |
| Culture-related | 0.15% | 0.00% | \ | 2.33% | 6.91 | 97.51% | 1.38 |
| Sport-related | 0.05% | 0.00% | \ | 12.72% | 6.03 | 87.23% | 1.73 |
| Total | 0.05% | 0.01% | 1.99 | 11.63% | 6.54 | 88.32% | 1.83 |