# OpenReview forum: "TechKG: A Large-Scale Chinese Technology-Oriented Knowledge Graph"
_ICLR.cc/2020/Conference — Reject_

### Official Review · AnonReviewer3 · 2019-10-23
**Official Blind Review #3**

**Rating:** 1

**Review:**


This paper introduces a large-scale knowledge graph database called TechKG, which is constructed from a massive repository of academic papers in Chinese. The authors have described in details the process and heuristics in use for constructing such knowledge base, and also reported important statistics and characteristics of the database, including duplicate name, imbalance issue etc. The main contribution is to provide a KG in Chinese, which wasn’t widely available compared to popular languages such as English.

The paper does not clearly demonstrate experiment showing how this new knowledge graph can be complementary to or benefit existing realm of machine learning tasks. How much does it help improve KG-dependent task in Chinese language, or in multi-lingual setting? Without these experiments, it’s hard to estimate precisely the contribution of the KG database to the machine learning community.

In Section 3.1, the authors should specify more clearly the source of the journals collected & the representativeness of the repository.

In Section 3.3, how is the threshold defined to determine the hierarchical relation from k1 to k2?

The writing can be polished further. Things to improve the paper that did not impact the score:

Page 1: Great progress have been -> great progress has been
Page 1: most of existing -> many existing
Section 2, paragraph 4: lack space between `WordNet` and `Miller`, also need adding space between `YAGO` and `suchanek`



**Experience Assessment:**

I do not know much about this area.

**Review Assessment: Checking Correctness Of Derivations And Theory:**

N/A

**Review Assessment: Checking Correctness Of Experiments:**

I carefully checked the experiments.

**Review Assessment: Thoroughness In Paper Reading:**

I read the paper thoroughly.

---

> ### Author Response · Authors · 2019-11-09
> **Response to Reviewer #3**
>
> We would like to thank the reviewer for the thorough and valuable feedback for the manuscript. But we want to clarify some misunderstandings of our paper.
>
> 1. The comment "The paper does not clearly demonstrate experiment showing how this new knowledge graph can be complementary to or benefit existing realm of machine learning tasks." is not accurate. Our experiments(on knowledge graph embedding, distantly supervised RE, and distantly supervised NER) show that TechKG is a more challenging dataset. This indicates that many state-of-the-art methods have to be improved to achieve a similar result as they achieved on some existing KGs. We think this is very important for increasing the robust and transplant of some methods. Obviously, this is of much benefit to existing realm of machine learning tasks.
>
> 2. Currently, many KG-related applications (such as knowledge graph embedding,  distantly supervised RE, knowledge based QA, etc) use non-Chinese KGs. Researchers won't know whether a model that performs well on these existing KGs could still perform well on other-language KGs. TechKG provides a new choice to evaluate the true ability of a model. This is a very important contribution of TechKG.
>
> 3. Table 8 shows that TechKG is a very useful completementary to current existing KGs. First, it has similar scale as existing KGs. Second, it has its own characteristics. Third, it can suport lots of AI-related applications.  Fourth, our primary experiments show that TechKG is a more challenging KG. TechKG is of much benefitcal for researchers to propose more robust models on the corresponding applications.

---

### Official Review · AnonReviewer1 · 2019-10-23
**Official Blind Review #1**

**Rating:** 1

**Review:**

I have read the author response, thank you for responding.

Original review:
This paper presents the extraction of a bibliographic database of Chinese technical papers.  This database could potentially be a valuable resource for the community.  However, the paper is mis-targeted to the ICLR conference, as it does not discuss learning representations or using deep learning (the 36-page appendix does include some material on knowledge graph embedding, but this is not covered in the main body of the paper).  Also, for important tasks like name deduplication, the paper does not discuss or compare against techniques from previous work, and instead proposes a small set of heuristics, which is a sensible approach for building a resource but will not be of interest to the ICLR audience.

The paper refers to their resource as a knowledge graph, but I would say it is more accurate to call it a bibliographic database (it consists primarily of paper titles, authors, and keywords).  This is very different from the broader KBs and KGs discussed in the related work.  YAGO, Freebase, Cyc, and so on capture a much wider variety of semantic relationships and entity types.  I think re-positioning this submission as being aimed at building a bibliographic database, rather than a Knowledge Graph, would help ensure it reaches the right audience.  Also, targeting a different venue like a digital libraries conference and making a strong case that this set of bibliographic data offers advantages in coverage or accuracy over other comparable resources, if any, would help.

Finally, one of the most valuable features of a bibliographic database is the citation graph, it would be exciting if TechKG could be extended to include citation information.

Minor:
Section 2, most of these citations should not be shortcites but should be regular full cites.  For example in the first paragraph, Miller (1995) should be (Miller, 1995).


**Experience Assessment:**

I have published one or two papers in this area.

**Review Assessment: Checking Correctness Of Derivations And Theory:**

N/A

**Review Assessment: Checking Correctness Of Experiments:**

I assessed the sensibility of the experiments.

**Review Assessment: Thoroughness In Paper Reading:**

I made a quick assessment of this paper.

---

> ### Author Response · Authors · 2019-11-09
> **Response to Reviewer #1**
>
> We would like to thank the reviewer for the thorough and valuable feedback for the manuscript. But we want to clarify some misunderstandings of our paper.
>
> 1.We claim in our paper that TechKG is a new Chinese KG and it can be used as a dataset for many diverse AI-related applications. In our primpary experiments in the appendix part, we use three completely different applications to show  the adaptability of TechKG. All of these applications are related to deep learning. Especially, the "knowledge graph embedding" application is directly related to representation learning. So we think that our paper matches the scope of ICLR well.
>
> 2. "name deduplication" is not an important task in TechKG, instead, it is just one of the important characteristics in TechKG (section 4, page 5). Totally, there are three important characteristics that make TechKG be a more challenging dataset for many applications. For example, in the "knowledge graph embedding" application, the ConvE model, one of the state-of-the-art method, achieves very poor results. We analyzed the reason in page14.
>
> 3. The comment that "This is very different from the broader KBs and KGs discussed in the related work. " is right. In Table 8 (page 8) of our paper, we compare the differences between different KGs. TechKG is a very different KG compared with current existing KGs mainly due to the following two reason. First, TechKG is technology-oriented KG, while other KGs are general purpose. Second, TechKG takes technical papers as data source, while most of other KGs take Wikipedia as datasource. Because of these two reasons, both the semantic relationships and the entity types are proper in TechKG.
>
> 4. TechKG is a completely different KGs and it provides a new choice for lots of applications: the AI-related applications are diverse, thus there should be diverse datasets. Besides, a lot of new characteristics hierent in TechKG raise new challenges for many existing methods. (page 10, "conclusion" section)

---

### Official Review · AnonReviewer2 · 2019-10-31
**Official Blind Review #3**

**Rating:** 3

**Review:**

The paper talks about creating a Tech Knowledge Graph of 260 million tripes, with 52 million entities coming from 38 research domains. The authors claim this is the first of it's technology specific Chinese KG, so they claim scale, specificity and list out various aspects that makes this distinct to existing KG's.

A few things that the paper is missing or not written well:
1. Related work is pretty much just listing existing KG's but no limitations and how those limitations are addressed by this work other than the fact that tech KG is technology specific. By being technology specific how does it help, where is KG being used and what metrics are current KG's fall short etc.
2. There is no novelty in the way KG is built, so there is no technical contribution to this paper making it a very weak submission for ICLR standards.
3. There is a huge appendix section with some results and lot of information. It would have been great to distill how KG's are being used in existing benchmarks and either show the value of a tech KG in the metrics/benchmarks or introduce a new metric or benchmark to measure the value of a specific domain KG like the tech KG.

If the authors can address these issues, the value of the work can increase significantly.

**Experience Assessment:**

I have published in this field for several years.

**Review Assessment: Checking Correctness Of Derivations And Theory:**

I assessed the sensibility of the derivations and theory.

**Review Assessment: Checking Correctness Of Experiments:**

I assessed the sensibility of the experiments.

**Review Assessment: Thoroughness In Paper Reading:**

I read the paper at least twice and used my best judgement in assessing the paper.

---

> ### Author Response · Authors · 2019-11-09
> **Response to Reviewer #3**
>
> We would like to thank the reviewer for the thorough and valuable feedback for the manuscript. But we want to clarify some misunderstandings of our paper.
>
> 1. Currently, many KG-related applications (such as knowledge graph embedding,  distantly supervised RE, knowledge based QA, etc) use non-Chinese KGs. Researchers won't know whether a model that performs well on these existing KGs could still perform well on other-language KGs. TechKG provides a new choice to evaluate the true ability of a model. This is a very important contribution of TechKG.
>
> 2. The comment "There is no novelty in the way KG is built," is right. But we think that there are three key factors in building a KG like Freebase: scale, cost, and quality. We think these three factors are also the main reason why there are few non-English available KG up to now. TechKG provides a very cheap way to construct a large scale KG with very high quality. The method can be use to construct a KG of any languages. We think his is another important contribution of TechKG.
>
> 3. TechKG is a completely different KGs and it provides a new dataset choice for lots of applications. Our primary experiments show that TechKG is a more challenging dataset for many AI-related applications. TechKG is of much benefitcal for researchers to propose more robust models on these corresponding AI-related applications.

---

### Decision · Program_Chairs · 2019-12-19

**Decision:**

Reject

**Comment:**

This paper presents a large-scale automatically extracted knowledge base in Chinese which contains information about entities and their relations present in academic papers. The authors have collected several papers that come from around 38 different domains. As such this is a dataset creation paper where the authors have used existing methodologies to perform relation extraction in Chinese.

After having read the reviews and followup replies by authors, the main criticisms of the paper still hold. In addition to the lack of technical contribution, I feel that the writing of the paper can be improved a lot, for example, I would like to see a table with some example entities and relations extracted. That said, with further improvements this paper could potentially be a good contribution to LREC which is focused on dataset creation.

In its current form, I recommend the paper to be rejected.